



# Heavy metal uptake of near-shore benthic foraminifera during multi-metal culturing experiments

Sarina Schmidt[1], Edmund Charles Hathorne[1], Joachim Schönfeld[1] and Dieter Garbe-Schönberg[2]

[1]GEOMAR Helmholtz Centre for Ocean Research Kiel, Wischhofstraße 1-3, 24148 Kiel, Germany
[2]Institute of Geosciences, Kiel University, Ludewig-Meyn-Straße 10, 24118 Kiel, Germany

*Correspondence to*: Sarina Schmidt (sschmidt@geomar.de)

**Abstract.** Heavy metal pollution originating from anthropogenic sources, e.g., mining, industry and extensive land use, is increasing in many parts of the world and influences coastal marine environments for a long time. The elevated input of heavy metals into the marine system potentially affects the biota because of their toxicity, persistence and bioaccumulation. An emerging tool for environmental applications is the heavy metal incorporation into foraminiferal tests calcite, which facilitates monitoring of anthropogenic footprints on recent and past environmental systems. The aim of this study is to investigate whether the incorporation of heavy metals in foraminifera is a direct function of their concentration in seawater. Culturing experiments with a mixture of dissolved chromium (Cr), manganese (Mn), nickel (Ni), copper (Cu), zinc (Zn), silver (Ag), cadmium (Cd), tin (Sn), mercury (Hg) and lead (Pb) in artificial seawater were carried out over a wide concentration range to assess the uptake of heavy metals by the near-shore foraminiferal species *Ammonia aomoriensis*, *Ammonia batava* and *Elphidium excavatum*. Seawater analysis exhibited the increasing metal concentrations between culturing phases and revealed high metal concentrations in the beginning of the culturing phases due to the punctual metal addition. Furthermore, a loss of metals during the culturing process was discovered, which lead to a deviation between the expected and the actual concentrations of the metals in seawater. Laser ablation ICP-MS analysis of the newly formed calcite revealed species-specific differences in the incorporation of heavy metals. The foraminiferal calcite of all three species reveals a strong positive correlation with Pb and Ag concentrations in the culturing medium. *Ammonia aomoriensis* further showed a correlation with Mn and Cu, *A. batava* with Mn and Hg and *E. excavatum* with Cr and Ni, and partially also with Hg. Zn, Sn and Cd showed no clear trend for the species studied, which may be caused by the little variation of these metals in seawater. Our calibrations and the calculated partition coefficients render *A. aomoriensis*, *A. batava* and *E. excavatum* as natural archives that enable the direct quantification of metals in polluted and pristine environments. This in turn allows monitoring of the ecosystem status of areas that are potentially under the threat of anthropogenic pollution in order to evaluate contemporary emission reduction measures.

## 1 Introduction

Particular heavy metals e.g., zinc (Zn), iron (Fe), molybdenum (Mo), cobalt (Co) and copper (Cu) serve as micronutrients (e.g., Hänsch and Mendel, 2009) for eukaryotic life and play an important role for metabolism, growth, reproduction and enzymatic activity of organisms (e.g., Martín-González et al., 2005; Gallego et al., 2007). Other metals like mercury (Hg), on the other hand, are not known to have any positive effect on the body and are therefore believed to have a higher toxic potential (Jan et al., 2015). All these metals occur naturally in the environment as geogenic traces in soils, water, rocks and, consequently, in plants and animals. However, at higher





concentrations, most heavy metals become toxic and have hazardous effects on marine biota (Stankovic et al., 2014). Furthermore, they are highly persistent in the marine environment and can be hardly degraded by organisms after the uptake of these metals into their system and cells (Flora et al., 2012; Kennish, 2019). Coastal environments act as natural catchments for anthropogenic pollutants because these areas are directly affected by industry, agriculture and urban runoff (e.g., Alloway, 2013; Julian, 2015; Tansel and Rafiuddin, 2016).

In marginal seas and coastal areas, benthic foraminifera are common, and they can be used as proxies for changing environmental parameters like water temperature (Mg/Ca; e.g., Nürnberg et al., 1995; 1996), salinity (Na/Ca; e.g., Wit et al., 2013, Bertlich et al., 2018), oxygen content or redox conditions (Mn/Ca; Groeneveld and Filipsson, 2013b; Koho et al., 2015; 2017; Kotthoff et al., 2017; Petersen et al., 2018). Foraminifera take up heavy metals and incorporate them into their calcium carbonate shells during calcification (e.g., Boyle, 1981; Rosenthal et al., 1997; Dissard et al., 2010a; 2009; 2010b; Munsel et al., 2010; Nardelli et al., 2016; Frontalini et al., 2018a; 2018b; Titelboim et al., 2018; Smith et al., 2020). Moreover, foraminifera have a short life cycle (< 1 year; e.g., Haake, 1967; Boltovskoy and Lena, 1969; Wefer, 1976; Murray, 1992) and thus, react immediately to changing environmental conditions and contamination levels of the surrounding environment.

Species of the foraminiferal genera *Elphidium* and *Ammonia* are among the most abundant foraminiferal taxa in near-shore environments worldwide. They are found from subtidal water depths to the outer continental shelves (Murray, 1991). Furthermore, their calcite tests are often well preserved in the fossil record (Poignant et al., 2000; McGann, 2008; Xiang et al., 2008) and therefore provide the opportunity to assess past environmental conditions. The combination of all these properties make foraminifera, and especially *Elphidium* and *Ammonia* species, suitable indicators of anthropogenic pollution (e.g., Sen Gupta et al., 1996; Platon et al., 2005). As such, this group of organisms are excellent candidates for monitoring the spatial and temporal distribution of heavy metals in seawater to evaluate, for example, the effectiveness of contemporary measures of reducing emissions caused by anthropogenic inputs.

The majority of culturing studies on heavy metal incorporation into benthic foraminifera were designed to assess the influence and uptake of one particular metal, e.g., copper (Cu) (De Nooijer et al., 2007), chromium (Cr) (Remmelzwaal et al., 2019), lead (Pb) (Frontalini et al., 2015), zinc (Zn) (e.g., Smith et al., 2020), mercury (Hg) (Frontalini et al., 2018a) or cadmium (Cd) (Linshy et al., 2013). This approach is adequate to detail the effects on shell chemistry, growth or physiology. Only one study reported a culturing experiment with elevated levels of Cu, Mn and Ni in the same culturing medium (Munsel et al., 2010). However, there is rarely only one but mostly a combination of several pollutants that occur in nearly all environments affected by heavy metal pollution (e.g., Mutwakil et al., 1997; Cang et al., 2004; Vlahogianni et al., 2007; Huang et al., 2011; Wokhe, 2015; Saha et al., 2017). How foraminifera incorporate and react to heavy metals when they are co-exposed to more than one metal at a time is less constrained to date. A mixture of different metals will lead to interactions, which may result in a more severe damage of tissue than exposure to each of them individually (Tchounwou et al., 2012). For example, a co-exposure to arsenic and cadmium causes a more distinct damage of human kidneys than only one of these elements (Nordberg et al., 2005). Furthermore, a chronic low-dose exposure to multiple elements can cause similar synergistic effects (e.g., Wang et al., 2008). It is therefore reasonable to assume that other organisms are likewise threatened more harmfully when exposed to several potentially toxic elements simultaneously.



Here we present results from culturing studies with *Ammonia aomoriensis*, *Elphidium exvacatum* and *Ammonia*
*batava* addressing the relationship of heavy metal concentration in seawater and foraminiferal tests. The
partitioning factor between concentration of an element in the ambient seawater and the calcium carbonate of the
foraminifers is constrained by determining both the dissolved metal concentrations in water and the metal contents
of individual chambers of the foraminiferal shell that have been precipitated in the culturing medium. In particular,
foraminifera were grown while exposed to a combination of ten different heavy metals, i.e., cadmium (Cd), copper
(Cu), chromium (Cr), lead (Pb), manganese (Mn), mercury (Hg), nickel (Ni), silver (Ag), tin (Sn) and zinc (Zn)
over a range of concentrations that prevail in polluted near-shore environments today. These metals are the most
common representatives of marine heavy metal pollution (Alve, 1995; Martinez-Colon et al., 2009). Once the
carbonate/seawater metal partitioning coefficients are known, investigations of the chemistry of benthic
foraminiferal shells offer a reliable method to monitor short-term changes in the chemistry and bioavailability of
toxic elements in seawater.

**2 Material and Methods**
**2.1 Field sampling**

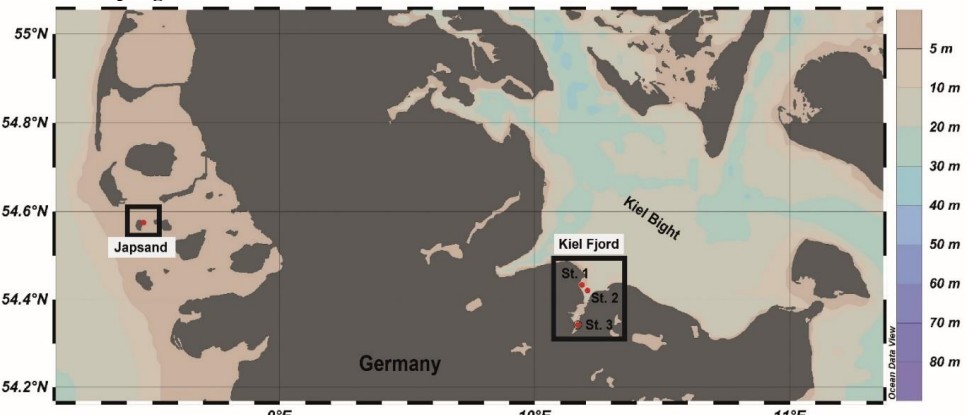


**Figure 1**: Location of the sampling stations in the North Sea (Japsand area) and in the Baltic Sea (Kiel Fjord, St.1
Strander Bucht, St. 2 Laboe, St. 3 Mönkeberg). The map was drawn with Ocean Data View (Schlitzer, 2016) on
the basis of bathymetric data. Depth in m are indicated by the color code.

**2.1.1 North Sea, Japsand**
Living specimens of *A. batava* were collected at the barrier sand Japsand near Hallig Hooge in the German Wadden
Sea in July 2019 at two stations (St. 1: 54°34.480´N, 8°27.919´E; St. 2: 54°34.491´N, 8°27.895´E) (Fig. 1). The
sediment was a glacial till or Eemian clay at Station 1 and fine to medium sand at Station 2. The samples were
recovered during low tide by scrapping off the uppermost centimetre of the surface sediment with a spoon made
out of stainless steel. Natural seawater (NSW) with a salinity of 30.3 PSU was collected near the sites for further
processing of the samples. Once back on Hallig Hooge, the sediment was washed with NSW through stacked
sieves with a mesh size of 2000 and 63 µm. The 2000 µm sieve was used to remove larger organisms and excess
organic material (macroalgae, gastropods, lugworms etc.) that could induce anoxic conditions in the sediment





during transport and storage. The residue was stored in Emsa CLIP and CLOSE® boxes, sparged with air and
some algae food was provided. Back in the laboratory at GEOMAR, the residue was stored at 8 °C in a fridge until
culturing. These stock cultures were fed twice a week with *Nannochloropsis* concentrate (BlueBioTech) and water
was partly exchanged with NSW from the sampling site once a week.

### 2.1.2 Baltic Sea, Kiel Bight

Living specimens of *A. aomoriensis* and *E .excavatum* were collected from different stations in Kiel Fjord, western
Baltic Sea (St.1, Strander Bucht, 54°26.001´N, 10°11.1078´E; St. 2, Laboe, 54°25.254´N, 10°12.346´E; St. 3,
Mönkeberg, 54°20.752´N, 10°10.150´E; water depth: 12.5 m, 12.3 m and 14.3 m, respectively) in September and
October 2019 with F.B. Polarfuchs and F.S. Alkor (Fig. 1). A Rumohr corer (inner diameter 55 mm) was used on
F.B. Polarfuchs and 9 cores were taken (2 at St. 1 and 7 at St. 3). One core at both stations was used for
foraminiferal assemblage analysis and the first 2 cm of the sediment from all other cores was collected in plastic
containers with NSW from the site.
The sediment surface was nearly horizontal and comprised a ~ 5 mm thick fluffy layer consisting of organic
detritus of a dark brownish color. Mussels, worm burrows and plant debris was found. The sediment underneath
the surface layer was a very fine mud. The redox boundary was shallower than 0.5 to 1 cm as indicated by the
color turning black underneath this depth, and the sediment smelling of $H_2S$.
On F.S. Alkor, a Reineck box corer was used (200 x 250 cm) and 3 replicates at each station were taken (St. 1 –
3). The first 1 to 2 cm of the sediment surface of the box core were scrapped off with a spoon made out of stainless
steel and the material was stored in a plastic box with NSW from the location. Additional samples for foraminiferal
assemblage analysis were taken at each station.
Back in the laboratory at GEOMAR, the samples were treated the same as Japsand samples from the North Sea.
Artificial seawater (ASW, Tropic Marin) with a salinity of 30 PSU was used for washing and storage of the surface
samples from Kiel Fjord.

**2.2 Culturing setup**

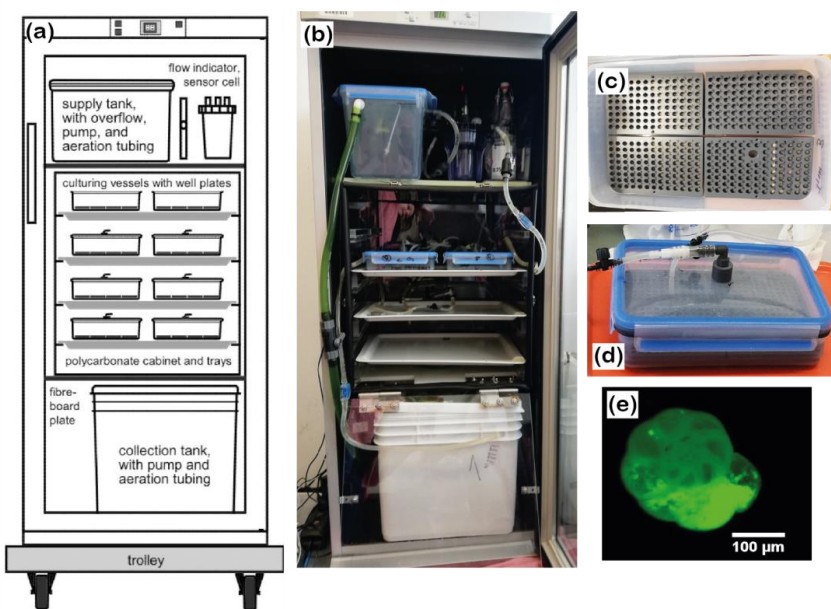

**Figure 2**: Culturing setup. a: conceptual draft (left, modified after Dagan et al., 2016) and b: assembly of the system (right). Tubing and hoses were omitted from the draft for clarity. c: a well plate with mounted specimens and sand, d: closed culturing vessel with well plates and conduits. e: with calcein stained foraminifer under a fluorescence microscope. The chambers formed in calcein are fluorescing brightly.

**2.2.1 Picking of the samples**

For extracting the foraminiferal specimens from the sediment, about 1 cm$^3$ of the 63 to 2000 µm size fraction was transferred to a petri dish. For maintaining optimal conditions for the foraminifera, the petri dish was filled with artificial seawater (ASW) with a salinity of 30 PSU. All living specimens were picked with a paint brush from this subsample and collected in a small petri dish of 55 mm diameter with ASW. The procedure was repeated until the whole sample residue was screened. Only specimens with a glossy, transparent and undamaged test were chosen. Furthermore, only individuals with the cytoplasm present in more than just a couple of chambers that were connected and included the innermost chambers were chosen. After picking, a drop of concentrated food (pure culture of *Nannochloropsis*, green colored algae) was added and the foraminifera were left untouched for a night.

Specimens that met one or more of the following criteria were considered as living and used for further procedures:

- Specimens showed a structural infill of cytoplasm with a bright green color, indicating they took up the food over night,
- they developed a film or strings of pseudopodia firmly sticking to sediment particles or food,
- they had covered themselves or gathered a cyst of sediment or food particles.



Specimens were identified and sorted by species, rinsed, sediment particles were removed, and the color of the
cytoplasm was checked before inserting them into a petri dish with calcein (16 mg l[-1]) (bis[N,N-
bis(carboxymethyl)aminomethyl]-fluorescein) (Sigma-Aldrich) for staining (Fig. 2e). Staining took place
immediately before the individual culturing phase 0 to 3 and lasted for 14 days. Petri dishes were stored at 8 °C in
a fridge, partial water exchanges and feeding of the foraminifera was performed twice a week. After the staining,
the foraminifera were transferred to a petri dish with ASW and left for 1 to 2 days to remove excess calcein from
seawater vacuoles in their cytoplasm prior to the introduction into the culturing system.

### 2.2.2 Culturing system

We used two closed-circulation incubation systems for foraminifera (Fig. 2a, b) provided by the Institute of
Microbiology, Kiel University (Dagan et al., 2016, their Fig. 5; Woehle et al., 2018, their Fig. S4). The systems
were a further development based on earlier closed-circulation systems for foraminiferal culturing (Hintz et al.
2004; Haynert et al., 2011). They were slightly modified for the requirements of this study, but the basic
operational principle is described by Dagan et al., 2016. In detail, the systems consisted of three levels with
different functions. They are built into Bauknecht WLE 885 fridges for temperature control. Each incubation
system accommodated two culturing vessels, which were arranged pairwise on a tray in a polycarbonate cabinet
(Fig. 2a, b). The water was pumped from the collection tank at the lowest level of the fridge to the top level into
the supply tank. From the supply tank, the water was directed to the culturing vessels and the flow was regulated
by 3-way wheels, which ensured that the same amount of water was provided to every culturing vessel. After
passing the culturing vessels, the water was redirected to the collection tank. The systems were filled with 15 L of
ASW with a salinity of 30.5 PSU. The water was aerated in the supply and the collection tank with air from outside
the system, which passed through a 0.2 µm filter. Monitoring of temperature and salinity were performed with a
WTW 3210 conductivity meter. Uncertainty of the conductivity measurements was ± 0.5% and ± 0.1 °C for
temperature according to the manufacturer's test certificate. All parts that were introduced into the system were
sterilized before use either by autoclaving, an UV–lamp exposure, or by applying DanKlorix®.

### 2.2.3 Preparation for incubation

For the incubation of the foraminifera, well plates made from PVC were used (Fig. 2c). Each well plate measured
x 82 mm with 104 cavities of which each one had a diameter of 6 mm and a depth of 5 mm. All well plates
had been used in previous experiments for culturing foraminifera in seawater, which ensured that potentially toxic
substances or additives were already released from the PVC material (Dagan et al., 2016; Woehle et al., 2018) and
should therefore not harm the foraminifera. Before the foraminifera were placed in the cavities, each cavity was
filled with sterile quartz sand up to 1.5 mm. The cavities were subsequently filled with artificial seawater and the
specimens were inserted randomly. Prepared well plates were left untouched for one night, to make sure that the
foraminifera are able to spread their pseudopodial network before incubation. This ensures that they are stable
anchored in the cavities and do not float when the culturing vessels are filled and mounted (Haynert et al., 2011).
Four well plates respectively were assembled in an airtight Emsa CLIP and CLOSE® box of 226 x 167 x 59 mm
(Fig. 2d). Of these four well plates, only three were engrafted with living foraminifera, one species per plate. The
fourth plates were left barren and were used to stabilize the arrangement in the culturing vessels. Each culturing



vessel had a lid with an inflow and an outflow conduit, for which Tygon® tubing were used. The tubing had an
outside diameter of 6.3 mm, and the PVC material was considered as nontoxic because no phthalate-based
emollients were used. To guarantee that the foraminiferal specimens are not flushed away by the incoming water,
the inflow conduit reached almost the bottom of the culturing vessel and was placed between two well plates. The
outflow conduit located at the lid of the vessel was attached to a custom-made PVC fitting (Fig. 2d). Once all well
plates were arranged in the culturing vessel, the lid was equipped with an additional, elastic sealing and closed.
Before the culturing vessels were placed in the incubators, each chamber was slowly filled with ASW. Thereafter,
the culturing chambers were placed on the shelve in the incubator, and were connected to the supply hoses. The
water supply was turned on once the system was completely filled.

### 2.2.4 Culturing experiment

The culturing experiment had four different phases. The first, phase 0 was dedicated as control phase. No heavy
metals were added. This phase allowed both systems to equilibrate in terms of physicochemical and biological
processes and made it possible to determine the background values in terms of seawater constituents. This phase
lasted 21 days. Afterwards, one system was used as the control system, where no heavy metals were added. In the
other system, three phases with elevated heavy metal concentrations were performed. The phases lasted 21 days
each. Tropic Marin Pro-Reef salt was mixed with deionized water for adjusting the salinity. A stock solution
containing all metals of interest was mixed and before each phase, the heavy metal concentration was elevated by
1 to 1.5 order of magnitude by adding an appropriate amount of the stock solution into the supply tank of the
system (see Fig. 2a) (phase 1 = 1 ml, phase 2 = 10 ml, phase 3 = 150 ml). The elevation was performed all at once
and for keeping the metal concentration at the same level over the culturing phase, a smaller amount of the stock
solution was fed into the system bi-weekly. Target concentrations are given in Table 1. The target concentration
of the elements at each phase were chosen after earlier culturing experiments with foraminifers (Mn, Cu, Ni:
Munsel et al., 2010; Pb: Frontallini et al., 2015 & 2018a; Zn: Nardelli et al., 2016; Cd: Linshy et al., 2013; Cu: De
Nooijer et al., 2007; Le Cadre and Debenary et al., 2006; Cr: Remmelzwaal et al., 2019) to resemble conditions
observed in polluted areas like Jakarta Bay (e.g., Williams, 2000) or at European ports (e.g., Fatoki and
Mathabatha, 2001). Furthermore, recommended threshold values provided from the EPA (Environmental
Protection Agency, USA) were taken into account to ensure that the foraminifera are not limited in their growth
and are able to maintain normal physiological functions. However, the heavy metal concentrations in the culturing
media obtained during each phase were monitored by frequent water sampling.
**Table 1**: Heavy metal concentration in the stock solution, target concentration of these metals in each phase and
used salt compounds. All salts used were provided in p.a. quality.

|  | Salt compound | Conc. in mg l$^{-1}$ Stock solution | Target conc. in µg l$^{-1}$ | | |
|---|---|---|---|---|---|
|  |  |  | Phase 1 | Phase 2 | Phase 3 |
| Chromium (Cr) | CrCl$_3$ · 6 H$_2$O | 25 | 0.5 | 5 | 50 |
| Manganese (Mn) | MnCl$_2$ · 4 H$_2$O | 40 | 40 | 400 | 4000 |
| Nickel (Ni) | NiCl$_2$ · 6 H$_2$O | 5 | 0.1 | 1 | 10 |
| Copper (Cu) | CuCl$_2$ * 2 H$_2$O | 2 | 0.05 | 0.5 | 5 |
| Zinc (Zn) | ZnCl$_2$ | 50 | 0.8 | 8 | 80 |
| Cadmium (Cd) | CdCl$_2$ | 4 | 0.08 | 0.8 | 8 |





| | | | | | |
|---|---|---|---|---|---|
| Silver (Ag) | AgNO$_3$ | 3.5 | 0.1 | 1 | 10 |
| Tin (Sn) | SnCl$_2$ * 2 H$_2$O | 10 | 0.1 | 1 | 10 |
| Mercury (Hg) | HgCl$_2$ | 0.04 | 0.01 | 0.1 | 1 |
| Lead (Pb) | PbCl$_2$ | 10 | 0.1 | 1 | 10 |


Over the entire culturing period, both systems were exposed to a natural day and night cycle and the flow rate was
adjusted to 0.017 ml s$^{-1}$ (one drop per second) within the culturing vessels. The foraminifera were fed with
*Nannochloropsis* concentrate twice a week (~ 2000 µg). After21 days (meaning after each culturing phase) one
culturing vessel per system was exchanged. Vessels and specimens were left in the culturing system for the
complete culturing phase (21 days) and no exchange took place during a culturing phase. One culturing vessel
containing all three species was left in the system from the beginning until the end of the experiment (from phase
0 to phase 3) for 84 days. Data of these specimens are not available due to time constraints caused by the outbreak
of the COVID-19 pandemic.
Temperature and salinity were kept stable at 15.0 °C and 30.2 units (trace metals) and at 14.9 °C and 30.4 units
(control) over the complete culturing period. As the system was mostly closed, evaporation had a minor effect.
Demineralized water was added when necessary to keep the salinity stable. The exchanges of culturing vessels
between phases inferred a partial water exchange of approximately 10 % (= 1.5 l) every three weeks, which ensured
a repetitive renewal of water with adequate quality.

**2.3 Water samples**
**2.3.1 Collection of water samples**
Water samples for determining the heavy metal concentrations were taken frequently from the supply tanks (see
Fig. 2a) of both systems using acid cleaned syringes (Norm-Ject® disposable syringe, 20 ml, sterile) and sample
bottles (LLG narrow neck bottles, 50 ml, LDPE = Low Density Polyethylene; Hg: GL 45 Laboratory bottle 250
ml with blue cap and ring, boro 3.3). Water samples to be analysed for mercury concentrations had to be treated
differently due to analytical constraints as detailed below. The water was filtered through a 0.2 µm PES filter
(CHROMAFIL Xtra disposable filters, membrane material: polyether sulfone pore) for heavy metal samples and
through a 0.2 µm quartz filter for Hg samples (HPLC syringe filters, 30 mm glass fibre syringe filters/ nylon).
Filters were rinsed with the sample before the sample was taken. Every water sample was immediately acidified
with concentrated ultrapure HCl to a pH of approximately 2 to avoid changes in the trace metal concentrations due
to adsorption to the sample bottle walls or the formation of precipitates.

**2.3.2 Preparation of water samples before analysis**
For Mn, Zn, Ni, Pb, Cu, and Cd concentration analyses, the water samples were pre-concentrated offline by using
a SeaFAST system (ESI, USA). Twelve ml of each sample were used to fill a 10 mL sample loop and concentrated
by a factor of 25 into 1.5M HNO$_3$. All samples were spiked with indium as an internal standard for monitoring
and correcting for instrumental drift. Both MilliQ water and bottle blanks of acidified MilliQ water (pH ~ 2) stored
in the same bottles until the samples were passed through the pre-concentration system. Additionally, procedural
blanks which were filtered as the samples were also pre-concentrated and measured. A variety of international



(Open Ocean Seawater NASS-6, River Water SLRS-6, Estuarine Seawater SLEW-3, all distributed by NRC-
CNRC Canada) and in-house (South Atlantic surface water, South Atlantic Gyre water) reference materials were
pre-concentrated like the samples. All samples were subsequently analysed by ICP-MS (inductively coupled
plasma mass spectrometry).
For the metals that cannot be preconcentrated by the SeaFAST system as they are not retained on the Nobias resin
(Cr, Ag and Sn) samples were diluted 1/25 and directly introduced into the ICP-MS. The dilution was performed
with indium-spiked nitric acid (2%) and to match the matrix of these samples blanks and standards with added
NaCl were prepared.
All trace metals except mercury were measured using an Agilent 7500ce quadrupole ICP-MS and raw intensities
calibrated with mixed standards, made from single element solutions, covering a wide concentration range.
Additionally, a dilution series (dilution factors: 1, 1/10, 1/100 and 1/1000) of SLRS-6 of river water reference
material (NRC Canada; Yeghicheyan et al., 2019) was measured for quality control. Mean values and relative
standard deviations (RSD) derived from the reference materials are summarised in the appendix (Table A2).
Prior to the measurements of Hg concentrations, all samples were treated with BrCl solution at least 24 hours
before the analysis to guarantee the oxidation and release of mercury species that are possibly present in a different
oxidation states or phases. The BrCl was removed again from the sample by adding hydroxylamine hydrochloride
at least one hour prior to analysis before the Hg was reduced to the volatile $Hg^0$ species with acidic $SnCl_2$ (20 %
w $v^{-1}$) during the measuring process. All preparations of the water samples took place in a Clean Lab within a
trace metal clean atmosphere and all vials were acid cleaned prior to use. Mercury concentrations were determined
using a Total Mercury Manual System (Brooks Rand Model III). The reduced volatile $Hg^0$ is nitrogen-purged onto
a gold-coated trap and released from again by heating before it is measured via cold vapour atomic fluorescence
(CVAFS) under a continuous argon carrier stream. Quality control of the Hg measurements was carried out by
measuring mixed standards, made from single element solutions and confirmed with replicate measurements
throughout each analysis. The measurement uncertainty was smaller than 4.5 % RSD for all analyses.
The calcium concentration of culture seawater was analysed using a VARIAN 720-ES ICP-OES (inductively
coupled plasma optical emission spectrometer). Yttrium was added as an internal spike and samples were diluted
1/10. IAPSO seawater standard (ORIL) was measured after every 15 samples for further quality control which
revealed a measurement uncertainty < 0.35 (RSD %) for the elements analysed (mean Ca concentration IAPSO
this study = 419.6 ± 0.15 mg $l^{-1}$; reference Ca concentration IAPSO Batch 161 = 423 mg $l^{-1}$).

### 2.4 Foraminiferal samples

After every culturing phase, the culturing vessels were taken out of the system and foraminiferal specimens where
collected from their cavities within one day. The individuals were cleaned with tap water and ethanol before they
were mounted in cell slides to mechanically remove salt scale and organic coatings with a paintbrush.
In order to check, whether the foraminifera had grown during the experiment, the total number of chambers was
counted before and after the experiment for every specimen (Table 2). As the foraminifera were stained with
calcein before the experiment, it was possible to cross-check the growth with a fluorescent microscope (Zeiss Axio



Imager 2) if new chambers without color were added, hence whether the particular specimen had grown or not
(Fig. 2e). Only individuals clearly showing new chambers were analysed by Laser ablation ICP – MS.
Prior to the laser ablation analyses, the foraminifera were transferred into individual acid-leached, 500 µl micro-
centrifuge tubes and thoroughly cleaned, applying a procedure adapted from Martin and Lea (2002). The
specimens were rinsed three times with MilliQ water and introduced into the ultrasonic bath for a few seconds at
the lowest power setting after each rinse. Afterwards, clay and adhering particles were removed by rinsing the
sample with Ethanol twice, which was followed by three MilliQ rinses again with minimal ultrasonic treatment.
Oxidative cleaning was applied using 250 µl of a 0.1M NaOH and 0.3 % $H_2O_2$ mixture added to each sample and
the vials were kept for 20 min in a 90 °C water bath. Afterwards, the samples were rinsed with MilliQ three times
to remove the remaining chemicals. The reductive step of the cleaning procedure by Martin and Lea (2002) was
not applied. This step is necessary to remove metal oxides, which of course could also influence the trace metal
concentration within the foraminiferal shell carbonate but these are usually considered to be added during early
deposition (e.g., Boyle, 1983) and therefore unlikely to occur during culture experiments. For Laser Ablation
Inductively Coupled Plasma Mass Spectroscopy (LA-ICP-MS) measurements, all cleaned specimens were fixed
on a double-sided adhesive tape (PLANO).
Micro-analytical analyses with LA-ICP-MS were performed at the Institute of Geosciences, Kiel University, using
a 193nm ArF excimer GeoLasPro HD system (Coherent) with a large volume ablation cell (Zurich-type
LDHCLAC, Fricker et al., 2011) and helium as the carrier gas with 14 mL min$^{-1}$ $H_2$ added prior to passing the
ablation cell. For the foraminiferal samples, the pulse rate was adjusted to 4 to 5 Hz with a fluence between 2 and
3.5 J cm$^{-2}$. The spot size was set to 44 or 60 µm depending on the size of the foraminiferal chamber. All chambers
of a foraminifer that were built up in the culturing medium were analysed, starting from the earliest, inner chamber
adjacent to the calcein-stained chamber. The laser was manually stopped once it broke through the foraminiferal
shell. The ablated material was analysed by an ICP-MS/MS instrument (8900, Agilent Scientific Instruments) in
no gas mode. The NIST SRM 612 glass (Jochum et al., 2011) was used for calibration and monitoring of instrument
drift while NIST SRM 614 was measured for quality control. Glasses were ablated with a pulse rate of 10 pulses
per second, an energy density of 10 J cm$^{-2}$ and a crater size of 60 µm. Carbonate matrix reference materials coral
JCp-1, giant clam JCt-1, limestone ECRM752-1 and synthetic spiked carbonate MACS-3 (Inoue et al., 2004;
Jochum et al., 2019) in the form of nano-particle pellets were analysed (Garbe-Schönberg and Müller, 2014).
MACS-3 was used for calibrating the mercury content in the samples as Hg is not present in the NIST SRM
glasses. All results for the reference materials are displayed in the appendix (Table A3). Trace element-to-calcium
ratios were quantified using the following isotopes: $^{26}$Mg, $^{27}$Al, $^{52}$Cr, $^{55}$Mn, $^{60}$Ni, $^{63}$Cu, $^{65}$Cu, $^{68}$Zn, $^{107}$Ag, $^{111}$Cd,
$^{114}$Cd, $^{118}$Sn, $^{201}$Hg, $^{202}$Hg and $^{208}$Pb normalised to $^{43}$Ca. If more than one isotope was measured for an element, the
average concentration of these was used after data processing. Uncertainty (in % RSD) was better than 5 % for all
TE/Ca ratios. The lowest RSD % based on the NIST SRM 612 glass was 2.1 % for Mn/Ca and the highest 5.0 %
for Ag/Ca. Uncertainties of all used standards and reference materials are expressed are summarized in Table A3.
Each acquisition interval lasted for 90 seconds, started and ended with measuring 20 s of gas blank, used as the
background baseline to subtract from sample intensities during the data reduction process. Furthermore, the
background monitoring ensured that the system was flushed properly after a sample. In cases when foraminiferal
test walls were very fragile causing the test to break very quickly and, hence, the length of the sample data
acquisition interval was less than 15 seconds, these profiles were excluded from further consideration.





Transient logs of raw intensities given in counts per seconds for all isotopes measured were processed with the
software Iolite (Version 4, Paton et al., 2011) producing averages of every time-resolved laser profile. The
determination of element/Ca ratios were performed after the method of (Rosenthal et al., 1999). High values of
$^{25}$Mg, $^{27}$Al or $^{55}$Mn at the beginning of an ablation profile were related to contamination on the surface of the
foraminiferal shell or remains of organic matter (e.g., Eggins et al., 2003) and these parts of the profiles were
excluded from further data processing. The detection limit was defined by 3.3*SD of the gas blank in counts per
seconds for every element in the raw data. Only values above this limit were used for further analyses. After
processing the data with Iolite, an outlier detection of the TE/Ca ratios of the samples was performed. If trace
metal values from a spot deviated more than ±2SD from the average of the samples from the corresponding
culturing phase, values were defined as outliers and discarded. The number of rejected points is indicated in the
supplementary material (Table S1).
All statistical tests were carried out using the statistical program PAST (Hammer, 2001). As the concentration of
heavy metals in seawater was varying during individual phases in the metal system (Table A1 and Fig. B1 in the
appendix), the mean concentration was calculated by applying an individual curve fit for every phase. The curve
was either linear, exponential or a power function depending on the covariance trend the particular metal showed.
Based on these curves, water values were calculated for every day and the weighted average from all days was
used for further calculations. This ensured that high concentrations in the beginning of each phase did not influence
the mean value disproportionately. The partition coefficients of the different trace metal-to-calcium ratios were
calculated using the trace element (TE) and calcium ratios in calcite and seawater. The following equation was
used:
$$D_{TE} = (TE/Ca)_{calcite}/(TE/Ca)_{seawater}.$$
When the correlation between the metal concentration in seawater and the metal concentration in the foraminiferal
test was positive and significant ($R^2 > 0.4$, $p < 0.05$), the $D_{TE}$´s are derived from the mean values of all phases and
represent the slope of the calculated regression line. In cases where a significant positive correlation between
phases could not be identified, the $D_{TE}$ values were calculated from the means of each phase separately and the
ranges given.

**3 Results**
**3.1 Survival Rates/ Growth rates / Reproductions**
**Table 2**: Number of inserted and recovered foraminifera from the different systems (C = control system, M =
metal system) and phases (0–3). Numbers of living individuals after the experiment and individuals that formed
chambers during their individual culturing phase are given in %. Note that the percentage of living foraminifera is
based on the number of foraminifera that could be recovered alive and not on the number of inserted individuals.
The number of laser spots is indicated as well.

|  | C0 | C1 | C2 | C3 | M0 | M1 | M2 | M3 | Total |
|---|---|---|---|---|---|---|---|---|---|
| **No. of inserted individuals** | | | | | | | | | |
| *Ammonia aomoriensis* | 50 | 24 | 20 | 20 | 19 | 70 | 70 | 72 | 345 |
| *Ammonia batava* | 22 | 20 | 20 | 20 | 16 | 43 | 72 | 72 | 285 |





| | | | | | | | | | |
|---|---|---|---|---|---|---|---|---|---|
| *Elphidium excavatum* | 45 | 24 | 20 | 20 | 19 | 70 | 69 | 70 | 337 |
| Total | 117 | 68 | 60 | 60 | 54 | 183 | 211 | 214 | 967 |
| **No. of recovered individuals** | | | | | | | | | |
| *Ammonia aomoriensis* | 43 | 20 | 10 | 19 | 11 | 57 | 58 | 56 | 274 |
| *Ammonia batava* | 11 | 15 | 16 | 14 | 7 | 29 | 65 | 56 | 213 |
| *Elphidium excavatum* | 36 | 20 | 20 | 14 | 7 | 62 | 58 | 53 | 270 |
| Total | 90 | 55 | 46 | 47 | 25 | 148 | 181 | 165 | 757 |
| **Living individuals (end of experiment) in %** | | | | | | | | | |
| *Ammonia aomoriensis* | 86 | 100 | 80 | 100 | 90.9 | 100 | 81 | 98.2 | 92.0 |
| *Ammonia batava* | 81.8 | 100 | 100 | 92.9 | 100 | 100 | 100 | 100 | 96.8 |
| *Elphidium excavatum* | 91.7 | 100 | 95 | 92.9 | 100 | 88.7 | 91.4 | 94.3 | 94.3 |
| Total | 86.5 | 100 | 91.7 | 95.3 | 97.0 | 96.2 | 90.8 | 97.5 | 94.4 |
| **Ind. that formed chambers (end of the experiment) in %** | | | | | | | | | |
| *Ammonia aomoriensis* | 62.8 | 84.2 | 100 | 93.8 | 81.8 | 100 | 92.3 | 90 | 88.1 |
| *Ammonia batava* | 45.5 | 85.7 | 100 | 100 | 71.4 | 100 | 100 | 100 | 87.8 |
| *Elphidium excavatum* | 69.4 | 65 | 56.3 | 38.5 | 57.1 | 67.7 | 75 | 62.3 | 61.4 |
| *Total* | 59.2 | 78.3 | 85.4 | 77.4 | 70.1 | 89.2 | 89.1 | 84.1 | 79.1 |
| **No. of laser spots** | | | | | | | | | |
| *Ammonia aomoriensis* | 22 | 18 | 17 | 20 | 9 | 39 | 40 | 36 | 201 |
| *Ammonia batava* | 14 | 20 | 19 | 19 | 6 | 17 | 52 | 57 | 204 |
| *Elphidium excavatum* | 14 | 13 | 13 | 12 | 1 | 36 | 24 | 31 | 144 |
| Total | 50 | 51 | 49 | 51 | 16 | 92 | 116 | 124 | 549 |


On average 74.5 % of the specimens inserted into the experiment could be recovered after their individual culturing
phase of 21 days and 94.4 % of these recovered specimens survived. Approximately 79.1 % of the surviving
specimens also formed at least one new chamber. Fewer specimens of *E. excavatum* formed new chambers (61.4
%) than *A. batava* (87.8%) or *A. aomoriensis* (88.1 %) (Table 2). On average, *E. excavatum* formed only one or
rarely two new chambers, whereas both *Ammonia* species formed usually more than four new chambers.
Reproduction happened very sporadically concerning between 2 and 6 specimens per phase, on average 5 %, for
the two *Ammonia* species but not for *E. excavatum*. No malformed chambers were observed in specimens that
were recovered from the trace-metal contaminated system.

**3.2 Culturing media**
**Table 3**: Weighted mean TE/Ca values in the culturing medium of the control and the metal system ± the standard
error of the mean (standard deviation $\sigma/\sqrt{n}$). Furthermore, the factors between the target concentrations (Table 1)
and the measured concentrations as well as the factors between individual phases are given. Values given without
a standard error originate from only one measurement. All values the calculations are based on can be found in the
appendix (Table A1). BDL = below detection limit.

| Cr/Ca | Mn/Ca | Ni/Ca | Cu/Ca | Zn/Ca | Ag/Ca | Cd/Ca | Sn/Ca | Hg/Ca | Pb/Ca |
|---|---|---|---|---|---|---|---|---|---|





| Control System | µmol mol⁻¹ | mmol mol⁻¹ | µmol mol⁻¹ | µmol mol⁻¹ | µmol mol⁻¹ | µmol mol⁻¹ | µmol mol⁻¹ | µmol mol⁻¹ | nmol mol⁻¹ | µmol mol⁻¹ |
|---|---|---|---|---|---|---|---|---|---|---|
| Phase 0 | BDL | 0.94 ± 0.02 | 7.0 ± 0.1 | 9.3 ± 4.3 | 118.3 ± 4.5 | 0.43 ± 0.214 | 0.41 ± 0.001 | 2.2 ± 0.4 | 5.8 ± 0.6 | 0.44 ± 0.06 |
| Phase 1 | BDL | 0.92 ± 0.00 | 6.3 ± 0.1 | 4.4 ± 1.4 | 91.6 ± 1.1 | 0.19 ± 0.013 | 0.41 ± 0.002 | 2.5 ± 0.1 | 4.5 ± 1.0 | 0.39 ± 0.02 |
| Phase 2 | 1.3 ± 0.3 | 0.90 ± 0.02 | 5.7 ± 0.1 | 2.1 ± 0.2 | 74.8 ± 2.0 | 0.19 ± 0.003 | 0.38 ± 0.006 | 2.1 ± 0.1 | 13.2 ± 5.8 | 0.31 ± 0.02 |
| Phase 3 | 2.0 ± 0.4 | 0.89 ± 0.01 | 6.8 ± 0.3 | 1.5 ± 0.1 | 78.3 ± 0.8 | 0.16 ± 0.009 | 0.37 ± 0.006 | 1.8 ± 0.1 | 5.8 ± 1.8 | 0.28 ± 0.01 |
| **Metal System** | µmol mol⁻¹ | mmol mol⁻¹ | µmol mol⁻¹ | µmol mol⁻¹ | µmol mol⁻¹ | µmol mol⁻¹ | µmol mol⁻¹ | µmol mol⁻¹ | nmol mol⁻¹ | µmol mol⁻¹ |
| Phase 0 | 8.0 ± 1.8 | 0.84 ± 0.01 | 7.4 ± 0.1 | 12.9 ± 4.5 | 104.8 ± 1.4 | 0.09 ± 0.02 | 0.43 ± 0.002 | 3.0 ± 0.1 | 5.28 | 0.50 ± 0.04 |
| Phase 1 | 8.6 ± 0.5 | 0.83 ± 0.004 | 7.3 ± 0.1 | 2.8 ± 0.3 | 95.2 ± 0.3 | 0.10 ± 0.02 | 1.12 ± 0.01 | 4.1 ± 0.1 | 39.7 ± 2.7 | 0.69 ± 0.03 |
| Phase 2 | 14.7 ± 0.1 | 0.81 ± 0.003 | 9.6 ± 0.1 | 2.4 ± 0.2 | 134.8 ± 0.5 | 0.40 ± 0.14 | 4.86 ± 0.03 | 5.2 ± 0.03 | 337.6 ± 52.1 | 2.63 ± 0.3 |
| Phase 3 | 36.3 ± 1.9 | 1.41 ± 0.004 | 61.3 ± 1.8 | 4.0 ± 1.0 | 547.5 ± 20.5 | 6.1 ± 2.5 | 78.92 ± 1.9 | 7.5 ± 1.0 | 3132.4 ± 323.7 | 57.84 ± 6.4 |
| **Factor between target conc. and measured conc.** | | | | | | | | | | |
| Phase 1 | 17.2 | 20.8 | 73.0 | 56.0 | 119.0 | 1.0 | 14.0 | 41.0 | 4.0 | 6.9 |
| Phase 2 | 2.9 | 2.0 | 9.6 | 4.8 | 16.9 | 0.4 | 6.1 | 5.2 | 3.4 | 2.6 |
| Phase 3 | 0.7 | 0.4 | 6.1 | 0.8 | 6.8 | 0.6 | 9.9 | 0.8 | 3.1 | 5.8 |
| **Factor between Phases** | | | | | | | | | | |
| Phase 0-1 | 1.1 | 1.0 | 1.0 | 0.2 | 0.9 | 1.1 | 2.6 | 1.4 | 7.5 | 1.4 |
| Phase 1-2 | 1.7 | 1.0 | 1.3 | 0.9 | 1.4 | 4.0 | 4.3 | 1.3 | 8.5 | 3.8 |
| Phase 2-3 | 2.5 | 1.7 | 6.4 | 1.7 | 4.1 | 15.3 | 16.2 | 1.4 | 9.3 | 22.0 |


**Figure 3**: Weighted mean TE/Ca values in the culturing medium in µmol mol$^{-1}$. Error bars display the standard error of the mean (standard deviation σ/√n). Open symbols represent the control system, where no extra metals were added during the complete culturing period (phase 0 to 3) and closed symbols represent the metal system. In this system, phase 0 is the control phase without any extra added metals and for phase 1 to 3, the heavy metal concentration in the culturing medium was elevated. Note that the standard error is comparably high in phase 3 because the heavy metal concentration in this phase varied more strongly, which is shown in the appendix (Table





A1, Fig. B1). Therefore, this error is derived from the real values in the seawater and not from analytical
uncertainties. Note that the Cr/Ca values from the control system in phase 0 and 1 are not given as these values
were below the detection limit.

In phases 1 and 0 the concentration in both systems were nearly equal for most elements. Only Cr and Sn had
slightly elevated concentrations in the metal system, whereas Cu and Mn concentration were higher in the control
system in phase 0 (Fig. 3). This also holds true for Mn in phase 2, when all other metals showed higher
concentrations in the metal system than in the control system. In phase 3, the concentration of all heavy metals
were elevated in the metal system as compared to the control system. The variation of the metal concentration was
in both systems highest in phase 3 for all elements but Cu, which showed the highest variation in phase 0 (Fig. 3).
The control system generally displayed a smaller degree of variation than the metal system.
The target concentration of the metals was not accomplished for most metals in phase 1 and 2, the only exception
is Ag in phase 1 (Table 3). The factors between the target and measured concentration was highest (> 50) for Ni,
Cu and Zn in phase 1 and gets smaller in phase 2 and 3. Generally, all elements but Mn were concentrated higher
in phases 1 and 2 than expected. In phase 3 Cr, Mn, Cu, Ag and Sn reached concentrations closer (factor 0.4 – 0.8)
to the target concentration and Ni, Zn, Cd, Hg and Pb were concentrated higher (factor 3.1 – 9.9) than expected.
Furthermore, the factor between individual phases (Table 3) was small for the transition from phase 0 to 1 (factor
< 1.4) for all elements but Cd (factor 2.6) and Hg (factor 7.5). Same patterns can be seen between phase 1 and 2,
while the difference between phase 2 and 3 was more distinct (factor > 4) for Ni, Zn, Ag, Cd, Pb and Hg. Mn, Cu
and Sn showed little variation between phase 2 and 3 (factor < 1.7). Generally, the factor between each phase
should have been approximately 10, which was not achieved in most cases. Exceptions were Ag, Cd and Pb, which
had factors >15 between phase 2 and 3. Furthermore, Hg showed concentrations that were higher by a factor
around 10 between all phases (phase 0-1 = 7.5, phase 1-2 = 8.5, phase 2-3 = 9.3).

### 3.3.1 Incorporation of trace metals into the foraminiferal shell

**Table 4**: Mean trace element–to–calcium values of *A. aomoriensis*, *A. batava* and *E. excavatum* in the control and
the metal system. Errors are standard errors of the mean (standard deviation $\sigma/\sqrt{n}$). Values marked with an asterisk
were derived from only one laser spot and thus are not considered for further discussion. Furthermore, the
calculated $D_{TE}$ values, the slope of the linear regression line (OLS – Ordinary Least Squares) of all means,
Pearson's correlation coefficient $R^2$ and its significance (p) are given for the calculation with all phases and when
removing phase 3 from the calculations. It's also indicated whether the regression line is forced through the origin
or not. In cases when a regression did not show significant correlation, the $D_{TE}$ range separately calculated from
the individual phases is given. In cases when the regression was significant, the $D_{TE}$ values represent the slope of
the regression line. Ph = Phase. Values in Table S1 are the basis of all calculations.

|  | Phase | Cr/Ca | Mn/Ca | Ni/Ca | Cu/Ca | Zn/Ca |
|---|---|---|---|---|---|---|
| **Control System** |  | µmol mol⁻¹ | mmol mol⁻¹ | µmol mol⁻¹ | µmol mol⁻¹ | µmol mol⁻¹ |
| *A. aomoriensis* | 0 | 18.6 ± 2.5 | 0.11 ± 0.02 | 1.3 ± 0.2 | 5.6 ± 0.9 | 53.2 ± 8.8 |
|  | 1 | 12.6 ± 0.6 | 0.53 ± 0.12 | 5.9 ± 0.8 | 8.6 ± 1.0 | 34.2 ± 4.7 |
|  | 2 | 13.6 ± 0.5 | 0.27 ± 0.07 | 2.1 ± 0.2 | 3.6 ± 0.2 | 18.6 ± 1.9 |
|  | 3 | 10.2 ± 0.6 | 0.43 ± 0.08 | 4.3 ± 0.7 | 8.1 ± 2.0 | 29.5 ± 6.1 |

| | | | | | | |
|---|---|---|---|---|---|---|
| A. batava | 0 | 11.6 ± 0.7 | 0.04 ± 0.01 | 1.4 ± 0.2 | 7.2 ± 1.1 | 23.9 ± 4.5 |
| | 1 | 10.9 ± 0.5 | 0.03 ± 0.00 | 2.6 ± 0.3 | 5.9 ± 0.6 | 17.8 ± 1.3 |
| | 2 | 9.0 ± 0.3 | 0.03 ± 0.00 | 0.9 ± 0.1 | 5.0 ± 1.0 | 12.9 ± 1.4 |
| | 3 | 9.1 ± 0.4 | 0.03 ± 0.01 | 1.9 ± 0.2 | 6.5 ± 1.3 | 14.9 ± 2.2 |
| E. excavatum | 0 | 22.9 ± 2.9 | 0.43 ± 0.13 | 9.4 ± 2.5 | 22.3 ± 7.9 | 28.1 ± 4.5 |
| | 1 | 88.9 ± 34.1 | 2.29 ± 0.56 | 7.8 ± 1.9 | 20.3 ± 8.0 | 48.9 ± 12.1 |
| | 2 | 16.2 ± 1.7 | 1.55 ± 0.26 | 5.9 ± 1.0 | 6.7 ± 1.4 | 21.9 ± 2.9 |
| | 3 | 26.7 ± 3.3 | 1.88 ± 0.55 | 4.4 ± 0.6 | 4.7 ± 0.7 | 16.8 ± 2.0 |
| **Metal System** | | | | | | |
| A. aomoriensis | 0 | 16.0 ± 0.5 | 0.08 ± 0.02 | 5.5 ± 0.9 | 15.2 ± 2.6 | 29.8 ± 5.1 |
| | 1 | 14.0 ± 0.7 | 0.39 ± 0.08 | 3.1 ± 0.3 | 6.7 ± 0.7 | 30.0 ± 4.0 |
| | 2 | 11.1 ± 0.3 | 0.20 ± 0.05 | 5.3 ± 0.5 | 5.8 ± 0.5 | 28.3 ± 2.3 |
| | 3 | 14.1 ± 1.0 | 0.71 ± 0.12 | 3.8 ± 0.3 | 6.3 ± 1.5 | 42.2 ± 6.1 |
| A. batava | 0 | 16.5 ± 0.7 | 0.07 ± 0.01 | 1.1 ± 0.1 | 7.7 ± 1.6 | 68.0 ± 9.6 |
| | 1 | 15.2 ± 1.2 | 0.04 ± 0.01 | 1.8 ± 0.3 | 2.5 ± 0.6 | 20.7 ± 2.7 |
| | 2 | 9.7 ± 0.2 | 0.02 ± 0.00 | 1.8 ± 0.1 | 8.3 ± 1.8 | 12.9 ± 1.2 |
| | 3 | 12.2 ± 0.3 | 0.17 ± 0.04 | 2.9 ± 0.2 | 8.3 ± 1.2 | 49.8 ± 3.5 |
| E. excavatum | 0 | 17.30* | 0.29* | 4.30* | 12.20* | 26.70* |
| | 1 | 32.9 ± 3.4 | 0.70 ± 0.12 | 8.2 ± 1.1 | 12.8 ± 1.8 | 18.5 ± 0.9 |
| | 2 | 41.8 ± 5.2 | 0.77 ± 0.15 | 8.6 ± 1.1 | 11.5 ± 1.5 | 29.8 ± 3.6 |
| | 3 | 54.1 ± 8.2 | 0.88 ± 0.15 | 17.0 ± 2.2 | 22.6 ± 3.6 | 43.1 ± 3.3 |
| **Calculations with Phase 3** | | | | | | |
| *A. aomoriensis* | | | | | | |
| Slope of regression line | | | 0.38 | | 1.18 | |
| Correlation coefficient ($R^2$) | | | 0.83 | | 0.80 | |
| Significance (p) | | | 0.05 | | 0.05 | |
| $D_{TE}$ | | 0.4-10.3 | 0.38 | 0.06-0.94 | 1.18 | 0.08-0.45 |
| Forced through origin | | Single points | Yes | Single points | Yes | Single points |
| *A. batava* | | | | | | |
| Slope of regression line | | | 0.23 | | | |
| Correlation coefficient ($R^2$) | | | 0.84 | | | |
| Significance (p) | | | 0.001 | | | |
| $D_{TE}$ | | 0.4-6.8 | 0.23 | 0.05-0.41 | 0.60-4.35 | 0.09-0.65 |
| Forced through origin | | Single points | No | Single points | Single points | Single points |
| *E. excavatum* | | | | | | |
| Slope of regression line | | 2.1 | | 0.19 | | |
| Correlation coefficient ($R^2$) | | 0.82 | | 0.79 | | |
| Significance (p) | | 0.01 | | 0.003 | | |
| $D_{TE}$ | | 2.1 | 0.34-2.50 | 0.19 | 0.95-5.67 | 0.08-0.53 |
| Forced through origin | | Yes | Single points | Yes | Single points | Single points |
| **Calculations without Phase 3** | | | | | | |
| *A. aomoriensis* | | | | | | |
| Slope of regression line | | | | | | |
| Correlation coefficient ($R^2$) | | | | | | |
| Significance (p) | | | | | | |
| $D_{TE}$ | | 0.74-10.3 | 0.09-0.53 | 0.19-0.94 | 0.61-5.42 | 0.21-0.45 |
| Forced through origin | | Single points | Single points | Single points | Single points | Single points |
| *A. batava* | | | | | | |
| Slope of regression line | | | | | | |
| Correlation coefficient ($R^2$) | | | | | | |
| Significance (p) | | | | | | |
| $D_{TE}$ | | 0.65-6.8 | 0.02-0.08 | 0.15-0.41 | 0.60-4.35 | 0.10-0.65 |
| Forced through origin | | Single points | Single points | Single points | Single points | Single points |
| *E. excavatum* | | | | | | |
| Slope of regression line | | | | | | |
| Correlation coefficient ($R^2$) | | | | | | |
| Significance (p) | | | | | | |
| $D_{TE}$ | | 2.5-13.4 | 0.34-2.50 | 0.64-1.35 | 0.95-4.73 | 0.22-0.53 |
| Forced through origin | | Single points | Single points | Single points | Single points | Single points |




**Table 4** continued.

| | Phase | Ag/Ca | Cd/Ca | Sn/Ca | Hg/Ca | Pb/Ca |
|---|---|---|---|---|---|---|
| **Control System** | | $\mu$mol mol$^{-1}$ | $\mu$mol mol$^{-1}$ | $\mu$mol mol$^{-1}$ | nmol mol$^{-1}$ | $\mu$mol mol$^{-1}$ |
| *A. aomoriensis* | 0 | 0.27 ± 0.08 | 7.6 ± 1.0 | 0.33 ± 0.07 | 1.54 ± 0.46 | 1.23 ± 0.22 |
| | 1 | 0.28 ± 0.05 | 3.8 ± 0.3 | 1.60 ± 0.30 | 3.11 ± 0.68 | 1.14 ± 0.16 |
| | 2 | 0.16 ± 0.04 | 3.6 ± 0.2 | 0.21 ± 0.03 | 1.13 ± 0.31 | 0.81 ± 0.10 |
| | 3 | 0.31 ± 0.11 | 2.9 ± 0.2 | 0.19 ± 0.03 | 8.02 ± 1.72 | 1.45 ± 0.42 |
| *A. batava* | 0 | 0.09 ± 0.03 | 4.7 ± 0.5 | 0.27 ± 0.05 | 1.3 ± 0.4 | 0.67 ± 0.10 |
| | 1 | 0.07 ± 0.01 | 2.5 ± 0.2 | 0.65 ± 0.09 | 1.2 ± 0.3 | 0.29 ± 0.03 |
| | 2 | 0.05 ± 0.00 | 2.7 ± 0.1 | 0.08 ± 0.02 | 1.5 ± 0.4 | 0.39 ± 0.03 |
| | 3 | 0.06 ± 0.01 | 1.9 ± 0.1 | 0.10 ± 0.02 | 4.4 ± 0.6 | 0.36 ± 0.05 |
| *E. excavatum* | 0 | 0.22 ± 0.09 | 3.6 ± 1.1 | 0.99 ± 0.40 | 15.0 ± 4.4 | 1.83 ± 0.59 |
| | 1 | 0.07 ± 0.01 | 20.1 ± 9.2 | 8.21 ± 2.63 | 83.0 ± 33.4 | 2.22 ± 0.54 |
| | 2 | 0.10 ± 0.03 | 1.2 ± 0.2 | 0.45 ± 0.08 | 16.9 ± 3.8 | 0.94 ± 0.10 |
| | 3 | 0.04 ± 0.01 | 2.3 ± 0.4 | 0.27 ± 0.03 | 35.8 ± 6.3 | 0.55 ± 0.11 |
| **Metal System** | | | | | | |
| *A. aomoriensis* | 0 | 0.08 ± 0.03 | 4.9 ± 0.3 | 0.62 ± 0.09 | 2.6 ± 0.6 | 1.17 ± 0.24 |
| | 1 | 0.25 ± 0.04 | 4.0 ± 0.4 | 0.84 ± 0.10 | 1.8 ± 0.2 | 0.90 ± 0.13 |
| | 2 | 0.52 ± 0.08 | 5.5 ± 0.4 | 1.70 ± 0.17 | 9.1 ± 1.7 | 3.85 ± 0.45 |
| | 3 | 3.03 ± 0.39 | 5.4 ± 0.4 | 0.55 ± 0.10 | 10.3 ± 1.3 | 22.14 ± 2.37 |
| *A. batava* | 0 | 0.06 ± 0.03 | 6.2 ± 0.2 | 0.19 ± 0.04 | 1.0 ± 0.2 | 1.27 ± 0.08 |
| | 1 | 0.04 ± 0.01 | 3.1 ± 0.3 | 0.59 ± 0.12 | 0.2 ± 0.0 | 0.42 ± 0.07 |
| | 2 | 0.18 ± 0.04 | 3.1 ± 0.2 | 0.46 ± 0.06 | 4.5 ± 1.1 | 0.52 ± 0.05 |
| | 3 | 1.05 ± 0.17 | 6.5 ± 0.3 | 0.21 ± 0.02 | 7.7 ± 1.0 | 29.82 ± 3.70 |
| *E. excavatum* | 0 | 0.40* | 5.60* | 0.18* | 6.80* | 1.59* |
| | 1 | 0.03 ± 0.01 | 3.0 ± 0.3 | 2.63 ± 0.32 | 85.7 ± 19.7 | 1.36 ± 0.15 |
| | 2 | 0.69 ± 0.18 | 3.9 ± 0.5 | 2.89 ± 0.47 | 120.4 ± 44.7 | 4.61 ± 0.86 |
| | 3 | 2.84 ± 0.64 | 4.7 ± 0.5 | 2.74 ± 0.42 | 94.9 ± 16.2 | 52.51 ± 6.17 |
| **Calculations with Phase 3** | | | | | | |
| ***A. aomoriensis*** | | | | | | |
| Slope of regression line | | 0.56 | | | | 0.39 |
| Correlation coefficient (R$^2$) | | 0.97 | | | | 0.97 |
| Significance (p) | | < 0.0001 | | | | < 0.0001 |
| D$_{TE}$ | | 0.56 | 0.07-18.49 | 0.07-0.63 | 0.003-1.39 | 0.39 |
| Forced through origin | | Yes | Single points | Single points | Single points | Yes |
| ***A. batava*** | | | | | | |
| Slope of regression line | | 0.17 | | | 0.003 | 0.52 |
| Correlation coefficient (R$^2$) | | 0.98 | | | 0.63 | 1 |
| Significance (p) | | < 0.0001 | | | 0.01 | < 0.0001 |
| D$_{TE}$ | | 0.17 | 0.08-14.42 | 0.03-0.26 | 0.003 | 0.52 |
| Forced through origin | | Yes | Single points | Single points | Yes | Yes |
| ***E. excavatum*** | | | | | | |
| Slope of regression line | | 0.47 | | | | 0.91 |
| Correlation coefficient (R$^2$) | | 0.96 | | | | 1 |
| Significance (p) | | < 0.0001 | | | | < 0.0001 |
| D$_{TE}$ | | 0.47 | 0.06-49.45 | 0.06-3.25 | 0.03-18.51 | 0.91 |
| Forced through origin | | Yes | Single points | Single points | Single points | Yes |
| **Calculations without Phase 3** | | | | | | |
| ***A. aomoriensis*** | | | | | | |
| Slope of regression line | | | | | | 1.6 |
| Correlation coefficient (R$^2$) | | | | | | 0.91 |
| Significance (p) | | | | | | < 0.001 |
| D$_{TE}$ | | 0.70-2.57 | 1.14-18.49 | 0.10-0.63 | 0.003-1.39 | 1.60 |
| Forced through origin | | Single points | Single points | Single points | Single points | Yes |
| ***A. batava*** | | | | | | |
| Slope of regression line | | 0.35 | | | | |
| Correlation coefficient (R$^2$) | | 0.91 | | | | |
| Significance (p) | | 0.03 | | | | |
| D$_{TE}$ | | 0.35 | 0.63-14.42 | 0.04-0.26 | 0.005-0.76 | 0.20-5.52 |
| Forced through origin | | Yes | Single points | Single points | Single points | Single points |



| *E. excavatum* | | | | | |
|---|---|---|---|---|---|
| Slope of regression line | | | | 0.26 | 2 |
| Correlation coefficient ($R^2$) | | | | 0.53 | 0.90 |
| Significance (p) | | | | 0.05 | 0.003 |
| $D_{TE}$ | 0.23-4.25 | 0.80-49.45 | 0.06-3.25 | 0.26 | 2.0 |
| Forced through origin | Single points | Single points | Single points | No | Yes |


Measurable incorporation into the foraminiferal calcite was found for all the trace metals analysed but the degree
of incorporation varied profoundly within and between species (Fig. 4 and Table 4). In both systems, the trace
metal concentration in *E. excavatum* was higher than in the other species (*A. aomoriensis* and *A. batava*) for Cr,
Mn, Ni, Cu, Hg and Sn. This trend is also visible but less pronounced in the Cu values of the control system.
Cr, Ni, Cu, Zn, Cd, Pb and Ag values of *A. aomoriensis* displayed the highest standard error of the mean paired
with highest concentrations in the water in the metal system. Sn, Mn and Hg did not show any clear pattern. In the
control system, all trace metal concentrations had higher standard errors of the mean when the concentration of
these metals in the culturing medium was higher. The trend is also shown in *A. batava* and *E. excavatum* for all
trace metals of the control and the metal system.
Calculations were performed with and without phase 3 (Fig. 4, Fig. B2 and Table 4) to address a possible overload
effect when it comes to higher metal concentrations in the seawater.
When phase 3 was included, a strong positive correlation ($R^2 > 0.9$, $p \leq 0.05$) of Ag and Pb concentrations in the
foraminiferal shell and in the culturing medium was recognised in all three species. Furthermore, *A. batava* also
revealed a positive correlation for Hg ($R^2 = 0.63$, $p < 0.01$), *A. aomoriensis* for Cu ($R^2 = 0.80$, $p < 0.05$) and *E.*
*excavatum* for Cr ($R^2 = 0.82$, $p < 0.01$) and Ni ($R^2 = 0.79$, $p < 0.003$). Weaker but still significant positive
correlations were recorded for Mn ($R^2 > 0.84$, $p \leq 0.05$) for both *Ammonia* species. An indistinct correlation of the
concentration in the seawater and in the foraminiferal test was recognised for Zn in all three species, whereas Cd
and Sn showed no covariance (Fig. 4 and Table 4).
When phase 3 was excluded from the calculations, *A. aomoriensis* and *E. excavatum* showed a positive correlation
for Pb ($R^2 > 0.9$, $p \leq 0.003$), *A. batava* for Ag ($R^2 = 0.91$, $p = 0.03$) and in *E. excavatum* Hg correlated weaker
positively ($R^2 > 0.53$, $p \leq 0.05$). All other elements show no significant correlation (Fig. 4 and Table 4).



**3.3.2 Partition coefficient ($D_{TE}$)**








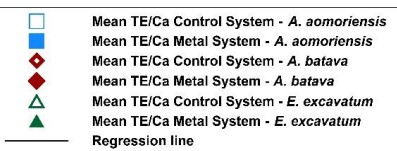


**Figure 4**: Mean TE/Ca values in the foraminiferal calcite versus the mean TE/Ca values in the corresponding


culturing medium based on phase 0 to 3. Each data point represents the mean value of all laser ablation ICP – MS


measurements on single foraminiferal chambers built up during the individual culturing phase plotted against the


mean metal concentrations in the seawater averaged over the culturing phase (Table 3). Error bars symbolize the


standard error of the mean. The linear regression line is displayed when elements showed a significant correlation


between seawater and calcite. $D_{TE}$'s of *E. excavatum* where considered without values for Phase 0 of the Metal


System as only data from one newly formed chamber are available. All values can be found in Table 4.An enlarged


graph based on the calculations without phase 3 is provided in the appendix (Fig. B2).



Partition coefficients for the different trace elements were deduced from molar foraminiferal test TE/Ca and the


values from the corresponding culturing medium. Note that the $D_{TE}$ values represent the slope of the regression


line when there was a positive correlation detected and were furthermore separately calculated for every individual


phase. The range of the calculations based on every individual phase are given when no positive correlation could


be detected. Furthermore, calculations were performed with and without phase 3 (Fig. 4 , Fig. B2 and Table 4).


Generally, the values varied between 0.003 and 49.45. The majority of $D_{TE}$ were lower than 1 in *A. aomoriensis*


(with phase 3 = 61 %, without phase 3 = 57%) and *A. batava* (with phase 3 = 75%, without phase 3 = 73%), i.e.,


uptake but no enrichment took place. $D_{TE}$ values derived from *E. excavatum* on the other hand showed a smaller


proportion < 1 (with phase 3 = 47%, without phase 3 = 42%). Note that $D_{TE}$ of *E. excavatum* were considered


without values for Phase 0 as there were data from only one newly formed chamber available. For most elements


(Cr, Mn, Ni, Cu, Cd, Sn, Pb and Hg) $D_{TE}$ derived from *E. excavatum* are higher than $D_{TE}$ from the two *Ammonia*


species (Table 4, Fig. 4), which showed comparable $D_{TE}$ values for most elements. One exception is Zn, where all


$D_{Zn}$ values are within a similar range ($D_{Zn}$ ~ 0.08-0.65) independent of the species. For *A. aomoriensis* $D_{Cu}$ was >


1 and $D_{Cd}$ as well as $D_{Pb}$ were also > 1 when phase 3 was excluded from the calculations. *A. batava* showed $D_{TE}$


values > 1 for most Cr and Cd values (excluding phase 3 and 2) when considering calculations from the individual


phases. *E. excavatum* displayed $D_{TE}$ values > 1 for Cr and Cu for the calculations with phase 3 and also for Pb


without phase 3. Furthermore, $D_{Cd}$ and $D_{Hg}$ of *E. excavatum* were also mostly > 1 when looking at calculations


based on individual phases (Table 4, Fig. 4). The highest variation between minimum and maximum $D_{TE}$ for all


species was found for Cd and Hg. In addition, some elements showed partition coefficients > 1 in a couple of


phases and/ or in one of the two culturing systems (e.g., *E. excavatum*: Mn = Control System Phase 1 to 3; *A.*


*batava*: Cu = all but Control System Phase 0 and Metal System Phase 1; Pb = Control System and Metal System


Phase 1; *A. aomoriensis*: Cr = Control System Phase 2 and 3, Metal System Phase 1; Ag = Control System Phase


1 and 3, Metal System Phase 2 and 3, Hg: Control System Phase 3, see Table 4, Fig. 4).





## 4. Discussion

### 4.1 Experimental Uncertainties

The element concentrations within the culturing medium of each culturing phase were comparably stable for most elements in the control system. In the metal system, the variations were higher, which is due to the punctual input of the stock solution for reaching the next phase concentration (Table A1, Fig. B1). This sudden adding of metals resulted in a high peak concentration in the beginning of the new phase, which equilibrated after a while. This trend is most pronounced in phase 3 as the added amount of the stock solution was highest for this phase, which is why the standard error of this phase is comparably high. Furthermore, the variations of the metal concentrations were in a comparable range than those presented in other culturing studies (e.g., Marechal-Abram et al., 2004; De Nooijer et al., 2007; Munsel et al., 2010; Remmelzwaal et al., 2019).

When taking into account that the amount of the stock solution added to the culturing medium of the metal system at the beginning of each culturing phase was elevated 1 to 1.5 order of magnitudes between phases, the measured metal concentrations are smaller than expected for phases 0, 1 and 2. This in combination with the varying metal concentration within one phase suggests that several processes are affecting the concentration in such a complex culturing system. One possible mechanism is sorption of the metals onto surfaces (e.g., tubing, culturing vessels, plates, organic matter or the foraminiferal test itself), which could have lowered the metal concentration in the culturing medium. Therefore, sorption could have contributed to the overall budget of the metals. On the other hand, Cu appears to have been released from components of the culturing system even though the system was cleaned before and was operated with seawater for 14 days before the experiments begun. For instance, the concentration of Cu was high in phase 0, where no metals were added suggesting release from system parts. In phase 1, the Cu concentration decreased meaning the contamination derived from the system was removed by a process similar to that observed for the other metals after additions were made. Similar effects have been reported by De Nooijer et al. (2007) for Cu and Havach et al. (2001) for Cd. Other processes like the uptake of the metals by the foraminifera itself and the growth of algae could further have an influence on the metal concentration in the culturing medium. Such processes are difficult to predict and even more challenging to avoid but probably mirror real environments better than sterile petri dish experiments do (e.g., Havach et al., 2001; Hintz et al., 2004; Munsel et al., 2010).

Neither the survival rate nor the formation of new chambers was influenced by the elevated metal concentrations during the culturing period. These features were rather constant between the four different phases. Furthermore, no malformations of the test morphology were recognised. Elevated heavy metal concentrations are thought to induce a higher rate of malformations in benthic foraminifera (e.g., Sharifi et al., 1991; Yanko et al., 1998), whereas recent studies constrained them as a reaction to stressful environments, not necessarily created by high heavy metal concentrations (Frontalini and Coccioni, 2008; Polovodova and Schönfeld, 2008). The lack of malformations in our experiments suggests that the foraminifera were neither poisoned by elevated trace metal concentrations nor stressed too much by strongly varying environmental parameters, maintaining a normal metabolism and growth. Reproduction was generally very rare, which may indicate that the conditions were not ideal. In field studies foraminiferal reproduction has been linked to short periods of elevated food supply (e.g., Lee et al., 1969; Gooday, 1988; Schönfeld and Numberger, 2007). The regular feeding of foraminifera in our experiment twice a week at constant rates therefore probably did not provide supply levels that trigger reproduction.



Calcein was used for staining the foraminiferal test before they were placed into the culturing system. Calcein
binds to Ca and is incorporated into the mineralised calcium carbonate (Bernhard et al., 2004). It is conceivable
that the trace metal incorporation could also be affected by calcein. However, no evidence has been found by a
variety of studies (e.g., Hintz et al., 2006; De Nooijer et al., 2007; Dissard et al., 2009). Furthermore, calcein was
only used prior to the experiment to mark the last chamber that was grown outside the culturing system. Therefore,
the incorporation of the metals measured in subsequent chambers was not affected by the calcein application.

**4.2 Incorporation of heavy metals in the foraminiferal test**
Many heavy metals have been demonstrated to be incorporated into the foraminiferal shell (e.g., Cr: Remmelzwaal
et al., 2019; Mn: Koho et al, 2015; 2017; Cu: De Nooijer et al., 2007; Ni: Munsel et al., 2010; Hg: Frontalini et
al., 2018; Cd: Havach et al., 2001; Pb: Titelboim et al., 2018; Sagar et al., 2021; Zn: Marchitto et al., 2000; Van
Dijk et al., 2017), and the incorporation of all of these metals was measured here. Additionally, to our knowledge,
Sn and Ag were investigated here for the first time. The levels observed were well above control values indicating
an elevated incorporation of Ag and Sn into the foraminiferal test calcite with increasing metal concentrations in
seawater.
Different factors can influence the incorporation of these metals into the foraminiferal test. First of all, the uptake
depends on metabolical pathways during the calcification process. Fundamental biomineralization processes of
foraminifera are the subject of an ongoing discussion and several (partly) competing models have been proposed
so far (e.g., Elderfield and Erez, 1996; Erez, 2003; De Nooijer et al., 2009b, 2014; Nehrke et al., 2013). One model
proposes that the foraminifera take up ions directly from the surrounding seawater by endocytosis or by building
seawater vacuoles, which are transported to the site of calcification (SOC) (Elderfield and Erez, 1996; Erez 2003;
De Nooijer et al., 2009b; 2009a; Khalifa et al., 2016). The SOC is located outside the foraminiferal cell and the
formation of new calcite takes place in this zone (see e.g., De Nooijer et al., 2014 for a summary and illustration).
There are evidence that this part is separated from the surrounding seawater (e.g., Spindler, 1978; Bé et al., 1979;
De Nooijer et al., 2009b; 2014; Glas et al., 2012; Nehrke et al., 2013). The other competing model suggests that
the uptake of ions and the transport to the SOC is performed directly from the seawater across the cell membrane
by active trans-membrane-transports (TMT) and/ or passive transport via gaps in the pseudopodial network of the
foraminifera (Nehrke et al., 2013; De Nooijer et al., 2014). The dependency of heavy metal concentrations in the
foraminiferal test on their seawater concentration relies on the prevailing mechanism. A biomineralization based
on endocytosis does not control the amount of ions that is introduced into the foraminiferal cell and is transported
to the SOC. Consequently, processes like Rayleigh fractionation are most important (e.g., Elderfield and Erez,
1996). This would also mean that the metal concentration in the seawater is directly mirrored by their concentration
in the foraminiferal shell, which cannot be supported by the results of our study. Indeed, several elements show
partition coefficients > 1 when the $D_{TE}$´s are calculated separately for each phase (see Results section in this
manuscript). Only Pb and Cr in *E. excavatum* and Cu and Pb in *A. aomoriensis* consistently display mean $D_{TE}$´s >
1 paired with a positive correlation of the concentration in seawater and in the foraminiferal shell, which could
indicate a non-selective uptake of these metals. On the other hand, the $D_{TE}$ values of many elements (Ni, Zn, Cd,
Hg, Pb) dramatically decrease with increasing concentration in the seawater in the highest metal treatment in all
species. This kind of overload effect was also noted by Nardelli et al. (2016) for Zn or by Munsel et al. (2010) for
Ni. Nardelli et al. (2016) suggested, that biological mechanism expulse or block these metals if the concentration





is getting too high and an imminent intoxication is probable, which may be managed by controlling the ion uptake
via TMT. Therefore, it may well be possible that the highest concentration of the metals in our study was close to
the tipping point of the biological mechanism taking over and protecting the organism.
Besides biologically controlled factors, also physicochemical properties play an important role when it comes to
the uptake of ions. One chemical factor is the aqueous speciation and solubility of the metals. Metals with a free
ion form with a charge of 2+ are more similar to $Ca^{2+}$, which makes incorporation more likely (Railsback, 1999).
Nearly all metals in this study were added as dissolved chlorides and therefore had a charge of 2+. The only
exceptions were Ag, which was added as $AgNO_3$ with a charge of 1+ and Cr, which was added as $CrCl_3*6\ H_20$.
The charge of the cation as such does not seem to make a major difference as Ag was incorporated into all three
species and Cr into *E. excavatum* with a significant positive correlation to concentrations in the culturing medium.
Furthermore, it is possible that the oxidative state of the elements is changing due to their pH dependency, which
will be discussed for every element separately. Furthermore, other ions with a charge of 1+ are also known to be
incorporated in calcite. Examples are $Li^+$ (e.g., Delaney et al., 1985; Hall et al., 2004) and $Na^+$ (e.g., Wit et al.,
2013; Bertlich et al., 2018), which are believed to occupy interstitial positions in calcite where the calcite lattice
has defects (Ishikawa and Ichikuni, 1984; Okumura and Kitano, 1986). In addition, rare earth elements with a
charge of 3+ are also detected in the foraminiferal calcite (e.g., Haley et al., 2005; Roberts et al., 2012).
The aqueous speciation of many metals is strongly influenced by the pH (e.g., Förstner, 1993; Pagnanelli et al.,
2003; Spurgeon et al., 2006; Powell et al., 2015; Huang et al., 2017). As the pH during the experiment was stable
around $8.0 \pm 0.1$ (measured twice a week), speciation changes between phases due to varying pH values can be
excluded. However, it is possible that some metals were not available in a form that could be readily incorporated
in the calcite such as the free ion or carbonate species. Cr is not available in an optimal speciation to substitute Ca
as a pH of 8 would favour $Cr^{3+}$ or $Cr^{4+}$ as well as oxides and hydroxides (Elderfield, 1970; Geisler and Schmidt,
1991). Furthermore, the used Cr-salt may not have dissolved completely, even though the stock solution was
heated and stirred during the process. Both in combination may lead to the small variation between the different
phases in the seawater. Interferences that could possible influence the Cr measurements in the water samples are
chlorine oxides or hydroxides (e.g., Tan and Horlick, 1986; McLaren et al., 1987, Reed et al., 1994; Laborda et
al., 1994). As NaCl blanks were measured, these interferences are most likely monitored and can be excluded as
a biasing factor. Furthermore, measurements of reference materials revealed accurate Cr concentrations compared
to those presented in the literature (Table A2), which also corroborates the assumption that these interferences can
be neglected. Similar pH processes could also have effected Cu as a pH around 8, like in this experiments, favours
copper carbonates over free $Cu^{2+}$ ions (e.g., Escudero et al., 2008, Millero et al., 2009), which means that the best
available speciation was not prevailing during the experiments. Nevertheless, Cu and Cr were taken up by all
species and therefore, this factor cannot be decisive when it comes to incorporation of these metals into the
foraminiferal shell.
If the incorporation of metals would be straightforward and would only depend on the speciation of the metal and
other physicochemical factors, the behaviour of the metals would mostly be influenced by the ionic radius in
combination with the charge of the metal ions as described for carbonate minerals by Rimstidt et al. (1998). The
endocytotic pathway of seawater components into the foraminifer provokes a behaviour of ion incorporation
comparable to inorganic calcite precipitation. It was found that cations are incorporated into inorganic calcite by



substitution of $Ca^{2+}$ (e.g., Reeder et al., 1999), especially when the effective ionic radius of these ions is comparable
to the one of calcium (= 1.0 Å).

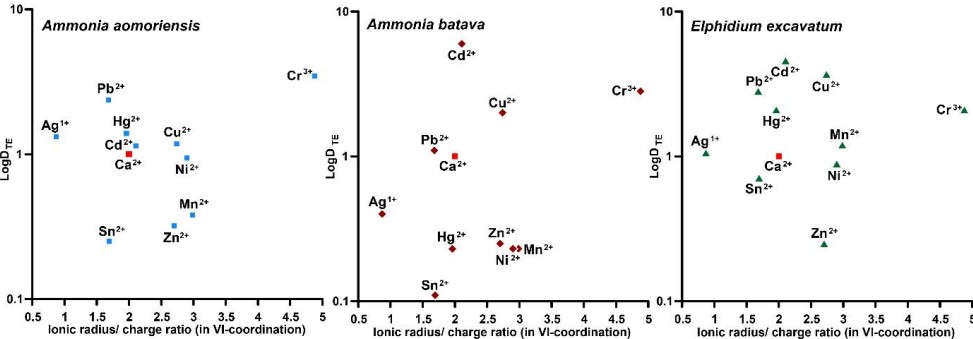


**Figure 5**: Partition coefficients (log $D_{TE}$) in dependency of the ionic radius-to-charge ratio (values from Shannon
and Prewitt, 1969) in 6-fold coordination for the trace elements analysed in this study. Displayed mean $D_{TE}$´s are
derived from single values calculated for each culturing phase individually (Metal and Control System Phase 0 to
3) and not from the regression lines. Calcium is marked in red. Blue squares represent values from *A. aomoriensis*
(left), red diamonds represent values from *A. batava* (middle) and green triangles represent values from *E.*
*excavatum* (right).

Some metals like Mn, Zn and Cu are known to be fundamentally necessary as micro-nutrients to maintain
biological and physiological function of a cell (e.g., Mertz, 1981; Tchounwou et al., 2012; Martinez-Colon et al.,
2009; Maret, 2016). Therefore, these elements should preferentially be taken up into the foraminiferal cell, where
they were used for further processes. This in turn could lead to the consumption of these metals before they can
be incorporated into the foraminiferal tests. All of these ions have a similar ionic radius (Cu = 0.73 Å, Mn = 0.67
Å, Zn = 0.74 Å) in six-fold coordination (Rimstidt et al., 1998), which would also suggest, that their behaviour is
comparable. The ionic radii are much smaller than that of Ca, but are rather similar to Mg (0.72 Å, Rimstidt et al.,
1998) (Fig. 5).
Mn shows a positive correlation between its concentration in seawater and the foraminiferal test in the two
*Ammonia* species when the calculations include phase 3. This indicates that this element serves as a well-behaved
proxy influenced mainly by its concentration in seawater. However, *E. excavatum* do not show this positive
correlation. Furthermore, $D_{Mn}$ values of this study calculated with phase 3 (*A. aomoriensis* $D_{Mn}$ = 0.38, *A. batava*
$D_{Mn}$ = 0.23, *E. excavatum* $D_{Mn}$ = 0.34-2.50) are comparable in range of those presented in Koho et al. (2015, 2017:
$D_{Mn}$ = 0.34-2.50) and Barras et al. (2018: $D_{Mn}$ = 0.09-0.35), but lower than Munsel et al. (2010) reported ($D_{Mn}$ >
2.4) except for *E. excavatum*. Species-specific differences in partition coefficients not only for Mn but also for
other elements like Mg or Na are common phenomena (e.g., Toyofuku et al., 2011; Barras et al., 2018; Wit et al.,
2013) and could also explain the offset of the $D_{TE}$ values from *E. excavatum* in this study. Furthermore, it is known
that the presence of toxic metals such as Cd, Ni or Hg can inhibit the uptake of essential metals like Mn if these
metals are present in low concentrations (e.g., Sunda and Huntsman, 1998a, 1998b). It is possible that this
mechanism is more pronounced in *E. excavatum* than in the *Ammonia* species. Zn was clearly incorporated above
control level into all three species, but it´s behaviour is influenced by more factors than the concentration of Zn in



the culturing medium. This can be inferred by the fact that there was no significant correlation recognised between
Zn concentration in calcite and seawater (Fig. 4, Table 4). $D_{Zn}$ values of this study are in good agreement with
those calculated by Van Dijk et al. (2017) for four hyaline species ($D_{Zn}$ = 0.2-0.36) and Nardelli et al. (2016) for
the miliolid *Pseudotriloculina rotunda* ($D_{Zn}$ = 0.2-4.0), especially when phase 3 is excluded from the calculation
(this study, without phase 3: *A. aomoriensis* $D_{Zn}$ = 0.21-0.45, *A. batava* $D_{Zn}$ = 0.10-0.65, *E. excavatum* $D_{Zn}$ = 0.22-
0.53). Other studies reported higher values between 3.5 and 9, though for rotaliid taxa like *Cibicidoides*
*wullerstorfi* and *Uvigerina spp.* (Marchitto et al., 2000; Van Dijk et al., 2017). It is again possible that the mixture
of metals is inhibiting the uptake of essential metals like Zn similar to Mn. Cu shows a simple well-behaved proxy
behaviour with a significant positive correlation in *A. aomoriensis* but not in the other two species. The $D_{Cu}$
presented by Munsel et al. (2010) ($D_{Cu}$ = 0.08-0.25) and De Nooijer et al. (2007) ($D_{Cu}$ = 0.1-0.4) is lower than $D_{Cu}$
from this study (*A. aomoriensis* $D_{Cu}$ = 1.18, *A. batava* $D_{Cu}$ = 0.60-4.35, *E. excavatum* $D_{Cu}$ = 0.95-5.67). Only the
lowest values of this study are in the same order of magnitude. These differences could arise from the lower
concentration of Cu in this study or from the mixture of metals. It is also reported, that the exposure to more than
one metal can cause an increased uptake of another metal into the cell (Archibald and Duong, 1984; Martinez-
Finley et al., 2012; Bruins et al., 2000; Shafiq et al., 1991). If more Cu is taken up into the cell, well may be that
after the usage of Cu as micro-nutrient, more Cu is left over and is actively deposited into the calcite. It is therefore
conceivable that one particular metal in our study was effecting a co-uptake of Cu, which lead to an elevated
incorporation into the calcite as compared to other studies.
The non – essential elements Hg, Cd and Pb are not used in physiological processes and are therefore believed to
have a higher toxic potential (Barbier et al., 2005; Raikwar et al, 2008; Ali and Khan, 2019). This in turn lead to
the assumption, that these metals are incorporated into the foraminiferal cell to a smaller amount. This could also
result in an enhanced removal of these metals, because they are potentially harmful for marine life and could
trigger mechanisms that prevent the foraminifera from intoxication as reported for various organisms (benthic
foraminifera: Bresler and Yanko, 1995; Yeast: Adle et al., 2007; Bacteria: Shaw and Dussan, 2015; Microalgae:
Duque et al., 2019). Furthermore, it could increase the input of these metals into the foraminiferal calcite as a
further removing mechanism. The ionic radii of $Pb^{2+}$ in calcite-coordination is 1.19 Å, which is remarkably higher
than those of $Hg^{2+}$ (1.02 Å) and $Cd^{2+}$ (0.95 Å), which are comparable to Ca (Fig. 5). This similarity should also
favour the incorporation of Cd and Hg into calcite, which holds only partly true, as Cd shows no trends, but Hg
correlates in *A. batava* and in *E. excavatum* if phase 3 is not integrated into the calculations. This indicated that
the incorporation of Cd is not straight-forward and is indeed depending on more complex factors than seawater
concentration of Cd. Nevertheless, Cd is incorporated well above control level in all three species. Because the
ionic radius of Pb is bigger than that of Ca a smaller degree of $Ca^{2+}$ substitution following the ionic radius to charge
ratio theory after Rimstidt et al. (1998) is expected. This is not the case as Pb emerged as a well-behaved proxy.
All three species incorporated Pb with a significant positive trend indicating that the main controlling factor is the
seawater concentration of Pb (Fig. 4, Table 4). Calculated $D_{Pb}$ values including phase 3 of this study are around 1
for *E. excavatum* ($D_{Pb}$ = 0.91), but lower for *A. aomorienis* ($D_{Pb}$ = 0.39) and *A. batava* ($D_{Pb}$ = 0.52). When excluding
phase 3, $D_{Pb}$ values of *E. excavatum* ($D_{Pb}$ = 2.0) and *A. aomoriensis* ($D_{Pb}$ = 1.6) are getting even higher, which is
may connected to the overload effect. Comparing the $D_{Pb}$ around 8.4 determined by Sagar et al. (2021), who
cultured the large benthic foraminifer *Amphisorus hemprichii*, a symbiont-bearing, miliolid species, with varying
Pb concentrations (0.5-80 µm l$^{-1}$) that are comparable to our concentration range (~ 0.11-30 µm l$^{-1}$), the $D_{Pb}$ of this
study are a little lower. Nevertheless, $D_{Pb}$ values from the present study are partly higher in individual phases (*A.*

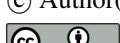



*aomorienis* $D_{Pb}$: individual phases without phase 3 = 1.3-5.14; *A. batava* $D_{Pb}$: Control System Phase 3 = 2.52; *E.*
*excavatum* $D_{Pb}$: Control System Phase 0-2 and Metal System Phase 1 $D_{Pb} \geq 3$) and thereby match the values from
Sagar et al. (2021) better. For Hg, no partition coefficients are published so far. $D_{Cd}$ values from different studies
ranged between 0.7 and 4 (Havach et al., 2001; Tachikawa and Elderfield, 2002; Maréchal-Abram et al., 2004)
and are overall lower than $D_{Cd}$ values from the present study (*A. aomoriensis* $D_{Cd}$ = 0.07-18.49, *A. batava* $D_{Cd}$ =
0.08-14.42, *E. excavatum* $D_{Cd}$ = 0.06-49.45), but mean $D_{Cd}$ values are of the same order of magnitude. Cd is known
to mimic other metals that are essential to diverse biological functions (Martelli et al., 2006; Urani et al., 2015;
Losada Ros et al., 2020). Consequently, one may argue that cadmium follows a Trojan horse strategy to get
assimilated (Martelli et al., 2006), which could also be the case for Hg. Smith et al. (2020) found a strong
correlation between the Cd concentration in the seawater and the foraminiferal shell in the species *Haynesina*
*germanica*, *Ammonia tepida*, *Quinqueloculina sabulosa* and *Triloculina oblonga*. This trend was not present in
our study, as no species showed a covariance (Fig. 4, Table 4) indicating the influence of more complex
mechanisms than simple incorporation of the seawater ion concentration. Investigations on deep water benthic
foraminifera like *Cibicidoides wuellerstorfi*, *Uvigerina peregrine* and *Melonis barleeanum* revealed a smaller
range in the $D_{Cd}$ values (e.g., McCorkle et al., 1995; Tachikawa and Elderfield, 2002), which most likely originated
from the smaller variability of the Cd concentration in their seawater as compared to this study.
The importance of other metals like Sn, Cr, Ag and Ni is not fully understood yet but some of them are believed
to have certain biological functions in the cells of animals or plants (Horovitz, 1988; Mertz, 1993; Lukaski, 1999;
Pilon-Smits et al., 2009; Hänsch & Mendel, 2009; Chen et al., 2009). For example, Ni is important for plants and
bacteria (Poonkothai and Vijayavathi, 2012; Maret, 2016). The ionic radii of these metals in calcite-coordination
is rather different (Sn = 1.18 Å; Ag = 1.15 Å; Cr = 0.62 Å; Ni = 0.69 Å) and deviate from the ionic radius of
calcite, too (Fig. 5).
Ni was incorporated with a positive trend in *E. excavatum,* but with no clear trend in the *Ammonia* species (Fig. 4,
Table 4). Munsel et al. (2010) found a similar trend in *A. tepida* and their calculated $D_{Ni}$ values ($D_{Ni}$ = 0.4-2.0) are
in good agreement with our results when the highest metal phase of this study is not taken into account (*A.*
*aomoriensis* $D_{Ni}$ = 0.19-0.94, *A. batava* $D_{Ni}$ = 0.15-0.41, *E. excavatum* $D_{Ni}$ = 0.64-1.35). Ag exhibited a strong
positive correlation between seawater and foraminiferal shell in all three foraminiferal species. Partition
coefficients for Ag (*A. aomoriensis* $D_{Ag}$ = 0.56, *A. batava* $D_{Ag}$ = 0.17, *E. excavatum* $D_{Ag}$ = 0.47) cannot be
compared to other studies as no literature is available, but the general trend, the ionic radius and the $D_{TE}$ values
are comparable to other elements in this study e.g., to Pb. Ag and Ni both display a well-behaved proxy for the
estimation of seawater concentration of these ions (Fig. 4, Table 4).
Cr and Sn, on the other hand, were not incorporated in a higher amount when the concentration of these metals in
the culturing medium was raised, except for Cr in *E. excavatum*, which showed a positive correlation. The $D_{Cr}$
values presented in Remmelzwaal et al. (2019) ($D_{Cr} > 107$), based on the tropical, symbiont bearing foraminifera
*Amphistegina spp.,* are at least one order of magnitude higher than $D_{Cr}$ values in this study (*A. aomoriensis* $D_{Cr}$ =
0.74-10.3, *A. batava* $D_{Cr}$ = 0.4-6.8, *E. excavatum* $D_{Cr}$ = 2.1). For Sn, no comparative studies are available. One
possible reason for dynamics of Cr are the comparable low concentrations in the culturing medium and
furthermore, the differences between the phases are also very low (Fig. 3, Fig. B1 and Table 3). It may be that the
concentration of Cr needs to be further elevated and the concentration range needs to be extended before the
foraminifera are able to incorporate Cr with significant differences between concentrations. We may speculate that





the same could apply for Sn. Besides, Remmelzwaal et al. (2019) suggested, that Cr in foraminiferal shells is
mainly a result of post-depositional overprinting. Diagenetic processes are very unlikely to play a role in our
experiments, which would explain, why we do not recognise a correlation between the concentration of Cr in the
culturing medium and in foraminiferal calcite.

### 725 4.3 Interspecies variability

The three different species cultured in this study clearly incorporate the same metal in different ways, which is
most visible in the overall higher TE/Ca values of *E. excavatum* compared to species from the genus *Ammonia*
(Fig. 4, Table 4). Koho et al. (2017) suggested that these differences in the incorporation result from different
microhabitats used by different foraminiferal species. This might be true in nature. In our experiments, however,
the sediment in the cavities was only a few mm thick and no redox horizon was recognised when recovering the
foraminifera after the experiment. Therefore, all foraminifera were living in the same microhabitat.
Another possible reason for the difference between *E. excavatum* and *Ammonia* species is their nutrition strategy.
As discussed above, $D_{TE}$ values were markedly higher in symbiont-bearing species. *Ammonia* species do not
harbour endosymbionts (Jauffrais et al., 2016), whereas at least five intertidal *Elphidium* species were husbanding
diatom chloroplasts, including *E. excavatum* (e.g., Pillet et al., 2011; Cesborn et al., 2017). However, an earlier
study could not corroborate the assumption that *Elphidium* species living at greater water depth in the Baltic Sea
may contain endosymbiontic zooxantellae (Schönfeld and Numberger, 2007). In our experiment, dead
*Nannochloropsis* were fed, which is certainly not the preferred food source for *E. excavatum* (Pillet et al., 2011).
This could lead to a slower growth and *E. excavatum* built on average only 1 chamber during the individual
culturing period of 21 days while *Ammonia* species built more than four chambers. Furthermore, *E. excavatum* did
not reproduce, even though the culturing period is close to the generation time of this species (Haake, 1962). When
growth is slower it could be possible that a higher amount of a metal is incorporated into the shell, which would
lead to higher TE/Ca values in this species. Another possibility is the timing of chamber formation. As *E.
excavatum* formed on average one new chamber, it is possible that this chamber was formed during the high peak
in the metal concentration during the beginning of the culturing phases (Fig. B1, Table A1). This could in turn
lead to a higher uptake of the metals and higher $D_{TE}$ values. Both *Ammonia* species on the other hand, formed
more chambers, which makes it most likely that not only the first high concentration influences their overall $D_{TE}$
value. Unfortunately, it is not possible to constrain exactly when the specimens formed their new chambers.

### 750 4.4 Application of TE/Ca values in the foraminiferal shell

**Table 5**: Comparison of the heavy metal concentrations in different regions of the world to values used for the
culturing experiments in this study. It is indicated whether the values of this study are comparable to environmental
values or if values from this study are higher or lower. (EPA = Environmental Protection Agency, USA)

| Element | Study area | Concentration in µg l⁻¹ | Comparable? | Reference |
|---|---|---|---|---|
| **Ag** | | 0.06-4.61 | | **This study** |
| | EPA Recommended Values (acute) | 1.9 | yes | Prothro, 1993 |



|  |  |  |  |  |
|---|---|---|---|---|
|  | Restronguet Creek, U.K. + Adriatic Sea | 0.0025-0.03 | yes | Barriada et al., 2007 |
|  | Ibaraki coast + Watarase river | 0.014-0.03 | yes | Shijo et al., 1989 |
|  |  | 0.14-30.61 |  | **This study** |
|  | EPA Recommended Values (chronic) | 7.9 | yes | Prothro, 1993 |
|  | Suva, Fiji | 150-250 | no, low | Arikibe and Prasad, 2020 |
|  | Black Sea in Rize, Turkey | 1-3 | yes | Baltas et al., 2017 |
|  | Gulf of Chabahar, Oman Sea | 0.15-0.19 | yes | Bazzi, 2014 |
|  | Gulf of Kutch, Arabian Sea | 200-1580 | no, low | Chakraborty et al., 2014 |
| **Cd** | East London + Port Elizabeth harbours, U.K. | 200-72600 | no, low | Fatoki and Mathabatha, 2001 |
|  | Yalujiang Estuary, China | 0.83-1.33 | yes | Li et al., 2017 |
|  | Gulf San Jorge, Argentina | 0.01-0.09 | yes | Muse et al., 1999 |
|  | Alang–Sosiya ship scrapping yard, Gulf of Cambay, India | 34-560 | yes | Reddy et al., 2005 |
|  | Kamal estuary, Jakarta | 0.01-0.02 | no, high | Putri et al., 2012 |
|  | Jakarta Bay | 0.04-0.104 | yes | Williams et al., 2000 |
|  | Kepez harbor of Canakkale, Turkey | 19-73800 | yes | Yılmaz and Sadikoglu, 2011 |
|  |  | 0.1-14.0 |  | **This study** |
|  | EPA Recommended Values (chronic) | 50 | no, low | Prothro, 1993 |
|  | Gulf of Chabahar, Oman Sea | 20.16-21.46 | yes | Bazzi, 2014 |
|  | Gulf of Kutch, Arabian Sea | 260-3010 | no, low | Chakraborty et al., 2014 |
| **Cr** | Yalujiang Estuary, China | 0.113-0.14 | yes | Li et al., 2017 |
|  | Gulf San Jorge, Argentina | 0.04-0.5 | yes | Muse et al., 1998 |
|  | Jakarta Bay | 0.511-5.25 | yes | Williams et al., 2000 |
|  | Alang–Sosiya ship scrapping yard, Gulf of Cambay, India | 35-765 | no, low | Reddy et al., 2005 |
|  |  | 0.6-6.2 |  | **This study** |
|  | EPA Recommended Values (chronic) | 3.1 | yes | Prothro, 1993 |
|  | Suva, Fiji | 880-10290 | no, low | Arikibe and Prasad, 2020 |
|  | Black Sea in Rize, Turkey | 30-242 | no, low | Baltas et al., 2017 |
|  | Gulf of Chabahar, Oman Sea | 3.37-5.74 | yes | Bazzi, 2014 |
|  | Gulf of Kutch, Arabian Sea | 1350-1850 | no, low | Chakraborty et al., 2014 |
| **Cu** | East London + Port Elizabeth harbours, U.K. | 500-42600 | no, low | Fatoki and Mathabatha, 2001 |
|  | Yalujiang Estuary, China | 1.8-4.7 | yes | Li et al., 2017 |
|  | Gulf San Jorge, Argentina | 0.02-0.65 | yes | Muse et al., 1998 |
|  | Jakarta Bay | 0.405-4.04 | yes | Williams et al., 2000 |
|  | Alang–Sosiya ship scrapping yard, Gulf of Cambay, India | 32-3939 | yes | Reddy et al., 2005 |
|  |  | 0.00035-0.273 |  | **This study** |
|  | EPA Recommended Values (chronic) | 0.94 | yes | Prothro, 1993 |
|  | South Florida Estuaries | 0.0034-0.0074 | yes | Kannan et al., 1998 |
| **Hg** | Guadalupe River and San Francisco Bay, California | 0.0017-0.135 | yes | Thomas et al., 2002 |
|  | Vembanad, India | 0.0024-0.206 | yes | Ramasamy et al., 2017 |
|  | Kamal estuary, Jakarta | 0.1-0.2 | yes | Putri et al., 2011 |
|  | Yalujiang Estuary, China | 0.006-0.049 | yes | Li et al., 2017 |
| **Mn** |  | 320-549 |  | **This study** |





| | | | | |
|---|---|---|---|---|
| | Black Sea in Rize, Turkey | 3-14 | yes | Baltas et al., 2017 |
| | Gulf of Chabahar, Oman Sea | 15.43-24.76 | no, high | Bazzi, 2014 |
| | Gulf of Kutch, Arabian Sea | 13000-18000 | no, low | Chakraborty et al., 2014 |
| | East London + Port Elizabeth harbours, U.K. | 300-23900 | yes | Fatoki and Mathabatha, 2001 |
| | Alang–Sosiya ship scrapping yard, Gulf of Cambay, India | 31-4920 | yes | Reddy et al., 2005 |
| **Ni** | | 2.3-24.3 | | **This study** |
| | EPA Recommended Values (chronic) | 8.2 | yes | Prothro, 1993 |
| | Suva, Fiji | 230-800 | no, low | Arikibe and Prasad, 2020 |
| | Black Sea in Rize, Turkey | 0.006-0.036 | yes | Baltas et al., 2017 |
| | Gulf of Chabahar, Oman Sea | 16.42-17.14 | yes | Bazzi, 2014 |
| | Gulf of Kutch, Arabian Sea | 190-330 | no, low | Chakraborty et al., 2014 |
| | Jakarta Bay | 0.058-5.25 | yes | Williams et al., 2000 |
| | Alang–Sosiya ship scrapping yard, Gulf of Cambay, India | 32-944 | yes | Reddy et al., 2005 |
| **Pb** | | 0.11-28.35 | | **This study** |
| | EPA Recommended Values (chronic) | 5.6 | yes | Prothro, 1993 |
| | Suva, Fiji | 880-1770 | no, low | Arikibe and Prasad, 2020 |
| | Black Sea in Rize, Turkey | 6-130 | yes | Baltas et al., 2017 |
| | Gulf of Chabahar, Oman Sea | 4.24-4.25 | yes | Bazzi, 2014 |
| | Gulf of Kutch, Arabian Sea | 20-120 | yes | Chakraborty et al., 2014 |
| | East London + Port Elizabeth harbours, U.K. | 600-16300 | no, low | Fatoki and Mathabatha, 2001 |
| | Yalujiang Estuary, China | 0.4-1.8 | yes | Li et al., 2017 |
| | Gulf San Jorge, Argentina | 0.1-0.5 | yes | Muse et al., 1998 |
| | Alang–Sosiya ship scrapping yard, Gulf of Cambay, India | 30-2036 | yes | Reddy et al., 2005 |
| | Kamal estuary, Jakarta | 1.3-4 | yes | Putri et al., 2011 |
| | Jakarta Bay | 0.485-3.62 | yes | Williams et al., 2000 |
| | Kepez harbor of Canakkale, Turkey | 49-9390 | yes | Yılmaz and Sadikoglu, 2011 |
| **Sn** | | 0.86-3.95 | | **This study** |
| | estuarine seawater, Galicia Coast, Spain | 0.53-1.23 | yes | Bermejo-Barrera et al., 1999 |
| | U.S. and European rivers | 0.0001-0.1 | yes | Byrd and Andreae, 1982 |
| **Zn** | | 30.0-226.9 | | **This study** |
| | EPA Recommended Values (chronic) | 81 | yes | Prothro, 1993 |
| | Suva, Fiji | 80-1450 | yes | Arikibe and Prasad, 2020 |
| | Black Sea in Rize, Turkey | 38-178 | yes | Baltas et al., 2017 |
| | Gulf of Chabahar, Oman Sea | 18.01-22.62 | yes | Bazzi, 2014 |
| | Gulf of Kutch, Arabian Sea | 11000-31000 | no, low | Chakraborty et al., 2014 |
| | East London + Port Elizabeth harbours, U.K. | 500-27600 | yes | Fatoki and Mathabatha, 2001 |
| | Yalujiang Estuary, China | 9.2-19.6 | yes | Li et al., 2017 |
| | Gulf San Jorge, Argentina | 0.01-0.55 | no, high | Muse et al., 1998 |
| | Jakarta Bay | 2-30.1 | yes | Williams et al., 2000 |
| | Alang–Sosiya ship scrapping yard, Gulf of Cambay, India | 33-5832 | yes | Reddy et al., 2005 |






During the past years, many studies were performed to assess the pollution level of seawater. The range of heavy
metal concentrations in the culturing medium of this study are compared to the metal concentrations in polluted
environments measured over the past 40 years in different regions all over the world (Table 5). The data
demonstrate that the concentration range from our study is in good agreement with threatened environments in
San Francisco Bay, California (Thomas et al., 2002), the Black Sea, Turkey (Baltas et al., 2017), the Gulf of
Chabahar, Oman Sea (Bazzi, 2014), the Restronguet Creek, U.K., the Adriatic Sea (Ag; Barriada et al., 2007), the
Yalujiang Estuary, China (Li et al., 2017), the Gulf of San Jorge, Argentina (Muse et al., 1999), Vembanad and
the Gulf of Cambay, India (Ramasamy et al., 2017; Reddy et al., 2005), Kepez harbor of Canakkale, Turkey
(Yılmaz and Sadikoglu, 2011), Jakarta (Williams et al., 2000; Putri et al., 2012) and with polluted U.S. and
European rivers (Byrd and Andreae, 1982; Kannan et al., 1998; Thomas et al., 2002). Furthermore, the maximum
metal concentration as recommended by the EPA is the lower boundary of the concentration range from this study
(Prothro, 1993). A lower concentration than the EPA value is also covered by our study during the control phase
or in the control system. This enables us to assess metal levels at the very beginning of the harmful pollution phase
in environments, which could be used as an early-warning system for the ecological status of an area (Sagar et al.,
2021). Furthermore, it allows to assess the effectiveness of contamination reducing measures. This advantage will
be important in the future for the possibility to intervene or to apply more promising measures within an adequate
time frame. In some regions of the world, seawater heavy metal concentrations are higher than in this study.
Examples are Suva, Fiji (Arikibe and Prasad, 2020), the Gulf of Kutch, Arabian Sea (Chakraborty et al., 2014) or
the East London and Port Elizabeth harbours, U.K. (Fatoki and Mathabatha, 2001) (Table 5). These areas seem to
be extremely polluted, which would make it necessary to apply a higher metal concentration to the cultured
foraminifera if a reconstruction covering these values should be made. However, this study clearly indicates a
reduced uptake of metals of interest, when the concentration of these metals in the seawater is exceeding a certain
threshold value (here between phase 2 and 3). This will make it generally difficult to model extreme high pollution
levels. Indeed, it is possible to distinguish between a heavy and a moderate pollution level. Overall, the
concentration of heavy metals in seawater should be decreasing all over the world due to a rigorous legislation for
reduction of the heavy metal input into the environment, and due to various emission reducing measures that are
applied already. This means that the concentration range of metals covered by this study is adequate for future
research and monitoring of polluted systems.

**5 Conclusion**

The aim of this study was to assess the incorporation of heavy metals into the foraminiferal calcite as a function
of their concentration in the seawater the foraminifera calcified in. Culturing experiments with different
foraminiferal species (*A. aomoriensis*, *A. batava* and *E. excavatum*) that were exposed to a mixture of ten different
metals (Cr, Mn, Ni, Cu, Zn, Ag, Cd, Sn, Hg and Pb) at varying concentrations (Table 3, Fig. 3, Fig. B1) were
carried out to gain further insights into the uptake of heavy metals. Laser ablation ICP-MS analysis of the newly
formed calcite revealed species-specific differences in the incorporation of heavy metals. Nevertheless, all metals
used in this study were incorporated into the foraminiferal calcite of all three species (Fig. 4, Table 4). Some
elements showed a behaviour inferring that the uptake of these metals mainly depends on its concentration in
seawater, which was indicated by strong positive correlations between the metal concentration in seawater and in
the foraminiferal calcite. All three species showed a strong positive correlation between Pb and Ag in the water





and their calcite. *A. aomoriensis* further revealed a slightly weaker correlation for Mn and Cu. *A. batava* holds a
strong positive correlation for Mn and Hg. *E. excavatum* depicts a strong positive correlation for Cr and Ni. Other
elements like Cd and Zn showed a more complicated behaviour indicating that factors other than seawater
concentration are effecting the uptake of these metals, which is demonstrated by no clear correlation between
seawater and calcite metal values. The reasons for this different behaviour are yet unclear. $D_{TE}$ values of Ni, Zn,
Cd, Hg and Pb decrease with increasing metal concentration in the seawater, which is most prominent in the
highest metal treatment in all species. This could be due to an overload effect arising when the concentration of
the metals is exceeding a certain threshold and could potentially be harmful or even lethal for the organism. This
in turn could lead to a removal or a prevention from uptake of the metal (Nardelli et al., 2016). The results of this
study facilitate a reconstruction of the heavy metal concentration in seawater for those elements showing a
correlation between TE/Ca ratios in calcite and seawater. Such estimates can be based on foraminiferal samples
from sediment cores and recent surface sediments, and facilitate a monitoring of anthropogenic footprints on the
environment today and in the past. The presented $D_{TE}$´s allow a direct quantification of metal concentrations in
polluted and pristine areas. The foraminiferal species considered prevail in nearly all coastal environments
worldwide, except polar latitudes. This in combination provides a powerful tool for monitoring the ecosystem
status in various areas of interest.

**Appendix**
**Appendix A: Additional Tables**

**Table A1**: TE/Ca$_{Seawater}$ values from single weeks during the culturing period of the metal system. Measurements
were carried out with ICP-MS. This values are the basis for the calculations of the mean TE/Ca values in Table 3
and for figure B1.

| Metal System | | | Sampling date | Cr/Ca | Mn/Ca | Ni/Ca | Cu/Ca | Zn/Ca | Ag/Ca | Cd/Ca | Sn/Ca | Hg/Ca | Pb/Ca |
|---|---|---|---|---|---|---|---|---|---|---|---|---|---|
| | Phase | Day | | µmol mol$^{-1}$ | µmol mol$^{-1}$ | µmol mol$^{-1}$ | µmol mol$^{-1}$ | µmol mol$^{-1}$ | µmol mol$^{-1}$ | µmol mol$^{-1}$ | µmol mol$^{-1}$ | nmol mol$^{-1}$ | µmol mol$^{-1}$ |
| FR0 W2 | 0 | 10 | 10.2.20 | 12.80 | 818.54 | 7.60 | 27.75 | 100.19 | 0.16 | 0.44 | 3.20 | | 0.63 |
| FR0 W3 | 0 | 17 | 19.2.20 | 3.16 | 858.94 | 7.23 | 3.74 | 107.69 | 0.05 | 0.43 | 2.94 | 5.28 | 0.43 |
| FR1 W1 | 1 | 2 | 27.2.20 | 13.59 | 862.52 | 7.08 | 6.25 | 97.45 | 0.37 | 1.00 | 4.98 | 43.07 | 1.03 |
| FR1 W2 | 1 | 9 | 5.3.20 | 5.86 | 796.65 | 6.69 | 2.23 | 93.09 | 0.04 | 1.06 | 3.87 | 19.13 | 0.69 |
| FR1 W3 | 1 | 13 | 9.3.20 | 7.03 | 819.38 | 6.86 | 2.14 | 95.50 | 0.06 | 1.08 | 4.23 | 27.17 | 0.62 |
| FR1 W4 | 1 | 20 | 16.3.20 | 7.75 | 844.23 | 7.94 | 2.77 | 95.75 | 0.11 | 1.19 | 4.11 | 60.20 | 0.68 |
| FR2 W1 | 2 | 2 | 19.3.20 | 13.68 | 825.59 | 10.02 | 4.15 | 129.09 | 1.88 | 5.20 | 5.37 | 933.50 | 5.70 |
| FR2 W2 | 2 | 8 | 26.3.20 | 16.49 | 820.63 | 9.75 | 2.78 | 134.85 | 0.41 | 4.96 | 5.46 | 494.26 | 3.07 |
| FR2 W3 | 2 | 15 | 2.4.20 | 13.31 | 811.64 | 9.44 | 2.23 | 132.12 | 0.31 | 4.89 | 5.10 | 287.70 | 2.50 |
| FR2 W4 | 2 | 19 | 6.4.20 | 15.47 | 789.96 | 9.77 | 2.23 | 135.50 | 0.33 | 4.75 | 5.19 | 210.66 | 2.20 |
| FR3 W1 | 3 | 2 | 9.4.20 | 52.74 | 1558.73 | 74.72 | 15.89 | 772.38 | 31.53 | 87.65 | 18.31 | 6123.75 | 125.25 |
| FR3 W2 | 3 | 7 | 14.4.20 | 39.90 | 1281.58 | 46.73 | 3.67 | 455.31 | 7.95 | 61.37 | 11.84 | | 70.27 |
| FR3 W3 | 3 | 16 | 23.4.20 | 26.97 | 1469.59 | 66.07 | 3.55 | 579.52 | 4.13 | 84.82 | 5.87 | 2858.26 | 53.51 |
| FR3 W4 | 3 | 20 | 27.4.20 | 25.59 | 1397.18 | 65.00 | 3.01 | 550.78 | 4.31 | 84.23 | 5.02 | 1640.01 | 45.72 |


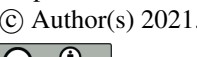



**Table A2**: Average concentration, RSD (1σ in %), literature values, accuracy in comparison to literature values
and number of measurements of the reference materials SLRS-6, SLEW-3, in-house reference materials (South
Atlantic surface water and South Atlantic Gyre water) and NASS-6 measured with ICP-MS. Average
concentration, RSD and accuracy values displayed here are averaged from single measuring days. Cr values are
analysed after dilution of the samples and all other elements were analyses after preconcentration with a SeaFAST
system. NRCC – National Research Council Canada. *Values originated from 1:10 dilution of SLRS-6.

| Reference Materials | Cr | Mn | Ni | Cu | Zn | Cd | Pb |
|---|---|---|---|---|---|---|---|
| **SLRS-6** | nmol kg⁻¹ | nmol kg⁻¹ | nmol kg⁻¹ | nmol kg⁻¹ | nmol kg⁻¹ | nmol kg⁻¹ | nmol kg⁻¹ |
| Average conc. | 4732 | 52956 | 9811 | 338014* | 31391* | 62 | 786 |
| RSD% | 3.5 | 3.9 | 6.0 | 1.7* | 7.2* | 12.8 | 0.8 |
| Yeghicheyan et al., 2019 | 4509 | 38616 | 10496 | 376378* | 26920* | 56 | 820 |
| Accuracy | 0.96 | 0.74 | 1.08 | 1.11* | 0.86* | 0.90 | 1.04 |
| Number | 4 | 11 | 11 | 13* | 13* | 7 | 7 |
| **SLEW-3** | | | | | | | |
| Average conc. | | 40007 | 17508 | 22907 | 4442 | 343 | |
| RSD% | | 4.3 | 3.5 | 4.2 | 9.1 | 4.8 | |
| Leonhard et al., 2002 | | 29326 | 20958 | 24409 | 3074 | 427 | |
| Accuracy | | 0.74 | 1.21 | 1.07 | 0.78 | 1.28 | |
| Number | | 12 | 12 | 12 | 12 | 12 | |
| **South Atlantic Gyre water** | | | | | | | |
| Average conc. | | 1615 | 2189 | 2649 | 5614 | | |
| RSD% | | 6.2 | 3.7 | 5.3 | 13.2 | | |
| Number | | 10 | 10 | 10 | 10 | | |
| **South Atlantic surface water** | | | | | | | |
| Average conc. | | 1959 | 2417 | 2646 | 39718 | | |
| RSD% | | 6.8 | 2.8 | 5.8 | 2.2 | | |
| Number | | 6 | 6 | 6 | 6 | | |
| **NASS-6** | | | | | | | |
| Average conc. | 6747 | 11162 | 3557 | 5206 | 5158 | 169 | |
| RSD% | 15.9 | 5.2 | 3.2 | 3.0 | 25.3 | 7.0 | |
| NRCC | 2293 | 9654 | 5129 | 3528 | 3931 | 165 | |
| Accuracy | 0.34 | 0.87 | 0.76 | 0.35 | 0.81 | 0.98 | |
| Number | 9 | 11 | 11 | 11 | 11 | 2 | |


**Table A3**: Average concentration, RSD (1 σ in %), literature values, accuracy in comparison to literature values
and number of measurements of the reference materials NIST SRM 610, NIST SRM 614, JCt-1, JCp-1, MACS-3
and ECRM752-1 measured with LA-ICP-MS. Please note that for the ECRM752-1 no reported values for the
elements of interest are available, which is also the case for some elements in other reference materials. It is
important to note that the Hg/Ca values in the NIST glasses are not reliable as Hg is volatile and most likely
volatilized during the glass formation. Average concentration, RSD and accuracy values displayed here are
averaged from single measuring days.





| Reference materials | Cr/Ca | Mn/Ca | Ni/Ca | Cu/Ca | Zn/Ca | Ag/Ca | Cd/Ca | Sn/Ca | Hg/Ca | Pb/Ca |
|---|---|---|---|---|---|---|---|---|---|---|
| **NIST SRM 612** | mmol mol$^{-1}$ | mmol mol$^{-1}$ | mmol mol$^{-1}$ | mmol mol$^{-1}$ | mmol mol$^{-1}$ | mmol mol$^{-1}$ | mmol mol$^{-1}$ | mmol mol$^{-1}$ | µmol mol$^{-1}$ | mmol mol$^{-1}$ |
| Mean value | 0.33 | 0.34 | 0.33 | 0.29 | 0.37 | 0.08 | 0.12 | 0.15 | 2.26 | 0.10 |
| RSD% | 3.49 | 2.09 | 2.30 | 4.05 | 2.97 | 5.01 | 3.90 | 2.08 | 26.32 | 2.52 |
| Jochum et al., 2011 | 0.33 | 0.33 | 0.31 | 0.28 | 0.28 | 0.10 | 0.12 | 0.15 | | 0.09 |
| Number of spots | 69 | 75 | 75 | 75 | 73 | 73 | 75 | 75 | 41 | 75 |
| **NIST SRM 614** | µmol mol$^{-1}$ | µmol mol$^{-1}$ | µmol mol$^{-1}$ | µmol mol$^{-1}$ | µmol mol$^{-1}$ | µmol mol$^{-1}$ | µmol mol$^{-1}$ | µmol mol$^{-1}$ | nmol mol$^{-1}$ | µmol mol$^{-1}$ |
| Mean value | 19.28 | 10.31 | 8.43 | 15.86 | 67.58 | 2.13 | 15.53 | 5.97 | 20.93 | 5.23 |
| RSD% | 10.57 | 4.47 | 4.66 | 3.03 | 2.44 | 4.92 | 5.69 | 2.98 | 20.69 | 1.98 |
| Jochum et al., 2011 | 10.78 | 12.18 | 8.83 | 10.16 | 20.11 | 1.83 | 2.35 | 6.67 | | 5.28 |
| Accuracy | 0.57 | 1.19 | 1.06 | 0.64 | 0.30 | 0.86 | 0.23 | 1.12 | | 1.01 |
| Number of spots | 35 | 38 | 37 | 39 | 38 | 38 | 38 | 39 | 19 | 39 |
| **MACS-3** | mmol mol$^{-1}$ | mmol mol$^{-1}$ | mmol mol$^{-1}$ | mmol mol$^{-1}$ | mmol mol$^{-1}$ | mmol mol$^{-1}$ | mmol mol$^{-1}$ | mmol mol$^{-1}$ | µmol mol$^{-1}$ | mmol mol$^{-1}$ |
| Mean value | 0.21 | 0.97 | 0.093 | 0.17 | 0.13 | 0.065 | 0.041 | 0.042 | 5.11 | 0.026 |
| RSD% | 1.60 | 1.36 | 1.90 | 1.92 | 2.19 | 6.37 | 2.83 | 2.68 | 9.23 | 2.18 |
| Jochum et al., 2019 | 0.23 | 0.99 | 0.10 | 0.19 | 0.20 | 0.054 | 0.051 | 0.049 | 5.41 | 0.031 |
| Accuracy | 1.13 | 1.02 | 1.09 | 1.11 | 1.50 | 0.84 | 1.24 | 1.15 | 1.07 | 1.16 |
| Number of spots | 45 | 45 | 44 | 46 | 46 | 42 | 46 | 46 | 44 | 46 |
| **JCt-1NP** | µmol mol$^{-1}$ | µmol mol$^{-1}$ | µmol mol$^{-1}$ | µmol mol$^{-1}$ | µmol mol$^{-1}$ | µmol mol$^{-1}$ | µmol mol$^{-1}$ | µmol mol$^{-1}$ | nmol mol$^{-1}$ | µmol mol$^{-1}$ |
| Mean value | 6.16 | 0.91 | 0.37 | 1.14 | 1.46 | 0.01 | 1.60 | 2.30 | 8.93 | 0.063 |
| RSD% | 14.25 | 15.59 | 9.56 | 7.44 | 10.37 | 6.57 | 11.75 | 5.06 | 23.95 | 5.86 |
| Jochum et al., 2019 | 0.93 | 1.01 | 1.03 | 1.48 | | | | | | 0.064 |
| Accuracy | 0.15 | 1.19 | 2.71 | 1.31 | | | | | | 1.04 |
| Number of spots | 44 | 38 | 45 | 47 | 45 | 11 | 46 | 13 | 26 | 48 |
| **JCp-1NP** | µmol mol$^{-1}$ | µmol mol$^{-1}$ | µmol mol$^{-1}$ | µmol mol$^{-1}$ | µmol mol$^{-1}$ | µmol mol$^{-1}$ | µmol mol$^{-1}$ | µmol mol$^{-1}$ | nmol mol$^{-1}$ | µmol mol$^{-1}$ |
| Mean value | 9.61 | 2.11 | 0.50 | 0.84 | 1.81 | 0.02 | 0.98 | 0.06 | 8.25 | 0.13 |
| RSD% | 7.91 | 4.62 | 6.89 | 6.36 | 6.53 | 11.34 | 11.08 | 10.68 | 20.96 | 6.15 |
| Jochum et al., 2019 | 1.27 | 2.16 | 1.05 | 1.29 | 3.53 | | | | | 0.15 |
| Accuracy | 0.15 | 1.06 | 2.10 | 1.25 | 1.96 | | | | | 1.19 |
| Number of spots | 37 | 41 | 41 | 40 | 41 | 21 | 36 | 30 | 21 | 47 |
| **ECRM752-1** | µmol mol$^{-1}$ | µmol mol$^{-1}$ | µmol mol$^{-1}$ | µmol mol$^{-1}$ | µmol mol$^{-1}$ | µmol mol$^{-1}$ | µmol mol$^{-1}$ | µmol mol$^{-1}$ | nmol mol$^{-1}$ | µmol mol$^{-1}$ |
| Mean value | 14.75 | 144.44 | 3.87 | 2.34 | 8.40 | 0.01 | 1.54 | 0.04 | 19.14 | 0.86 |
| RSD% | 7.78 | 2.54 | 4.97 | 6.21 | 2.37 | 87.11 | 7.76 | 9.22 | 18.03 | 3.82 |
| Number of spots | 27 | 31 | 26 | 28 | 27 | 15 | 29 | 24 | 19 | 31 |


**Appendix B: Additional Figures**




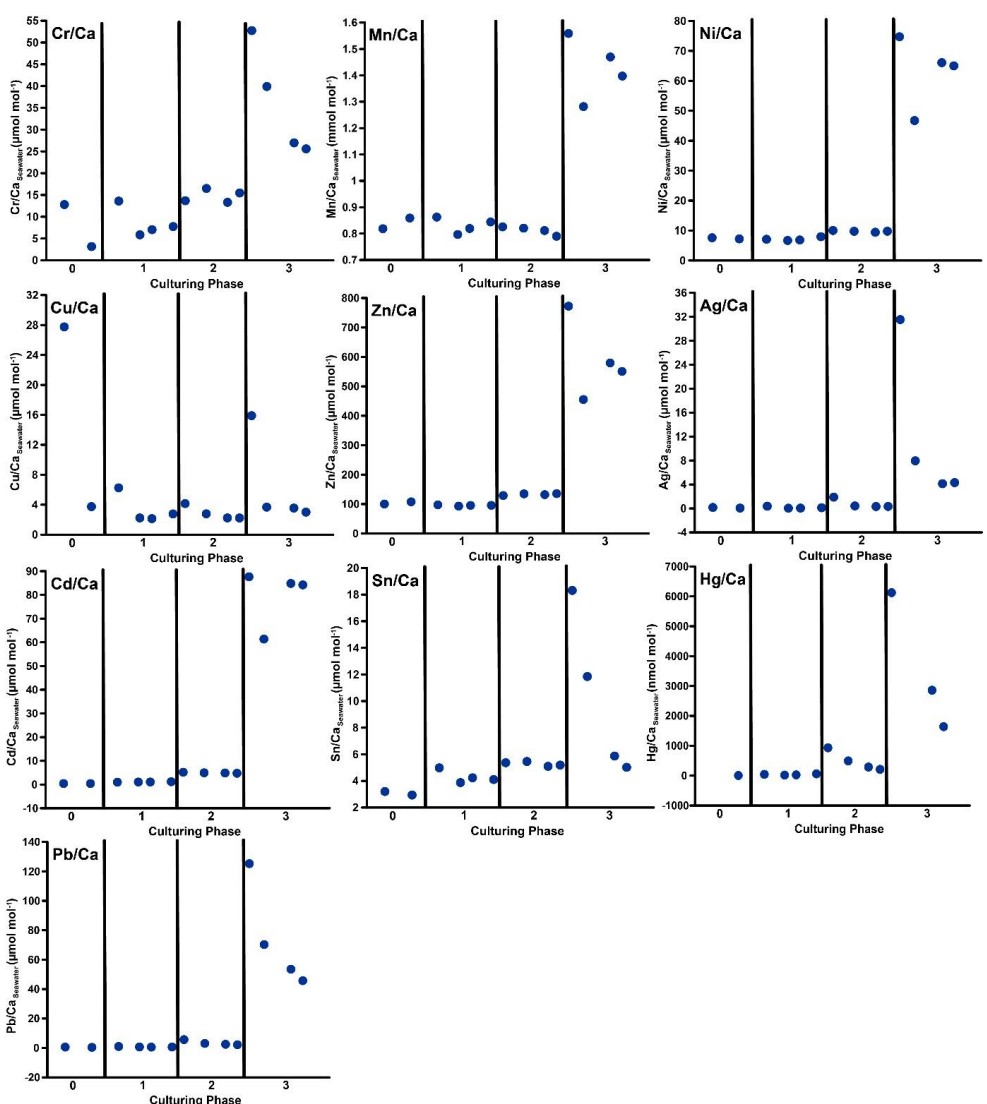

**Figure B1**: TE/Ca values in the culturing medium of the metal system in $\mu$mol mol$^{-1}$ or nmol mol$^{-1}$ divided by individual culturing phases. In this system, phase 0 is the control phase without any extra added metals and for phase 1 to 3, the heavy metal concentration in the culturing medium was elevated. The data the figure is based on can be found in Table A1.














**Figure B2**: Mean TE/Ca values in the foraminiferal calcite versus the mean TE/Ca values in the corresponding culturing medium without phase 3. Each data point represents the mean value of all laser ablation ICP – MS measurements on single foraminiferal chambers built up during the individual culturing phase plotted against the mean metal concentrations in the seawater averaged over the culturing phase (Table 3). Error bars symbolize the standard error of the mean. The linear regression line is based on the calculations excluding phase 3 and is only displayed when elements showed a significant correlation between seawater and calcite. $D_{TE}$'s of *E. excavatum* where considered without values for Phase 0 as only data from one newly formed chamber are available. All values can be found in Table 4.

**Supplements**

**Table S1–S3**: TE/Ca$_{Calcite}$ values from *Ammonia aomoriensis* (Table S1)*, Ammonia batava* (Table S2) and *Elphidium excavatum* (Table S3). Values represent single laser ablation spots on foraminiferal chambers that were formed during the individual culturing period in the control and the metal system. Only values above the detection limits of the individual element are presented. Furthermore, outliers are also excluded. These values are the basis for the calculation of the mean TE/Ca values in Table 4 and Fig. 4. The sample ID indicates the species (AA = *A. aomoriensis*, AB = *A. batava*, E = *E. excavatum*), the culturing phase, the system (R = metal system, L = control system), the individual and the chamber that was ablated, starting from the innermost chamber going to the youngest one.

**Data availability**

All data generated or analysed during this study are included in this published article and its supplementary information files.

**Author contribution**

This study was initiated by JS and EH. SS collected the samples, cultured the foraminifera, processed the samples in the laboratory and acquired, analysed and interpreted the water and foraminiferal data. JS helped with the sampling logistics, design and implementation of the culturing experiments. EH advised and helped with the processing and analysis of the water samples and EH and DGS advised and helped with the measurements of the foraminiferal samples. SS wrote the manuscript with all the authors contributing to the discussion and data interpretation, and editing of the work.

**Competing interests**

The authors declare that they have no conflict of interest.

**Acknowledgements**

We are indebted to Tal Dagan and Alexandra-Sophie Roy, Kiel University, for providing the basic compounds of the culturing setup and for helping us with setting up the systems. Claas Hiebenthal, KIMOCC, had previously helped with the system design. Furthermore, Regina Surberg carried out the ICP-OES measurements, Kathleen



Gosnell performed the Hg measurements in the water samples and Ulrike Westernströer set up and helped with
the laser ablation measurements, which was vitally important for this manuscript. The fieldwork was supported by
"Schutzstation Wattenmeer" on Hallig Hooge, in particular by a guided tour to the Japsand and by providing
laboratory facilities at their station. Leif Boyens is thanked for his flexibility and his accommodation space on
Hooge. The help of Danny Arndt during fieldwork is gratefully acknowledged.

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
