# Peer review of "Heavy metal uptake of near-shore benthic foraminifera during"

_Biogeosciences, 2021_

## Author Comment (AC1)

**Heavy metal uptake of near-shore benthic foraminifera during multi-element culturing experiments**

Authors: Schmidt, S., Hathorne, EC., Schönfeld, J., Garbe-Schönberg D.

**Response to Reviewer#1 comments**

In the following, the Reviewers' comments or questions on the manuscript are given in black italics, and our response is highlighted in blue and indented. We only consider points that provide information and clarifications that are of general interest to the readership. Minor points like " Line 258: add a comma (,) after samples" or "Break this sentence into two" will be performed during the manuscript revision, and hence are not to be discussed here.

For interactive discussions:

*1) Higher distribution coefficient values are reported for symbiont-bearing foraminifera species (Line 727). Is it possible or logical to think that lack of food can make the cultured specimens (non-symbiont foraminifera) weak, which can inhibit the incorporation of heavy metals from the surrounding culture medium?*

- This is unlikely as a sufficient amount of algal food was added during the experimental period. This was evident by leftovers covering the sediment surfaces in the cavities at the end of each phase. This would have been consumed by the foraminifera if they would have needed more. Furthermore, the foraminifera calcified, which wouldn't be the case if any undersupply occurred (e.g. Lee et al., 1991; Kurtarkar et al., 2019). Therefore, the nutritional status is unlikely to have influenced the metal uptake by the foraminifera.

*2) Does the different species of Ammonia show similar rates of incorporation (comparison of this study with published culture-studies with other species of Ammonia?*

- We can have a further look for culture studies addressing the incorporation of metals into the tests of other *Ammonia* species and add this to the discussion. Examples are Munsel et al. (2010) addressing Mn, Ni and Cu in the test of *Ammonia tepida* or Marechal-Abrams et al. (2004), who looked at the Cd incorporation into the shell of *Ammonia beccarii*. Furthermore, de Nooijer et al. (2007) cultured *Ammonia tepida* with different Cu concentrations or van Dijk et al. (2017) investigated the Zn uptake of *Ammonia tepida*. These examples are already integrated into the manuscript, but we can go further into detail and point out the species-specific differences. Furthermore, we will add a comparing table or figure.

*3) DZn obtained in this study are in good agreement with hyaline species and also a miliolid (Lines 634-636). Some previous work on several miliolid species report elevated concentrations of Zn in their shell compared to the ambient seawater. It would be interesting to discuss about the difference in metal incorporation between miliolid and rotalid species in the discussion section 4.3 Interspecies variability.*

- We agree with the reviewer. It is interesting to note that our study compared to both miliolid and rotaliid species. The differences in calcification of miliolid and rotaliid

foraminifera and its implication on the heavy metal concentration in the foraminiferal should indeed be discusses further. We will add this to a comparing figure.

**Abstract**

*From the culture experiments, Pb and Ag are seen to incorporate linearly (Figure 4) in the new calcite of all three foraminiferal species. It would value if the distribution coefficients obtained for these metals are added in the abstract.*

-   The $D_{TE}$ of these elements (apparent partition coefficients for Ag: *A. aomoriensis* = 0.56, *A. batava* = 0.17, *E. excavatum* = 0.47; for Pb: *A. aomoriensis* = 0.39, *A. batava* = 0.52, *E. excavatum* = 0.91) can be added to the abstract.

**Introduction**

*Line 65: another multi-element culture experiments on large benthic foraminifera Amphisorus hemprichii reports the proportional enrichment of Mn, Ni and Cd (Sagar et al., 2021) in the foraminiferal tests from the culture solutions, and the thresholds*

-   We thank the reviewer for pointing out this recently published study, the results of which will be considered in the revised manuscript. For instance, Sagar et al. (2021a) found that the partition coefficient of *Amphisorus hemprochii* for Mn was 1.3±0.2, which is slightly higher, but in the same order of magnitude than our $D_{Mn}$ values. The presented $D_{Ni}$ of 0.3±0.04 is comparable to our findings and the partition coefficient of Cd ($D_{Cd}$ = 2.6±0.3) reported by Sagar et al., (2021a) is also in the same range. Nevertheless, it should be noted that $D_{Cd}$ is more variable in our data. The applied concentrations of Mn in the culturing medium of our study were higher, while the concentration of Ni was lower and the concentration of Cd was comparable. We will add all this information to the table (or figure) comparing foraminiferal species and culturing experiments.

**Material and Methods**

*Line 105-106, 116-117, and 124: Were the containers pre-cleaned? If yes, mention the pre-cleaning of the EMSA CLIP and Close boxes used for storing samples from Japsand, and the plastic containers used to collect samples from Kiel Fjord*

-   Yes, all boxes were pre-cleaned with Mucasol soap water and 5% $HNO_3$ before use. We will add this information to the revises version of the manuscript.

*Line 118-119: Does this apply to all the 9 cores collected from Kiel Fjord? If yes, change the sentence and include the term 'all the 9 cores'*

-   Yes, all sediment cores displayed a very similar sediment succession. This will be specified in the revised version of the manuscript.

*Lines 127-128: Any reason why artificial seawater was used for washing and storage of the samples.*

-   Artificial seawater was used to ensure that no microorganism that could potentially have harmful effects on the foraminifera were introduced, for instance ciliates. If

natural seawater were used, it would have to be sterilized before use (e.g., by autoclavation or filtration). Considering the high volume of water that was necessary to process the samples, it was less time consuming to use artificial seawater.

*Lines 139-141: Mention the cleaning protocols/steps followed for laboratory ware, which were used to handle the foraminifera specimens*

- The following information will be added to the revised manuscript:
- Plastic utensils: First of all, a pre-cleaning with Mucasol-water mixture (soap) was performed to remove oils and other contaminants that could remain from production. Therefore, the laboratory ware was stored in MilliQ water with added Mucasol (soap) overnight in an oven with 35 °C. Secondly, the Mucasol water was rinsed well and 5% $HNO_3$ was applied for at least 2 days before rinsing.
- Paint brush and other materials made of different material than plastic were rinsed with Ethanol to avoid any biological contamination.

*Line 152: cite reference for calcein (16 mg/l)*

- This was a mistake. The Calcein concentration used in this study was 10 mg/l as described by Bernhard et al. (2006).

*Line 154: Were temperature measurements carried at the sampling locations? If yes, mention it in the section 'Field Sampling'*

- Water temperature measurements were performed at the North Sea stations.

Line 218: All salts used were provided in p.a. quality. (does p.a. means pro analysi; please write in full). Please mention the provider of the salts (e.g., SigmaAldrich or CarlRoth or ?)

- Yes, p.a. is an abbreviation of "pro analysi". The providers of the chemicals were Carl Roth ($CrCl_3 \cdot 6\ H_2O$; $SnCl_2 \cdot 2\ H_2O$ and $PbCl_2$), Walter CMP ($CdCl_2$) and Sigma Aldrich ($MnCl_2 \cdot 4\ H_2O$, $NiCl_2 \cdot 6\ H_2O$, $CuCl_2 \cdot 2\ H_2O$, $ZnCl_2$, $AgNO_3$ and $HgCl_2$).

*Line 210: Reference 'Frontallini et al., (2018a)" studies the effect of mercury pollution on cultured benthic foraminifera. Frontallini et al., (2018b) studies the ultrastructural alteration in benthic foraminifera induced by heavy metals (e.g., Pb).*

- The study of Frontalini et al. (2018b) is already included. We were not aware of the mercury study by Frontalini et al. (2018a), which will be cited in the revised manuscript.

*Lines 229-230: Each experimental phase lasted 21 days (three-weeks) and water with heavy metal concentration was fed into the system bi-weekly. Does it mean that in the three-weeks duration, of each phase, the multi-element culture experiment was fed once with the multi-element spike? Please explain clearly.*

- The multi element stock solution was added at the beginning of each phase to reach the targeted concentration. Additionally, a smaller aliquot of the same stock solution was introduced twice a week during the three weeks of a phase. The reason was that

we expected a loss of metals during the culturing phase (e.g., uptake by foraminifera or algae, adsorption to surfaces of the culturing system).

*Line 311: ICP-MS/MS (do the authors mean ICP-MS)*

- A tandem ICP-MS/MS instrument (Agilent 8900) was used for analysis.

**Results**

*Line 392-393: In contrast to the text, the figure shows higher Cu/Ca concentrations in the metal experiment of phase 0.*

- This sentence was unclearly formulated and the Cu concentration is higher in the metal systems in phase 0, which can be clarified in the revised version of the manuscript.

*Line 393: The control system in Mn/Ca shows higher concentration in all phases. Reword lines 391-394.*

- Indeed, the Mn/Ca concentration was higher in the control system than in the metal system for the phase 0, 1 and 2, but not for phase 3.

*Lines 398-399: For every metal phase experiment, were the samples cultured in the spiked multi-element solution (for that particular phase) for the cultured duration of 21 days. Example: For M2 culture, were the samples spiked with M2-concentration (Table 1) for the whole duration of 21 days? Please see Lines 229-230. Please explain.*

- Yes, the culturing water from which the samples were taken, was maintained with M2 concentration for the 21 days. As this concentration was expected to decrease over the 21 days of culturing, a smaller aliquot of the stock solution (Phase 1 = 0.1 ml, Phase 2 = 1 ml, Phase 3 = 10 ml) was added twice a week to keep the concentration stable (see above). This will be clarified in the revised manuscript.

*Lines 429-432: In the experiments, there are two systems; Control system (no-spike) and Metal system (spike); Line 374. Each system has 4 phases (0,1,2,3), and phase 0 representing no addition of heavy metals (Lines 200-203). Lines 430-431 says ....when the concentration of these metals in the culturing medium was higher. Please explain.*

- Yes, no extra metals were added to the Control system. Nevertheless, there are differences in the metal concentrations, which can be seen in figure 3 and table 4. These changes in concentration can occur due to exchanges with artificial seawater, which contains a certain amount of metals, or by the release of metals like Cu leaching from brass-made system parts.

*Line 433: does the phase 3 here belongs to metal system. Add to text.*

- Yes.

**Discussions**

*Line 487: Fig. B1; Line 830. The TE/Ca values, for most studied metals, are nearly same for phases 0, 1, and 2 of the metal system, although phase 2 is 10x more than phase 1 (Table 1), and 0 being the control phase. Are there any measurements of spiked-seawater (stock and dilutions) before adding to the culture medium? The culture seawater of phases 1 and 2 (in metal system) should show elevated concentrations (in proportions) compared to phase 0, but is not the case. Please explain why?*

- First question: No, the stock solution was not measured prior to the introduction to the system. No dilutions were made and the stock solution was added directly to the system in different amounts, depending on the concentrations required for the specific phase.

- Second question: Yes, the metal concentration in phases 1 and 2 should be elevated, which is the case for some elements (e.g., Sn, Hg for phase 2). However, there is no elevation visible in most cases. Sorption to surfaces or the uptake of metals by the foraminifera and algae are possible reasons, but no explicit cause could be identified. This is already discussed in chapter 4.1 "Experimental Uncertainties" and is why we monitored the metal concentrations so closely.

*Lines 493-495: ....are smaller than expected for phases 0, 1, and 2. Phase 0 is mentioned as the control phase with no addition of heavy metals (Lines 200-201). Then why is the metal concentrations of phase 0 smaller than expected (for normal seawater?). is it also because of reasons mentioned in subsequent lines.*

- Phase 0 was accidentally mentioned in this context, which needs to be corrected.

*Lines 521-522: Is it possible that the low level of food supplies (as inferred from lack of reproduction) might make the cultured foraminifera specimens weaker and relatively lower amount of metal incorporation in them?*

- We assume that the food supply was sufficient because there was leftover food after the experiments and furthermore, the foraminifera calcified, which also requires enough food. See comment above.

*Lines 615-616: This in turn ..... into the foraminiferal tests. Are the Mn, Zn, and Cu concentrations in the normal seawater (non-polluted) are sufficient as micronutrients considering the fact that these metals are present in the tests of benthic foraminifera recovered from pollution-free environments?*

- The artificial salt used for the culturing medium contained all elements or nutrients that are necessary for marine organisms in a sufficient amount, for most elements at concentrations higher than present in seawater naturally.

*Lines 634-636: Titelboim et al., (2018) based on their studies report that miliolid shells might have advantage over hyaline as bio archives, since they record higher values than rotaliids from the same ambient seawater. As mentioned, DZn values of this study are in good agreement with results from hyaline as well as miliolid foraminifera. Please discuss the findings of this study with the findings from Titelboim et al. 2018.*

- We fully agree with the reviewer and will include Titelboim et al., 2018 in our discussion. The maximum Zn/Ca in our experiments was ~ 68 µmol/mol, which is little lower than reported in Titelboim et al. (2018) (Zn/Ca in *P. calcariformata*= 195

µmol/mol), which may be due to different concentrations in the seawater the foraminifera grew in. It is thinkable, that Zn as a nutrient is in first place used as such and only gets incorporated into the shell after enough Zn was provided to the cell itself. If this was not the case or if the seawater Zn concentrations in our study was not exceeding the necessary nutrient level, this would also explain, why we could not find any correlation between Zn/Ca in seawater and in the foraminiferal calcite. In the revised manuscript version, we will include the findings of Titelboim et al. (2018) into the comparison between species and studies in form of a table or a figure.

*Lines 670-676: The mean Pb distribution coefficients obtained from Amphistegina spp. by Titelboim et al., (2021) is 12.9. Please add this to your discussion.*

- The information from this study will be displayed in a table (and/ or figure) for direct comparison and discussion. However, it should be noted that *Amphistegina* is a tropical symbiont-bearing species and the symbionts can influence the uptake of certain metals. This could facilitate variations in the incorporation of the metals compared to non-symbiont bearing species like in our study.

*Lines 678-679: Please add the results from Sagar et al., 2021b (partition coefficients for Mn, Ni, and Cd from Amphisorus hemprichii).*

- See Comment above.

*Lines 715-717: Figure 4 shows a positive correlation between the concentrations of Cr in culturing medium and in the foraminiferal calcite of Elphidium excavatum. A distribution coefficient of 2.1 has been calculated by the authors for E. excavatum. These results are also stated in lines 707-710. The variability of incorporation in Ammonia spp. and E. excavatum might be because of individual species response. Lines 716-717 are in contrast of the results obtained in this study.*

- We regret this confusion. In the revised manuscript, we will clarify that Cr/Ca values of *E. excavatum* calcite are correlated with the Cr/Ca values in the culturing medium resulting in a $D_{Cr}$ of 2.1. We will also clarify that Cr/Ca values of the two *Ammonia* species are not correlated like this.

*Lines 735-736: When growth is slower, is there a possibility of weak E. excavatum specimens and lower incorporation of artificially elevated heavy metals in the culture medium than what they should have done with preferred food source?*

- It is possible that a more preferred food source would have stimulated enhanced growth and promoted the incorporation of heavy metal into the shells of *E. excavatum*. For instance, the closely related species *E. clavatum* prefers bacillariophycean diatoms (Schönfeld and Numberger, 2007). It may also be possible that *E. excavatum* is simply a slower growing species than *Ammonia*, which seems not to be necessarily connected to a specific food source (e.g. Haynert et al., 2020). This information will be added to the revised manuscript.

*Lines 719-742: Ammonia beccari, Ammonia tepida, have been cultured, by Havach et. al., 2001; Maréchal Abram et al., 2004; De Nooijer et al., 2007; Munsel et al., 2010, for heavy metal partitioning studies. How do their results compare with the findings of this study for Ammonia batava and Ammonia aomoriensis – a comparison table for the common metals*

*should give a clear picture for Ammonia spp. foraminifera. The same can be done for Elphidum spp.*

- We will add a comparison table (or figure) for *Ammonia, Elphidum* and other species from different culturing studies. Furthermore, we can also include a comparison of the heavy metal concentration in the culturing medium that were applied in other studies.

*Line 745; Table 5: The table is a nice compilation of heavy metal contamination studies in various parts of the globe. The studies referred to in the table have used various natural archives such as water, sediment, bacteria, microalgae, living organisms, and others including benthic foraminifera. A column mentioning the natural archives used by the various researchers is important for the readers. This will help them to not only know the polluted regions of the globe but also give them a quick idea of the archives used for those studies, which might help some researchers to pursue similar studies in the area they live and the best archives available at that place.*

- This is maybe a misunderstanding. The Table 5 presented in this study is only comparing the metal concentrations in the seawater. The studies indeed addressed the metal concentrations in various archives but for the comparison to the metal concentrations in the culturing medium of this study, only the seawater values of the other studies were taken into account. This issue is to be clarified in the table.

*Line 749: 'During the past years, many studies were performed to assess the pollution level of seawater.' The natural archives used in the study (for example water, sediments, bacteria, algae, and other should be included in this sentence (See above comment for Table 5).*

- This information will be added to the introduction of the revised manuscript as this is the appropriate place.

*Line 744: The title of this section is: 'Application of TE/Ca values in foraminiferal shell' – The description in the text talks about the range of concentration used in the culture medium in the current study. The concluding lines of the section (Lines 775-776) says "This means that the concentration range of metals covered by this study is adequate for future research and monitoring of polluted systems". The main point of this research work is to see the incorporation levels of elevated heavy metals in the foraminiferal calcite tests so that they can be used as natural bio archives for monitoring of polluted near-shore marine environments (Abstract Lines 12-13).*

- Indeed, the main subject of this study was to address the incorporation of heavy metals into the foraminiferal calcite for using them as natural archives for this environmental signal. Therefore, we decided to shorten or skip this chapter and move the information to an earlier part of the manuscript.

*This section lacks the description on the results (TE/Ca values in foraminiferal shell) obtained in this study from culture experiments with A. aomoriensis, A. batava and E. excavatum and their application as potential pollution indicators. This section needs modification.*

- We agree with the reviewer. The section will be removed. See Comment above.

**Conclusions**

*It will be helpful for researchers and readers to pick important findings from this study. Those may be written in point format. For example: '1) All three species showed a strong positive correlation between Pb and Ag in the culture water and their calcite.' The authors should mention the distribution coefficient obtained for these metals from their studies. Others important findings be written in point format.*

- This is a good idea and will help the reader to pick up take-home messages from this manuscript.

*Line 801: 'the presented DTE's' – The DTE's obtained be mentioned here – also which DTE should be chosen, with phase 3 or without, be mentioned in this section.*

- We agree and will mention this in the revised manuscript.

**New References**

Bernhard, J. M., Blanks, J. K., Hintz, C. J., & Chandler, G. T. (2004). Use of the fluorescent calcite marker calcein to label foraminiferal tests. *The Journal of Foraminiferal Research*, *34*(2), 96-101.

Haynert, K., Gluderer, F., Pollierer, M. M., Scheu, S., & Wehrmann, A. (2020). Food spectrum and habitat-specific diets of benthic Foraminifera from the Wadden Sea–A fatty acid biomarker approach. *Frontiers in Marine Science*, *7*, 815.

Kurtarkar, R. S., Saraswat, R., Kaithwar, A., & Nigam, R. (2019). How will benthic foraminifera respond to warming and changes in productivity?: A Laboratory Culture Study on *Cymbaloporetta plana*. *Acta Geologica Sinica-English Edition*, *93*(1), 175-182.

Lee, J. J., Sang, K., Ter Kuile, B., Strauss, E., Lee, P. J., & Faber, W. W. (1991). Nutritional and related experiments on laboratory maintenance of three species of symbiont-bearing, large foraminifera. *Marine Biology*, *109*(3), 417-425.

Sagar, N., Sadekov, A., Jenner, T., Chapuis, L., Scott, P., Choudhary, M., & McCulloch, M. (2021b). Heavy metal incorporation in foraminiferal calcite under variable environmental and acute level seawater pollution: multi-element culture experiments for *Amphisorus hemprichii*. *Environmental Science and Pollution Research*, 1-14.

---

## Author Comment (AC2)

**Response to Reviewer#2 comments**

In the following, the Reviewers' comments or questions on the manuscript are given in black italics, and our response is highlighted in blue and indented. We only consider points that provide information and clarifications that are of general interest to the readership. Minor points like " Line 258: add a comma (,) after samples" or "Break this sentence into two" will be performed during the manuscript revision, and hence are not to be discussed here.

*The authors are going too far in their conclusions: in the abstract lines 25-27 "Our calibrations and the calculated partition coefficients… enable the direct quantification of metals in polluted and pristine environments" and in conclusion lines 801-802 "The presented DTE´s allow a direct quantification of metal concentrations in polluted and pristine areas". First, given the really high DTE ranges found in this study (including or not phase 3) and/or DTE (from linear regression) strongly based on the phase 3 data point where the seawater element concentrations are variable, it is not possible to maintain that "quantification" of metal pollution in natural environments is possible. Secondly, the authors are contradicting themselves since they explain in the introduction that a mix of metal may result in interactions that can lead to different incorporation of the metal. Therefore, the mix proposed in this study, which is peculiar since including 10 trace metals at a time (in polluted environment, most often only 1 or 2 metals are above the threshold limit, not 10 at the same time), is not representative of other type of pollutions. The authors should be more measured in their conclusion. The elemental concentration in the shell may definitely be used to look at relative variation of heavy metal concentrations in the seawater through time and space but definitely not to give quantitative data… and only for elements where a positive correlation has been found between TE/Ca foram and seawater.*

- We respectfully disagree. Any approximation of environmental signals by using the elemental or isotope composition of calcareous shells grown in the respective environment is only that precise as the variability of the calibration data set. The uncertainties are discussed in the paper indeed. Furthermore, the heavy metal mixture we applied was found very often in natural environments, for example in the vicinity of harbors or bigger cities. Some examples are given in Table 5 of our manuscript. This is only a brief overview and much more work on this is available. Therefore, our metal mix is definitely representative for heavy metal pollution in near-shore environments. Indeed, it was intended in this study too investigate the impact of metals that do co-occur and are potentially interacting.

*I think that, instead of describing and discussing each trends or absence of trends observed, they should maybe realize that the absence of systematic tendency (within one element or one species) is unexpected and might be the result of multi-metal experiment since singe metal culture exhibit usually positive correlation between shell and seawater element ratios (cf literature). I would also advise to elaborate more the interspecific differences and maybe on the new elements that have never been measured before.*

- We have specifically chosen to conduct experiments that mimic natural conditions as much as possible. This brings added complexity but by carefully monitoring the changes in culturing medium metal concentrations the results are robust. It is important scientifically to report non-results but we will focus more on the elements with linear partitioning and species differences in the revised manuscript.

*To my opinion, there is confusion between the toxicity of metals to the organism and their incorporation into the shell (cf line 27 "This in turn allows monitoring of the ecosystem 27 status of areas"). What are expected from environmental/governmental studies is to evaluate the impact of heavy metal concentrations on the organism life (ability to survive, grow, reproduce…). Here the authors measure the elemental concentration in the shell. The speciation of the metal incorporated in the shell might be different from the one causing toxicity and bioaccumulation in the cell. The elemental concentration in the shell may help to reconstruct variations of seawater elemental concentration but for the moment, the link between this concentration and its effect on marine life is still unknown (and may depends on elements!). I think that the authors should discuss more precisely about this aspect.*

- We take the message to more precisely distinguish between toxicity, bioavailability and incorporation into the shell at respective sections in the manuscript and will rephrase the sentence.

*I have problem to understand different aspects regarding the metal mix solution:*

1. *How and where was added this solution? Was it added in the supply tank located on top of the system? In this case, knowing that the pump is flowing at 0.017ml/s, how long would it take to replace and reach the same metal concentration in the culturing vessel (ie Tupperware) as in the tank?*

- The solution was added in the supply tank and according to the flow rate, the metal concentration in the culturing vessels would be the same as in the supply tank after a few hours. Not only the flow rate and water exchange in the vessels but also sorption processes can have an effect on the metal concentration, which is addressed in the manuscript in detail.

2. *When was added the solution? According to line 206, we understand that this is added once before each phase. But on line 229-230, it is written "For keeping the metal concentration at the same level over the different culturing phases, water with elevated heavy metal concentrations was fed into the system bi-weekly.". I'm lost, what is this "water with elevated heavy metal concentrations" you are referring to? Is it the stock solution? This need to be clarified…*

- The stock solution was added at the beginning of each phase to reach the targeted concentration. Additionally, a smaller aliquot of the same stock solution, termed "water with elevated metal concentration", was introduced twice a week during the three week of a phase, because a loss of metals during the culturing phase was expected, e.g., because of the uptake by foraminifera or algae or because of adsorption to surfaces (see Response to Reviewer 1 above).

3. *Where are taken the samples for trace metal analyses in the seawater? I don't think the information is given (Lines 215-216, line 234 "from both systems")… Are they taken at the outflow of the vessels so that it really corresponds to the concentration of the seawater in which the foraminifera are growing? Or they were sampled in one of the tank, which would be of course less precise…*

- The water samples were taken from the supply tank. This information is to be added in the Methods chapter. This point in the system was considered appropriate because the high flow rate and hence well-mixed system facilitated a representative sampling.

4. *How often was measured the metal concentrations in seawater over the course of the experiment. This should be indicated in the material and method part. For the moment, it is written "frequently" (line 216 and 234) however, when looking at Figure B1, only 1 to 4 data point are available within each phase. This is to my opinion problematic when applying individual curve fit for every phase to calculate the weighted mean value… (see comment after)*

- Fourteen samples were taken from the metal system over the course of the whole experiment, which can be seen in Table A1. From beginning of phase 1, sampling took place twice a week. Indeed, the metal concentration was expected to be more stable during the culturing phase, which was why sampling twice a week was considered as appropriate. Nevertheless, the application of a fit curve for every phase is in our opinion the only way to approach a representative mean value for a given phase especially when taking into account that the metal concentration in the culturing medium varied.

5. *I don't understand the calculation in table 1. The factor between each phase is 10 times but on line 207, it is written "phase 1 = 1 ml, phase 2 = 10 ml, phase 3 = 150 ml". How were calculated these target values?*

- In phase one, 1 ml of the stock solution was added and in phase 2, 10 ml of the same stock solution were added, which is the factor of 10 mentioned above. The target values were not calculated they were taken from the literature (see line 208 – 213). Based on this, we calculated how much of the stock solution was needed to be added to the system for each phase.

6. *The authors used "stock solution" all over the manuscript when referring to the metal mix solution. However, I think it would help to clarify this on line 205 when you first used this term so that it is clear in the discussion (on line 487 and after) that you talk about this stock metal mix solution.*

- This will be clarified in the revised manuscript.

*The culturing system description is very precise but also very long and all the details makes it difficult to understand the general principle. I think that the authors would gain clarity if they explain earlier that a different vessel, freshly filled with calcein labelled forams (if this is correct), is incubated for each different phase of the experiment. At the moment, this essential information appears only (if I'm correct!) on lines 222-223 whereas it should already be said in chapter 2.2.2 or at least beginning of 2.2.4. The description would also be clearer by keeping the same term to describe the same "object".*

- We agree and will move this information to an earlier point in the manuscript. Furthermore, the description is to be rewritten taking into account that the same terms are used for corresponding parts of the culturing system.

**Abstract**

*Line 17: "Seawater analysis… between culturing phases". This sentence is not 100% correct since the increases between phase 0, 1 and 2 are not very obvious for all elements (e.g. Cu, Mn). This is however clear for phase 3.*

- This needs to be specified.

*Line 24-25: I have the idea that Zn and Cd are showing variations that are more or less similar to other elements (eg. Cd like Pb), no?*

- Indeed, the small variation applies to Sn only but not to Cd and Zn.

**Introduction**

*There is confusion between "heavy metals" and "trace metals" throughout the manuscript. To my knowledge, the 10 metals studied here are not all considered as "heavy" metals, some are trace metals. I think that this depends on the atomic weight of the element… Please check and use the appropriate terms.*

- "Heavy metal" as term is not clearly defined. This issue is nicely described in Duffus et al (2002). The atomic weight is one possible criterion, but no threshold is set for a minimum weight an element must have to be considered as "heavy metal". Some authors pretend that the atomic weight needs to be greater than sodium, which would apply for all of our metals, and others take Hg or Ca as a boundary weight. Another criterion is the density. The boundary value for this parameter is ranging between 3.5 and 7.0 $g/cm^3$ depending on the author. Other criteria involve their behavior as Lewis acids… It is therefore difficult to apply an "appropriate" term, but we will define our use of the term in the outset of the revised manuscript.

*Lines 69-75: here you talk about the physiological effects. This is interesting but you are looking at the incorporation in the shell which is different (cf comment earlier). The information is correct but it has to be clear that you will not have a look at this aspect yourself in this study.*

- See Response to Reviewer 1.

*Line 86: "bioavailability". I guess that this is correct to say that if the element is found in the shell, it is bioavailable since it might (depending on the biomineralisation process involved) goes through the cell. However, I would say that this is different from toxicity effect (cf comment earlier).*

- Bioavailability and toxicity are definitely different and the sentence needs to be reformulated to appropriately discriminate these two terms.

*Line 46: check in Kotthoff et al. (2017) that Mn/Ca is actually used for O2 or redox reconstructions and not for contamination.*

- This is true and should be added at this place.

*Line 53: These species are also dominant in intertidal mudflats, not only subtidal areas.*

- This is a misunderstanding. "near-shore" does not mean subtidal as the "shore" is legally defined by the Mean High Water level. As such, mudflats are well near the shore. Perhaps it is more precise to say "intertidal and shelf environments".

**Material and Methods**

*It would be nice to document with SEM pictures and light pictures the 3 species of this study. I think it is even more important knowing that Ammonia and Elphidium are species rising lot of identification discussions! Whatever the name given, it is essential to have to possibility to look at the picture and compare it to literature and also recent DNA papers.*

- This is not necessary. The species from our sampling locations are already well documented in the literature (Lutze, 1965; Nikulina et al., 2008; Schweizer et al., 2011; Francescangeli et al., 2021; Schmidt and Schönfeld, 2021). We will add this information and citations to the revised version of the manuscript.

*Lines 118-121: I am wondering if this information is relevant for the manuscript. Since the text is too long, I would suggest to delete this part. Also, the authors mention cores sampled for ecological study which are not presented in the manuscript. This is maybe not necessary?*

- We will delete these lines.

*Lines 138-150: There are too many details here (eg the size of the petri dish). Some information is repeated several times. For example, the fact that the authors checked several times to be sure that the forams were alive (lines 142-143: "glossy, transparent and undamaged test... cytoplasm present", line147 "structural infill of cytoplasm", line 151 "the color of the cytoplasm was checked"). I don't think the precision of this check at each step is necessary... The important information is that the forams used at the end in the experiment were labelled with calcein and exhibited a green cytoplasm proving that they were active.*

- We will shorten this section where possible but avoid losing any important information in the revised manuscript.

*Lines 151-156: I had some difficulties to understand (when I first read the manuscript) when this labelling step happened? Is it only once at the beginning of the entire experiment (before phase 0)? But in this case, the forams added for example at phase 3 could have calcified new chambers in the meantime... Or is it done before each phase in order to add freshly labelled forams in the new introduced vessels? Here the authors should precise this aspect.*

- The labelling took place before each phase to ensure that freshly labelled foraminifera are inserted in the well plates. This will be clarified in the revised manuscript.

*Although the culturing system is well described, it is difficult to not get lost since everything is described with lots of details. Therefore I would recommend to always use the same term when describing one part of it (eg "vessels" for the box containing the well plates, that you should name this way on line 186). On line 195-196, the term chamber is used but we do not really know to what it refers to: well-plate cavities? Vessel? Please try to keep it simple and clear.*

- We will clarify this section and we will also consistently use the same terms for respective parts in the culturing system.

*Lines 223-224: One vessel was left from phase 0 to phase 4 (84 days). What was the interest of this vessel? Were the forams from this vessel analysed? If this is not the case, you should say it to avoid any confusion!*

- The interest was to have a look at the metal incorporation during all four phases in one individual specimen, but the foraminifera have not been analysed yet. We will delete

this sentence to avoid any confusions.

*Lines 286-290 "the total number of chambers was counted before and after the experiment for every specimen (Table 2)": I don't see the interest of counting all the chambers of each foraminifera before and after the phase since the authors used calcein. And this information is not given in Table 2. Moreover, I agree that this is possible to count the total number of chamber in Ammonia species since they are trochospiral. However, this is not the case for Elphidium species since spires of new chambers recover the initial chambers! Therefore, if forams where indeed labelled with calcein just before their introduction into the culture system, I would keep it simple and only mention calcein to identify newly formed chambers.*

- Chamber counting was to double check if foraminifera grew during the experiment, because calcein staining may eventually fail. This needs to be stated in the manuscript.

*Line 312-313: Could you explain why you chose to use NIST612 for calibration and monitoring of instrument drift since the elemental concentrations in this standard are way above the concentrations found in the forams? Moreover, you chose to use a glass standard as quality control whereas it would be more appropriate, to my opinion, to use a carbonate standard with similar matrix to your forams. Moreover, the conditions are similar between carbonate standards and forams (I guess) whereas NIST standards are measured with higher energy and frequency. Please explain.*

- The glass standard was chosen because all elements of interest but Hg are reported in the literature, which is not the case for carbonate standards. For further quality control, a variety of carbonate-based reference materials have been measured. All values can be found in Table A3 in the appendix. Furthermore, Dueñas-Bohórquez et al. (2009) demonstrated that different energy density between the foraminiferal calcite and the glass standard does not affect the Laser ablation analyses.

*Line 334: The authors considered the data as usable if above LOD. However, the limit above which the data can be used for quantitative purposes is commonly defined as the LOQ (limit of quantification). This is defined as 10*SD of the blank. How many data would be excluded from the dataset if the authors use LOQ instead of LOD?*

- We can check the LOQ.

*Lines 364-365: It is not described in the Material and Methods how the living forams were differentiate from the dead ones at the end of each phase. Did the forams lost the colour of the cytoplasm (or their cytoplasm itself) so quickly that you could see it?*

- Indeed, the foraminifera loose the color of their cytoplasm quickly. Furthermore, they do not gather particles or food any more, thus are lacking a detritus cyst before their aperture.

*Line 102: what is Hallig Hooge? Is it still on the field?*

- "Hallig Hooge" is an island in the North Frisian Wadden Sea and yes, it is yet still there.

*Line 225: Use PSU everywhere or even no unit at all for salinity.*

- We agree, this needs to be unified.

*Figure 2a: If I understood properly, there were only 2 vessels per incubator so, to avoid confusion, you should remove 6 of the 8 vessels drawn in figure 2a.*

- This of course needs to be adjusted.

*Figure 2e: this picture is not very clear. Is the shell of the foram entirely fluorescent (ie born in calcein bath)? Otherwise, how many chambers are labelled here? I have the feeling that this is the cytoplasm that exhibit high fluorescence at the bottom since the fluorescence is patchy and fill half of the last chamber...Could you try to show a better picture?*

- No, the shell is not entirely labelled. Only the last 2 ½ chambers are labelled. It can be excluded that only the cytoplasm is fluorecenting because the specimen was dead, cleaned and dried. Therefore, no cytoplasm should be there anymore.

*Line 152: Why did the authors used a concentration of 16mg/L which is different from the recommended concentration given by Bernhard et al. (2006)?*

- See Reply to Reviewer 1.

*Line 156: To my opinion, this is not enough time to remove the calcein from the vesicles in the cytoplasm. Anyway, if this seawater is used to calcify 1 new chamber in your experiments, you can hope that this new chamber would exhibit a small fluorescence.*

- Reviewer 2 is right. A sufficient time is needed to remove the calcein from seawater vesicles in the cytoplasm. If the foraminifera are taken directly from the calcein staining bath for incubation, all subsequent chambers will be stained (see Haynert et al., 2011). In our case, the youngest chambers were not stained in that a purification time of 1 or 2 days was sufficient.

*Line 160: Dagan et al., 2016 is a report. Is it available online somewhere?*

- The report is not available to the public but Woehle et al., 2018 reported the experimental setup as well in the online supplement. Dagan et al., 2016 is therefore to be deleted.

*Line 171: it is the air that was filtered?*

- Yes.

*Line 171: The authors do not mentioned pH or alkalinity measurements. Did they measure carbonate chemistry during the experiments? At least pH has been measure since it is mentioned in discussion on line 580 "As the pH during the experiment was stable around 8.0 ± 0.1 (measured twice a week)". This information should arrive in material and methods.*

- Carbonate chemistry was not measured during the experiment. We will add the information, that pH was measured in the "Material and Methods" chapter.

**Results**

*Table 2: In C2 for A. aomoriensis, does it mean that on the 10 forams recovered, 2 were dead but all of them (10) had calcified new chambers?*

- Yes.

*Line 368: Since the Ammonia calcified usually more than 4 new chambers, is it possible to see the evolution of seawater metal concentration in the successive chambers of 1 given individual? At least in phase 3? This could help to gain precision in the estimated DTE…*

- The evolution of the metal concentration in seawater of phase 3 was only indicated in some individuals *of Ammonia aomoriensis* and *Ammonia batava*. Particularly, the first high concentration of certain heavy metals could be found in the first chambers after the staining (i.e. the first chamber built in culture). But this was not the case for all individuals, which is most likely due to the individual timing of calcification. It also cannot be determined, at which point in time the foraminifera calcified within one phase. Therefore, a mean value over the whole culturing phase is most representative.

*Figure B1: Could you indicate the error of the measurement on the graph? ON line 340-344, the authors explained that they fit a regression curve on the data to calculate a weighted mean per phase. This seems a good idea when 4 data points are available within a given phase and that a trend can be seen (eg phase 3 for Cr, Ag, Sn). However, this seems difficult when only 2 data points are available and very different (eg Cu) or when the trend is not regular (eg phase 3 for Mn, Ni…). Actually, did you realise that Mn, Ni, Zn and Cd show similar variation though time in phase 3 (lower value at the second sampling time) compared to Cr, Ag, Sn or Pb which show decreasing trends?*

- As these are single measurements, the error that could be provided would be based on frequent measurements of the seawater reference materials. The respective values are given in Table A2 but will also be added to this figure.

- When only 2 data points are available a linear regression was made, which is in our opinion the only way to account for the different concentrations because we do not know at which time within a phase the new chamber was built. If no clear trend was observed, the regression with the highest fit (highest p-value) was chosen.

- It is indeed interesting that Mn, Ni, Zn and Cd show similar patterns in phase 3. We could think about possible mechanism affecting all of these metals at the same time.

*Figure 3: I have the idea that the use of weighted means and standard error of the mean instead of standard deviations, the authors reduce artificially a lot the real elemental variations that they have, mainly in phase 3. Maybe the figure could be completed showing the range of values actually measured in shadow or use box plot to better represent the variability of this artificially created dataset...*

- It is true, that the variation carry less weight in this figure and this is why we added figure B1 in the appendix. Nevertheless, the variability in the seawater during one phase can be added to this graph or an extra Box-Plot can be created to clarify this.

*Figure 4:*

*How are calculated the statistics of the correlations? These correlations should not be based only on the mean values per phase but on the all data set. For example for Ag and Pb, the R²and p values are really good but the D is only based on the Phase 3 data which has a high variability! Therefore the D value is not precise and robust.*

- The statistics of figure 4 are indeed based on the mean value per phase and not on the entire data set. The plots were made using the software Grapher, which is calculating statistics along plotting. Furthermore, the program PAST was used to calculate statistics. We actually also calculated $R^2$ and p-value based on the whole data set, which was e.g., for *A. aomoriensis* with phase 3 comparable to the statistics based on the mean. This is why we decided to go with the mean values. We can add this information to the manuscript.

*Figure 4 and Table 4: The authors have no objective reasons to fit the correlation through 0 for some elements and not for others. It could be decided on statistical arguments but I have the idea that the authors did not check this.*

- We tried to fit all element correlations in all species through the origin, because a real correlation would also include the origin. Only in cases where this was clearly not possible (Mn of *A. batava* with phase 3 and Hg of *E. excavatum* without phase 3), because the course of the regression line changed significantly or the $R^2$ value decreased, no forcing through the origin was applied. We can clarify this in the revised version of the manuscript.

*For A. aomoriensis Mn/Ca, there is a problem with the correlation line. This is not possible that the line don't go through the phase 3 datapoint. Please check.*

- The line is not going through the data point from phase 3 because the line is forced through the origin, which is changing the course of the line minimal. It should be clear to the readership that not only the data point from phase 3 but also the data from the other phases are driving the course of the regression line. Furthermore, the $R^2$ – value of the regression line did not decrease when forcing through the origin, this is why we decided to include the origin.

*The graphs for this figure should have similar y axis range for a given element for the 3 species so that the difference of incorporation between species is highlighted. All graphs should start at 0 on the y and x axis. I think that the main (and most robust) output of this study is the difference of incorporation between Ammonia and Elphidium species and this is at the moment only shortly discussed and observable in graphs. This is a shame.*

- The axis can be adjusted and the differences between species will be more of a focus in the revised manuscript.

*This is a really good idea that the authors also analysed their data without the data from phase 3. To my opinion, this phase is important to get a trend because the problem when you remove it is that you have no correlation anymore, probably because the range of seawater elemental concentration is not wide enough. On the other hand, when phase 3 is considered, then a more relevant D value can be calculated but the correlation are only based on this data points and therefore the correlation is not statistically robust.*

- Yes, it is true that the correlation gets lost because the range of the metals in the seawater is very narrow without phase 3. Furthermore, it is also true that point three makes the correlation statistically more robust but nevertheless, figure B2 also shows that the general trend is still visible without phase 3 for some elements. Forcing through the origin further adds a fix point, which provides at 3 points, though artificially, and not 2 only. We can clarify this in the revised version of the manuscript.

*Line 473: Now authors are removing phase 3 and 2?*
- We did not remove phase 2 from the calculation of the regression line but if one has a look at every data point from a phase individually (meaning without any calculation of regression), the $D_{Cd}$ and $D_{Cr}$ values from phase 0 and 1 are >1 while the $D_{Cd}$ and $D_{Cr}$ from phase 2 and 3 are <1. This needs to be clarified and rephrased.

*Line 464 to 481: this is very descriptive and difficult to follow…*
- As this is part of the "Results" section, a description of the data is appropriate. We rearrange the paragraph to make it easier to follow for the reader and focus more on the results that will be discussed later in the manuscript.

**Discussion**

*It is not possible to discuss the significance and meaning of partitioning coefficient that are showing a very high range since this variation is meaningless to my opinion in terms of biomineralisation processes… For example, DCd are varying from values below 1 to values such as 10-20 even 50 in all species (lines 678-679). In terms of incorporation mechanisms, that would mean that some specimens are fractionating against Cd whereas some others (from the same species and in the same condition) would concentrate this element! I would suggest to the author to rather focus on:*

*Elements were a positive correlation is found but instead of using the mean TE/Ca value (eg line 557-559), they should take into account the variability of the data and give a SD for the slope (ie for the DTE). They should also be aware and acknowledge in the manuscript that these correlations are driven by the phase 3 data and might be imprecise.*
- It is probably useful to add the SD for the slope of the partition coefficients and to go more into detail concerning the uncertainties of the calculated $D_{TE}$. It is also reasonable to note that the correlation is driven by the data point from phase 3. We will add this to clarify the circumstances for the reader.

*Elements were the range is relatively low so that a general tendency/interpretation might be given.*
- We can separate the elements with a smaller variability and discuss the behavior of those elements individually.

*Finally, do not discuss further forward the other elements that exhibit vary wide DTE also if no literature is available on this element, it is interesting to know that this is incorporated and measurable in foraminiferal calcite.*

- We may shorten the discussion of the elements with higher variability. But nevertheless, a proxy is only as good as it´s variability and therefore we think, that it is important to mention variable $D_{TE}$´s too.

*I have the feeling that the authors use the DTE with or without phase 3 when it helps them to compare with the literature. This is bothering me: is phase 3 really usable to calculate a partitioning coefficient knowing that the seawater concentration of the metal was not stable during this phase and the regression line is totally driven by this single condition?*

- The regression is not only driven by the data point from phase 3, because other points and the origin also play a role, which is already demonstrated above and can be seen in figure B2. But nevertheless, phase 3 is very much driving the slope of the regression line. Even though the seawater concentration was not as stable as during phases 1 or 2, we are convinced that it is appropriate and justified to use a mean value calculated from the individual fit curve for every element and to create it to the mean value of the foraminiferal calcite. It is possible, that the variability of the seawater concentration in our study is higher because we measured more often than other studies did. This means that other studies simply not monitor the variability. Furthermore, pollution events in nature are also transient events rather than stable once. We can discuss this maybe in the "Experimental Uncertainties" section a little further.

*Line 506: The authors mentioned the growth of algae as a reason for element concentration changes in the seawater but I understood that the algae were given dead. Therefore, one would not expect algal growth in the experimental set up?*

- This is a misunderstanding. The algae that were fed were dead, but germs of other algae were introduced without purpose together with the living foraminifera and grew during the experiment. These algae preferentially grow on plastic surfaces and create biofilms. Therefore, it is well possible that these films also took up metals.

*Line 523-528: this paragraph should be more or less upside down. Since you used calcein prior to the experiment, you do not have to worry that this probe could have impacted the elemental concentration in your forams. This paragraph could therefore be shorten.*

- We agree.

*Lines 551-552: according to Erez endocytosis biomineralisation, I thought that the composition of the seawater vesicle (ie Mg content) was also modified somehow?*

- Yes, this is partially correct. "Endocytosis" as such describes only the uptake of a seawater vacuole, which is subsequently modified during their pathway in the cell. This needs to be clarified.

*Lines 559-561: this is interesting but where can we see this information (ie. D vs seawater trace element concentration)?*

- Figure 4 shows this indirectly, we will refer to this figure.

*Line 559: if D>1, this means that the foram is concentrating the element inside its shell. Therefore, I would not define this as a "non-selective uptake", no?*

- "Non-selective" at this point referrers to an uptake that is not driven by the chemical property of the ion size of the metal ion itself. This can be clarified in the revised version of the manuscript.

*Line 561: other studies have observed the same trend of decreasing D with increasing seawater concentrations: Mewes et al. (2015) for Mg and Barras et al. (2018) for Mn.*

- The references will be taken into account.

*Figure 5: The authors refer to this figure for each element but I think that this is also interesting to observe that there is apparently no trend between D and the ionic radius to charge ratio.*

- Yes, this is true, but this figure should mainly provide information whether the $D_{TE}$ is higher or lower than 1 to the reader. Nevertheless, we can add that no clear trend between $D_{TE}$ and the ionic radius is observed. It would maybe make sense to remove this figure.

*Figure 5: it is strange to me that the author used a single data point for each LogD value. Is it the mean of all measurements? In this case, it would be nice to see the SD since D might be highly variable.*

- Yes, this is the mean value in cases were no significant correlation between the heavy metal concentrations in seawater and calcite was found. In cases were a correlation was significant, the slope of the regression line was used. Indeed, $D_{TE}$ is variable and the SD will be added in the revised manuscript.

*The authors compare their D values to the literature. Sometimes they compare these values to tropical symbiont-bearing large benthic forams (high Mg content species) or miliolids (line 635, 671, 708) which are known to incorporate much more elements than Ammonia for example and other small benthic foraminifera (low Mg species) (cf van Dijk et al.,2017). This should be specified and discussed.*

- See Reply to Reviewer 1.

*Chapter 4.3: as mentioned before I think that Figure 4 should be reworked (or a new figure) in order to observe more easily the differences between species (e.g. similar axis for Ammonia species and different (if needed) for Elphidium). Maybe differences between species would be even better observed when considering only phases where the seawater elemental concentrations are stable?*

- See Response to Reviewer 1. Stable concentrations occurred in phases 0 to 2 and only phase 3 had higher variations in the trace element concentrations. Therefore, figure B2 in the appendix, which is showing TE/Ca in calcite versus TE/Ca in seawater without phase 3, can clarify the species-specific differences in the heavy metal incorporation. This figure will be adapted in the same way as Figure 4 and we will add a figure comparing different species.

*Line 724-725: Food is added quite regularly during the experiment. Could the deposition of a layer of food at the surface of the "sediment" could create microenvironments within the hole of the weel-plate? Indeed, the food would be degraded and could influence pH and O2 conditions for exemple…*

- It is indeed possible that the food deposited as a thin layer on top of the sediment, which could have created a microhabitat. This effect would be the same for all cavities and therefore for all three species. In account of this, species –specific differences in the heavy metal incorporation cannot be caused by this effect. This will be mentioned in the revised version of the manuscript.

*Line 727: Read van Dijk et al. (2017) paper but I don't think that the hhigh elemental incorporation of symbiont bearing forams is due to the presence of symbionts but rather to the fact that they are high-Mg content species. Other symbiont barren large benthic forams exhibit high elemental incorporation.*

- We agree, the high Mg-content of the calcium carbonate of the species in the tropics at high temperatures and salinities could play a role, which will be discussed in the revises version of the manuscript.

*Line 735-737: be aware that there is a difference between number of chamber added (individual growth rate) and calcification rate (crystal growth rate). Depending on the element, one could expect that slower calcificiation would give more time to remove (or discriminate more against) the element as it is the case for example for Mg.*

- This is an interesting aspect, which we will include in the discussion.

*Line 738-742: this is an interesting point. I think the authors could potentially unravel this problem if they compare Elphidium data with the first chambers calcified after the calcein stained chamber. Indeed, that would be the forst chamber calcified in the experiment when the seawater elemental concentration was probably the highest. Moreover, as previously mentioned, you could have a look at successive chamber composition to see if you can observe a decreasing elemental composition for the elements exhibiting decreasing trend in seawater.*

- *Elphidium* mostly build only one chamber, which means that the data presented here are already from the first chamber calcified after staining. This makes a tracking of the decreasing concentration impossible. For both *Ammonia* species, see comment above.

*Table 5: this table is very interesting and complete but to my opinion, it could be moved in supplementary materials.*

- We can move the table to the supplements.

*Table 5 : how were the metals analysed in these studies? Analytical techniques used? Extractions? Speciation of the metal?*

- This can add the information, which analytical technique was used to determine the heavy metal concentratin in the comparing studies.

*Line 707: this paper from Remmelzwaal refers to post-depositional overprinting. I don't kown this study but are you sure that this DCr corresponds to primary calcite values?*

- Yes. They performed culturing experiments with different foraminiferal species and calculated this D$_{Cr}$ based on these experiments.

**Conclusion**

*I think that the authors could highlight the interest to use fossil records (or regular sampling of living forams through time or space) to determine the relative variations of seawater metal concentrations in porewater through time. Although quantitative reconstructions are to my opinion not feasible at the moment, relative variations are usable for elements where a correlation was observed between shell and seawater ratios (not for all elements). The authors should be more realistic in their conclusions.*

- We agree, reference to the fossil record is to be given.

*The authors could also highlight the interest of forams as they are integrating in their shell the metal concentration over a certain period of time. Indeed, dissolved metal concentrations measured directly in seawater (for monitoring purposes) give the concentration the day of the sampling but this concentration may vary very rapidly… Both aspects should even be mentioned already in the introduction.*

- We agree that foraminifera offer the opportunity for long- and short-term monitoring of changes in the heavy metal concentration, because they are recording the environmental signal. This will be added in the introduction and in the conclusions.

*Line 795-796: ok but there is no impact on survival or growth in your experiments.*

- Any organism reacts in a protective way before harmful or lethal effects do occur. This is also why a reduced incorporation of a certain metal could point towards the onset of a protective mechanism prior to damage of the organism and may also prior to a reduced growth and following death.

**New References**

Dueñas-Bohórquez, A., da Rocha, R. E., Kuroyanagi, A., Bijma, J., & Reichart, G. J. (2009). Effect of salinity and seawater calcite saturation state on Mg and Sr incorporation in cultured planktonic foraminifera. *Marine Micropaleontology*, *73*(3-4), 178-189.

Duffus, J. H. (2002). " Heavy metals" a meaningless term? (IUPAC Technical Report). *Pure and applied chemistry*, *74*(5), 793-807.

Francescangeli, F., Milker, Y., Bunzel, D., Thomas, H., Norbisrath, M., Schönfeld, J., & Schmiedl, G. (2021). Recent benthic foraminiferal distribution in the Elbe Estuary (North Sea, Germany): A response to environmental stressors. *Estuarine, Coastal and Shelf Science*, *251*, 107198.

Lutze, G. (1965). Zur Foraminiferen-Fauna der Ostsee. *Meyniana*, *15*, 75-142.

Nikulina, A., Polovodova, I., & Schönfeld, J. (2008). Environmental response of living benthic foraminifera in Kiel Fjord, SW Baltic Sea. *eEarth*, *3*, 37-49.

Schweizer, M., Polovodova, I., Nikulina, A., & Schönfeld, J. (2011). Molecular identification of Ammonia and Elphidium species (foraminifera, Rotaliida) from the Kiel Fjord (SW Baltic Sea) with rDNA sequences. *Helgoland Marine Research*, *65*(1), 1-10.

Schmidt, S., & Schönfeld, J. (2021). Living and dead foraminiferal assemblage from the supratidal sand Japsand, North Frisian Wadden Sea: distributional patterns and controlling factors. *Helgoland Marine Research*, *75*(1), 1-22.

---

## Author Response (AR1)

**Heavy metal uptake of near-shore benthic foraminifera during multi-element culturing experiments**

Authors: Schmidt, S., Hathorne, EC., Schönfeld, J., Garbe-Schönberg D.

We sincerely thank the editor and the two reviewers for the constructive comments on the manuscript. We acknowledge that our paper is generally of interest to be published in Biogeoscience. We have addressed all the issues, include a point-by-point response to all comments and questions, and provide a revised manuscript that we believe is substantially improved. In the following, the reviewer comments or questions on the manuscript are given in black italic, and our response is highlighted in blue and indented. The line numbers in this response refer to the revised manuscript with track changes mode on. Furthermore, we spotted a few typos, inconsistencies and flaws along the text, which were corrected as well.

**Response to Reviewer#1 comments**

For interactive discussions:

*1)    Higher distribution coefficient values are reported for symbiont-bearing foraminifera species (Line 727). Is it possible or logical to think that lack of food can make the cultured specimens (non-symbiont foraminifera) weak, which can inhibit the incorporation of heavy metals from the surrounding culture medium?*

-   This is unlikely as a sufficient amount of algal food was added during the experimental period. This was evident by leftovers covering the sediment surfaces in the cavities at the end of each phase. The food would have been consumed by the foraminifera if they would have needed more. Furthermore, the foraminifera calcified, which wouldn't be the case if any undersupply occurred (e.g. Lee et al., 1991; Kurtarkar et al., 2019). Therefore, the nutritional status is unlikely to have influenced the metal uptake by the foraminifera. This information was added (lines 625-629).

*2)    Does the different species of Ammonia show similar rates of incorporation (comparison of this study with published culture-studies with other species of Ammonia?*

-   We had a further look for culture studies addressing the incorporation of metals into the tests of other *Ammonia* species and add this to the discussion. Examples are Munsel et al. (2010) addressing Mn, Ni and Cu in the test of *Ammonia tepida* or Marechal-Abrams et al. (2004), who investigated the Cd incorporation into the shell of *Ammonia beccarii*. Furthermore, de Nooijer et al. (2007) cultured *Ammonia tepida* with different Cu concentrations or van Dijk et al. (2017) investigated the Zn uptake of *Ammonia tepida*. These examples are already integrated into the manuscript, but we went further into detail and pointed out the species-specific differences (lines from 905). Furthermore, we added a comparing Figure 5.

*3)    DZn obtained in this study are in good agreement with hyaline species and also a miliolid (Lines 634-636). Some previous work on several miliolid species report elevated concentrations of Zn in their shell compared to the ambient seawater. It would be interesting to discuss about the difference in metal incorporation between miliolid and rotalid species in the discussion section 4.3 Interspecies variability.*

-   We agree with the reviewer. It is interesting to note that our study compared to both miliolid and rotaliid species. The differences in calcification of miliolid and rotaliid foraminifera and its implication on the heavy metal concentration in the foraminiferal calcite was discusses further in chapter 4.3 (starting from line 905) and we added a comparing figure (new Fig. 5).

**Abstract**

*Line 11: change 'foraminiferal tests' calcite to 'foraminiferal calcite tests'.*

-   Done (line 12).

*From the culture experiments, Pb and Ag are seen to incorporate linearly (Figure 4) in the new calcite of all three foraminiferal species. It would value if the distribution coefficients obtained for these metals are added in the abstract.*

-   The $D_{TE}$ of these elements were added to the abstract (apparent partition coefficients for Ag: *A. aomoriensis* = 0.50, *A. batava* = 0.17, *E. excavatum* = 0.47; for Pb: *A. aomoriensis* = 0.39, *A. batava* = 0.52, *E. excavatum* = 0.91) (lines 25-27).

**Introduction**

*Line 65: another multi-element culture experiments on large benthic foraminifera Amphisorus hemprichii reports the proportional enrichment of Mn, Ni and Cd (Sagar et al., 2021) in the foraminiferal tests from the culture solutions, and the thresholds*

-   We thank the reviewer for pointing out this recently published study, the results of which are considered in the revised manuscript. For instance, Sagar et al. (2021b) found that the partition coefficient of *Amphisorus hemprichii* for Mn was 1.3±0.2, which is slightly higher, but in the same order of magnitude than our $D_{Mn}$ values. The presented $D_{Ni}$ of 0.3±0.04 is comparable to our findings and the partition coefficient of Cd ($D_{Cd}$ = 2.6±0.3) reported by Sagar et al., (2021b) is also in the same range. Nevertheless, it should be noted that $D_{Cd}$ is more variable in our data. The applied concentrations of Mn in the culturing medium of our study were higher, while the concentration of Ni was lower and the concentration of Cd was comparable. We added all this information to Figure 5 comparing foraminiferal species and integrated the findings in the discussion (lines 819, 909-910). Furthermore, the study was added to the introduction (line 77).

*Line 76: change 'exvacatum' to 'excavatum'*

-   Done (line 88).

**Material and Methods**

*Line 111: change 'E .excavatum' to 'E. excavatum'*

-   Done (line 126).

*Line 105-106, 116-117, and 124: Were the containers pre-cleaned? If yes, mention the pre-cleaning of the EMSA CLIP and Close boxes used for storing samples from Japsand, and the plastic containers used to collect samples from Kiel Fjord*

- Yes, all boxes were pre-cleaned with Mucasol solution and 5% $HNO_3$ before use. We added this information to the revised version of the manuscript (lines 120, 139).

*Line 118-119: Does this apply to all the 9 cores collected from Kiel Fjord? If yes, change the sentence and include the term 'all the 9 cores'*

- We deleted these lines. See Comment reviewer 2 (lines 133-136).

*Lines 127-128: Any reason why artificial seawater was used for washing and storage of the samples.*

- Artificial seawater was used to ensure that no microorganism that could potentially have harmful effects on the foraminifera were introduced, for instance ciliates. If natural seawater were used, it would have to be sterilized before use (e.g., by autoclavation or filtration). Considering the high volume of water that was necessary to process the samples, it was less time consuming to use artificial seawater. We added this information (lines 143-144).

*Lines 139-141: Mention the cleaning protocols/steps followed for laboratory ware, which were used to handle the foraminifera specimens*

- The following information were added to the revised manuscript:
- Plastic utensils: First of all, a pre-cleaning with Mucasol-water mixture (detergent) was performed to remove oils and other contaminants that could remain from production. Therefore, the laboratory ware was stored in MilliQ water with Mucasol overnight in an oven with 35 °C. Secondly, the Mucasol water was rinsed well and 5% $HNO_3$ was applied for at least 2 days before rinsing (lines 163-164).
- Paint brush and other materials made of different material than plastic were rinsed with Ethanol to avoid any biological contamination (lines 164-165).

*Line 145: 'green colored algae'; mention in Line 107*

- Done (line 122).

*Line 152: cite reference for calcein (16 mg/l)*

- This was a mistake. The Calcein concentration used in this study was 10 mg/l as described by Bernhard et al. (2004) (lines 177-178).

*Line 154: Were temperature measurements carried at the sampling locations? If yes, mention it in the section 'Field Sampling'*

- Water temperature measurements were performed at the North Sea stations, which was added in the revised version of the manuscript (lines 111-114).

*Line 161: Replace 'development' with 'developed'*

-   Done (line 189).

*Line 161: Interchange words 'foraminiferal culturing' to 'culturing foraminifera'*

-   Done (line 189).

*Line 184: Replace 'stable' with 'stably'*

-   Done (line 215).

*Line 218: All salts used were provided in p.a. quality. (does p.a. means pro analysi; please write in full). Please mention the provider of the salts (e.g., SigmaAldrich or CarlRoth or ?)*

-   Yes, p.a. is an abbreviation of "pro analysi". The providers of the chemicals were Carl Roth ($CrCl_3 \cdot 6\ H_2O$; $SnCl_2 \cdot 2\ H_2O$ and $PbCl_2$), Walter CMP ($CdCl_2$) and Sigma Aldrich ($MnCl_2 \cdot 4\ H_2O$, $NiCl_2 \cdot 6\ H_2O$, $CuCl_2 \cdot 2\ H_2O$, $ZnCl_2$, $AgNO_3$ and $HgCl_2$). All this information was provided in the revised version of the manuscript (lines 267-269).

*Lines 200-201: The first, phase 0 ……control phase. No heavy ……added. Combine these two sentences.*

-   Done (lines 231-232).

*Line 210: Reference 'Frontallini et al., (2018a)'' studies the effect of mercury pollution on cultured benthic foraminifera. Frontallini et al., (2018b) studies the ultrastructural alteration in benthic foraminifera induced by heavy metals (e.g., Pb).*

-   The study of Frontalini et al. (2018b) is already included. We were not aware of the mercury study by Frontalini et al. (2018a), which is cited in the revised manuscript (line 249 and 251).

*Lines 229-230: Each experimental phase lasted 21 days (three-weeks) and water with heavy metal concentration was fed into the system bi-weekly. Does it mean that in the three-weeks duration, of each phase, the multi-element culture experiment was fed once with the multi-element spike? Please explain clearly.*

-   The multi element stock solution was added at the beginning of each phase to reach the targeted concentration. Additionally, a smaller aliquot of the same stock solution was introduced twice a week during the three weeks of a phase. The reason was that we expected a loss of metals during the culturing phase (e.g., uptake by foraminifera or algae, adsorption to surfaces of the culturing system). This was clarified in the revised version of the manuscript (lines 239-244).

*Lines 256-257: Reword the sentence to 'Metals (Cr, Ag and Sn) were diluted 1/25 and directly introduced into the ICP-MS as they were not retained on the Nobias resin during preconcentration by the SeaFAST system.'*

-   Done (lines 311-312).

*Line 258: add a comma (,) after samples*

- Done (line 313).

*Lines 260-261. Break the sentence into two. All trace metals except mercury were measured using an Agilent 7500ce quadrapole ICP-MS. Raw intensities were calibrated with mixed standards, which were made from single element solutions covering a wide range.*

- Done (lines 314-316).

*Line 272: Delete 'from'*

- Done (line 327).

*Line 271-272: Check the grammar tense of the manuscript; make it uniform (is/was). Example: Line 268: was reduced, Line 271: is nitrogen-purged*

- Done (lines 326-328).

*Line 283: Replace 'where' with 'were'*

- Done (line 340).

*Lines 286-287: Reword the sentence "In order to check the growth of foraminifera during the culture experiments, the total number of chambers were counted before and after the experiment for every specimen.*

- Done (lines 344-345).

*Lines 294-295: ....removed by rinsing (twice) the sample with Ethanol, which was....*

- Done (lines 355-356).

*Line 311: ICP-MS/MS (do the authors mean ICP-MS)*

- A tandem ICP-MS/MS instrument (Agilent 8900) was used for analysis. This is added in the revised manuscript (line 372).

**Results**

*Lines 377-378: All values the calculations are based on can be found in the appendix. Reword the sentence to make it clear.*

- The sentence was rephrased and important details for a better understanding have been added in the revised manuscript (lines 449-451).

*Line 392-393: In contrast to the text, the figure shows higher Cu/Ca concentrations in the metal experiment of phase 0.*

- This sentence was unclearly formulated and the Cu concentration was higher in the metal systems in phase 0, which was clarified in the revised version of the manuscript (lines 467-468).

*Line 393: The control system in Mn/Ca shows higher concentration in all phases. Reword lines 391-394.*

- Indeed, the Mn/Ca concentration was higher in the control system than in the metal system for the phase 0, 1 and 2, but not for phase 3. This was clarified in the revised manuscript (lines 469-470).

*Lines 395-396: Reword the sentence. 'The variation of the metal concentration was highest in phase 3, in both systems, for all elements but Cu, which showed highest variation in phase 0.'*

- Reworded (lines 471-473).

*Lines 398-399: For every metal phase experiment, were the samples cultured in the spiked multi-element solution (for that particular phase) for the cultured duration of 21 days. Example: For M2 culture, were the samples spiked with M2-concentration (Table 1) for the whole duration of 21 days? Please see Lines 229-230. Please explain.*

- Yes, the culturing water from which the samples were taken, was maintained with M2 concentration for the 21 days. As this concentration was expected to decrease over the 21 days of culturing, a smaller aliquot of the stock solution (Phase 1 = 0.1 ml, Phase 2 = 1 ml, Phase 3 = 10 ml) was added twice a week to keep the concentration stable. This was clarified in the revised manuscript (see above, lines 239-244 and 475-476).

*Line 401: Change to 'In phase 3, metals Cr, Mn, Cu, Ag and Sn......*

- Done (line 480).

*Line 416: Place R2 in brackets.*

- Done (line 496).

*Line 417: It's also indicated.........or not. Change to 'Cases were regression lines were forced through the origin is indicated.'*

- Rephrased (lines 497-498).

*Lines 417-419: In cases when a regression........phases is given. Interchange separately and calculated.*

- Done (lines 497-498).

*Lines 429-432: In the experiments, there are two systems; Control system (no-spike) and Metal system (spike); Line 374. Each system has 4 phases (0,1,2,3), and phase 0 representing no addition of heavy metals (Lines 200-203). Lines 430-431 says ....when the concentration of these metals in the culturing medium was higher. Please explain.*

- Yes, no extra metals were added to the Control system. Nevertheless, there are differences in the metal concentrations, which can be seen in figure 3 and table 4. This information was added (lines 513-516). These changes in concentration can occur due to exchanges with artificial seawater, which contains a certain amount of metals, or by the release of metals like Cu that probably has been leached from brass-made system parts.

*Line 433: does the phase 3 here belongs to metal system. Add to text.*

- Yes. This was added (line 517).

*Line 455: Change 'where' with 'were'*

- Done (line 543).

**Discussions**

*Line 487: Fig. B1; Line 830. The TE/Ca values, for most studied metals, are nearly same for phases 0, 1, and 2 of the metal system, although phase 2 is 10x more than phase 1 (Table 1), and 0 being the control phase. Are there any measurements of spiked-seawater (stock and dilutions) before adding to the culture medium? The culture seawater of phases 1 and 2 (in metal system) should show elevated concentrations (in proportions) compared to phase 0, but is not the case. Please explain why?*

- First question: No, the stock solution was not measured prior to the introduction to the system. No dilutions were made and the stock solution was added directly to the system in different amounts, depending on the concentrations required for the specific phase (lines 239-242).

- Second question: Yes, the metal concentration in phases 1 and 2 should be elevated, which is the case for some elements (e.g., Hg for phase 2). However, there is no elevation visible in most cases. Sorption to surfaces or the uptake of metals by the foraminifera and algae are possible reasons, but no explicit cause could be identified. This was already discussed in chapter 4.1 "Experimental Uncertainties". Therefore, we monitored the metal concentrations so closely (from line 595).

*Lines 493-495: ....are smaller than expected for phases 0, 1, and 2. Phase 0 is mentioned as the control phase with no addition of heavy metals (Lines 200-201). Then why is the metal concentrations of phase 0 smaller than expected (for normal seawater?). is it also because of reasons mentioned in subsequent lines.*

- Phase 0 was accidentally mentioned in this context, which was corrected (lines 596-597).

*Lines 521-522: Is it possible that the low level of food supplies (as inferred from lack of reproduction) might make the cultured foraminifera specimens weaker and relatively lower amount of metal incorporation in them?*

- We assume that the food supply was sufficient because there was leftover food after the experiments and furthermore, the foraminifera calcified, which also requires enough food. See comment above (lines 625-629).

*Line 587: change 'possible' with 'possibly'*

- Done (line 708).

*Lines 615-616: This in turn ..... into the foraminiferal tests. Are the Mn, Zn, and Cu concentrations in the normal seawater (non-polluted) are sufficient as micronutrients considering the fact that these metals are present in the tests of benthic foraminifera recovered from pollution-free environments?*

- The artificial salt used for the culturing medium contained all elements or nutrients that are necessary for marine organisms in a sufficient amount, for most elements at concentrations higher than present in seawater naturally. This information was added (lines 236-237 and 747).

*Lines 625-626: Reword the sentence.*

- Reworded (lines 759-760).

*Lines 634-636: Titelboim et al., (2018) based on their studies report that miliolid shells might have advantage over hyaline as bio archives, since they record higher values than rotaliids from the same ambient seawater. As mentioned, DZn values of this study are in good agreement with results from hyaline as well as miliolid foraminifera. Please discuss the findings of this study with the findings from Titelboim et al. 2018.*

- We fully agree with the reviewer and included Titelboim et al. (2018) in our discussion. The maximum Zn/Ca in our experiments was ~ 68 µmol/mol, which is little lower than reported in Titelboim et al. (2018) (Zn/Ca in *P. calcariformata*= 195 µmol/mol), which may be due to different concentrations in the seawater the foraminifera grew in. In the revised manuscript, we included the findings of Titelboim et al. (2018) (lines 913-926) and compared the different species and calcification modes (high- vs low-Mg calcite) (in 4.3 Interspecies variability, lines from 905).

*Lines 669-670: change 'which is may connected' to 'which is possibly connected'*

- The sentence was removed (lines 809-810).

*Lines 670-676: The mean Pb distribution coefficients obtained from Amphistegina spp. by Titelboim et al., (2021) is 12.9. Please add this to your discussion.*

- The information from this study was displayed in Figure 5 for direct comparison and discussion (line 913). However, it should be noted that *Amphistegina* is a tropical symbiont-bearing species and the symbionts can influence the uptake of certain metals. This could facilitate variations in the incorporation of the metals compared to non-symbiont bearing species as in our study.

*Lines 678-679: Please add the results from Sagar et al., 2021b (partition coefficients for Mn, Ni, and Cd from Amphisorus hemprichii).*

- See Comment above (Figure 5, lines 819 and 909-910).

*Lines 715-717: Figure 4 shows a positive correlation between the concentrations of Cr in culturing medium and in the foraminiferal calcite of Elphidium excavatum. A distribution coefficient of 2.1 has been calculated by the authors for E. excavatum. These results are also stated in lines 707-710. The variability of incorporation in Ammonia spp. and E. excavatum*

*might be because of individual species response. Lines 716-717 are in contrast of the results obtained in this study.*

- We regret this confusion. In the revised manuscript, we clarified that Cr/Ca values of *E. excavatum* calcite are correlated with the Cr/Ca values in the culturing medium resulting in a $D_{Cr}$ of 2.1. We clarified that Cr/Ca values of the two *Ammonia* species are not correlated like this (line 862).

*Lines 735-736: When growth is slower, is there a possibility of weak E. excavatum specimens and lower incorporation of artificially elevated heavy metals in the culture medium than what they should have done with preferred food source?*

- It is possible that a more preferred food source would have stimulated enhanced growth and promoted the incorporation of heavy metal into the shells of *E. excavatum*. For instance, the closely related species *E. clavatum* prefers bacillariophycean diatoms (Schönfeld and Numberger, 2007). It may also be possible that *E. excavatum* is simply a slower growing species than *Ammonia*, which seems not to be necessarily connected to a specific food source (e.g. Haynert et al., 2020). This information was added to the revised manuscript (lines 884-890).

*Lines 719-742: Ammonia beccari, Ammonia tepida, have been cultured, by Havach et. al., 2001; Maréchal Abram et al., 2004; De Nooijer et al., 2007; Munsel et al., 2010, for heavy metal partitioning studies. How do their results compare with the findings of this study for Ammonia batava and Ammonia aomoriensis – a comparison table for the common metals should give a clear picture for Ammonia spp. foraminifera. The same can be done for Elphidum spp.*

- We added a comparison figure for *Ammonia, Elphidum* and other species from different culturing studies (Figure 5).

*Line 745; Table 5: The table is a nice compilation of heavy metal contamination studies in various parts of the globe. The studies referred to in the table have used various natural archives such as water, sediment, bacteria, microalgae, living organisms, and others including benthic foraminifera. A column mentioning the natural archives used by the various researchers is important for the readers. This will help them to not only know the polluted regions of the globe but also give them a quick idea of the archives used for those studies, which might help some researchers to pursue similar studies in the area they live and the best archives available at that place.*

- This is maybe a misunderstanding. Table 5 (now Table A4) presented in this study is only comparing the metal concentrations in the seawater. The studies indeed addressed the metal concentrations in various archives but for the comparison to the metal concentrations in the culturing medium of this study, only the seawater values of the other studies were taken into account. This issue was clarified in the caption of Table A4 of the revised manuscript (line 1029).

*The sequence of some authors needs correction in Table 5. For example: Reddy et al., 2005 moves up Williams et al., 2000 for Ni, Zn. Check for other metals too.*

- This was corrected (see Table A4).

*Line 749: 'During the past years, many studies were performed to assess the pollution level of seawater.' The natural archives used in the study (for example water, sediments, bacteria, algae, and other should be included in this sentence (See above comment for Table 5).*

- We skipped the whole paragraph. See Comment below (start from line 928).

*Line 744: The title of this section is: 'Application of TE/Ca values in foraminiferal shell' – The description in the text talks about the range of concentration used in the culture medium in the current study. The concluding lines of the section (Lines 775-776) says "This means that the concentration range of metals covered by this study is adequate for future research and monitoring of polluted systems". The main point of this research work is to see the incorporation levels of elevated heavy metals in the foraminiferal calcite tests so that they can be used as natural bio archives for monitoring of polluted near-shore marine environments (Abstract Lines 12-13).*

- Indeed, the main subject of this study was to address the incorporation of heavy metals into the foraminiferal calcite for using them as natural archives for this environmental signal. Therefore, we skipped this chapter and moved the information to an earlier part of the manuscript (lines 252-262).

*This section lacks the description on the results (TE/Ca values in foraminiferal shell) obtained in this study from culture experiments with A. aomoriensis, A. batava and E. excavatum and their application as potential pollution indicators. This section needs modification.*

- We agree with the reviewer. The section was removed. See Comment above.

**Conclusions**

*It will be helpful for researchers and readers to pick important findings from this study. Those may be written in point format. For example: '1) All three species showed a strong positive correlation between Pb and Ag in the culture water and their calcite.' The authors should mention the distribution coefficient obtained for these metals from their studies. Others important findings be written in point format.*

- This is a good idea and will help the reader to pick up take-home messages from this manuscript. We changed this (lines 969-983).

*Lines 797-799: 'for those elements' – these elements should be mentioned for a quick take away for researchers.*

- We named "those elements" (lines 976-978).

*Line 801: 'the presented DTE's' – The DTE's obtained be mentioned here – also which DTE should be chosen, with phase 3 or without, be mentioned in this section.*

- We agree and mentioned the $D_{TE}$ values in the revised manuscript (lines 976-978).

**Response to Reviewer#2 comments**

*The authors are going too far in their conclusions: in the abstract lines 25-27 "Our calibrations and the calculated partition coefficients… enable the direct quantification of metals in polluted and pristine environments" and in conclusion lines 801-802 "The presented DTE´s allow a direct quantification of metal concentrations in polluted and pristine areas". First, given the really high DTE ranges found in this study (including or not phase 3) and/or DTE (from linear regression) strongly based on the phase 3 data point where the seawater element concentrations are variable, it is not possible to maintain that "quantification" of metal pollution in natural environments is possible. Secondly, the authors are contradicting themselves since they explain in the introduction that a mix of metal may result in interactions that can lead to different incorporation of the metal. Therefore, the mix proposed in this study, which is peculiar since including 10 trace metals at a time (in polluted environment, most often only 1 or 2 metals are above the threshold limit, not 10 at the same time), is not representative of other type of pollutions. The authors should be more measured in their conclusion. The elemental concentration in the shell may definitely be used to look at relative variation of heavy metal concentrations in the seawater through time and space but definitely not to give quantitative data… and only for elements where a positive correlation has been found between TE/Ca foram and seawater.*

- We respectfully disagree. Any approximation of environmental signals by using the elemental or isotope composition of calcareous shells grown in the respective environment is only that precise as the variability of the calibration data set. The uncertainties are discussed in the paper indeed. Furthermore, the heavy metal mixture we applied was found very often in natural environments, for example in the vicinity of harbors or bigger cities (see Table A4 and lines 248-264). Some examples are given in Table 5 (now Table A4) of our manuscript. This is only a brief overview and much more work on this is available. Therefore, our metal mix is definitely representative for heavy metal pollution in near-shore environments. Indeed, it was intended in this study to investigate the impact of metals that do co-occur and that are potentially interacting (lines 80-87 and 633-636).

*I think that, instead of describing and discussing each trends or absence of trends observed, they should maybe realize that the absence of systematic tendency (within one element or one species) is unexpected and might be the result of multi-metal experiment since singe metal culture exhibit usually positive correlation between shell and seawater element ratios (cf literature). I would also advise to elaborate more the interspecific differences and maybe on the new elements that have never been measured before.*

- We have specifically chosen to conduct experiments that mimic natural conditions as much as possible. This brings added complexity but by carefully monitoring the changes in culturing medium metal concentrations, the results are robust (lines 633-636). It is important scientifically to report non-results but in the revised version of the manuscript, we focused more on the elements with linear partitioning and on species-specific differences (chapter 4.2 and 4.3, e.g., lines 821-822).

*To my opinion, there is confusion between the toxicity of metals to the organism and their incorporation into the shell (cf line 27 "This in turn allows monitoring of the ecosystem 27 status of areas"). What are expected from environmental/governmental studies is to evaluate the impact of heavy metal concentrations on the organism life (ability to survive, grow,*

*reproduce…). Here the authors measure the elemental concentration in the shell. The speciation of the metal incorporated in the shell might be different from the one causing toxicity and bioaccumulation in the cell. The elemental concentration in the shell may help to reconstruct variations of seawater elemental concentration but for the moment, the link between this concentration and its effect on marine life is still unknown (and may depends on elements!). I think that the authors should discuss more precisely about this aspect.*

- We take the message to more precisely distinguish between toxicity, bioavailability and incorporation into the shell at respective sections in the manuscript and rephrased the sentences (lines 31-33).

*I have problem to understand different aspects regarding the metal mix solution:*

1. *How and where was added this solution? Was it added in the supply tank located on top of the system? In this case, knowing that the pump is flowing at 0.017ml/s, how long would it take to replace and reach the same metal concentration in the culturing vessel (ie Tupperware) as in the tank?*

- The solution was added in the supply tank (line 239)and according to the flow rate, the metal concentration in the culturing vessels would be the same as in the supply tank after a few hours. Not only the flow rate and water exchange in the vessels but also sorption processes can have an effect on the metal concentration, which is addressed in the manuscript in detail (lines 242-244 and chapter 4.3).

2. *When was added the solution? According to line 206, we understand that this is added once before each phase. But on line 229-230, it is written "For keeping the metal concentration at the same level over the different culturing phases, water with elevated heavy metal concentrations was fed into the system bi-weekly.". I'm lost, what is this "water with elevated heavy metal concentrations" you are referring to? Is it the stock solution? This need to be clarified…*

- The stock solution was added at the beginning of each phase to reach the targeted concentration. Additionally, a smaller aliquot of the same stock solution, termed "water with elevated metal concentration", was introduced twice a week during the three weeks of a phase, because a loss of metals during the culturing phase was expected, e.g., due to of the uptake by foraminifera or algae or by adsorption to surfaces (see Response to Reviewer 1 above). This was also specified in the revised manuscript version (lines 239-244).

3. *Where are taken the samples for trace metal analyses in the seawater? I don't think the information is given (Lines 215-216, line 234 "from both systems")… Are they taken at the outflow of the vessels so that it really corresponds to the concentration of the seawater in which the foraminifera are growing? Or they were sampled in one of the tank, which would be of course less precise…*

- The water samples were taken from the supply tank. This information is already in the Methods chapter (line 287). This point in the system was considered appropriate because the high flow rate and hence well-mixed system facilitated a representative sampling.

4. *How often was measured the metal concentrations in seawater over the course of the experiment. This should be indicated in the material and method part. For the*

*moment, it is written "frequently" (line 216 and 234) however, when looking at Figure B1, only 1 to 4 data point are available within each phase. This is to my opinion problematic when applying individual curve fit for every phase to calculate the weighted mean value... (see comment after)*

- Fourteen samples were taken from the metal system over the course of the whole experiment, which can be seen in Table A1. From beginning of phase 1, sampling took place once a week. This was added to the Material and Methods section (line 290). Indeed, the metal concentration was expected to be more stable during the culturing phase, which was why sampling once a week was considered as appropriate. Nevertheless, the application of a fit curve for every phase is in our opinion the only way to approach a representative mean value for a given phase especially when taking into account that the metal concentration in the culturing medium varied.

5. *I don't understand the calculation in table 1. The factor between each phase is 10 times but on line 207, it is written "phase 1 = 1 ml, phase 2 = 10 ml, phase 3 = 150 ml". How were calculated these target values?*

- In phase one, 1 ml of the stock solution was added and in phase 2, 10 ml of the same stock solution were added, which is the factor of 10 mentioned above. The target values were not calculated, they were taken from the literature (see lines 248-261). Based on this, we calculated how much of the stock solution was needed to be added to the system for each phase.

6. *The authors used "stock solution" all over the manuscript when referring to the metal mix solution. However, I think it would help to clarify this on line 205 when you first used this term so that it is clear in the discussion (on line 487 and after) that you talk about this stock metal mix solution.*

- This was clarified in the revised manuscript (lines 237-239).

*The culturing system description is very precise but also very long and all the details makes it difficult to understand the general principle. I think that the authors would gain clarity if they explain earlier that a different vessel, freshly filled with calcein labelled forams (if this is correct), is incubated for each different phase of the experiment. At the moment, this essential information appears only (if I'm correct!) on lines 222-223 whereas it should already be said in chapter 2.2.2 or at least beginning of 2.2.4. The description would also be clearer by keeping the same term to describe the same "object".*

- We agree and moved this information to an earlier point in the manuscript (lines 176-180). Furthermore, the description was rewritten taking into account that the same terms are used for corresponding parts of the culturing system (chapters 2.2.2, 2.2.3, 2.2.4).

**Abstract**

*Line 17: "Seawater analysis... between culturing phases". This sentence is not 100% correct since the increases between phase 0, 1 and 2 are not very obvious for all elements (e.g. Cu, Mn). This is however clear for phase 3.*

- This was specified (lines 18-20).

*Line 24-25: I have the idea that Zn and Cd are showing variations that are more or less similar to other elements (eg. Cd like Pb), no?*

- Indeed, the small variation applies to Sn only but not to Cd and Zn. This was specified (lines 29-30).

**Introduction**

*There is confusion between "heavy metals" and "trace metals" throughout the manuscript. To my knowledge, the 10 metals studied here are not all considered as "heavy" metals, some are trace metals. I think that this depends on the atomic weight of the element… Please check and use the appropriate terms.*

- "Heavy metal" as term is not clearly defined. This issue is nicely described in Duffus et al. (2002). The atomic weight is one possible criterion, but no threshold is set for a minimum weight an element must have to be considered as "heavy metal". Some authors pretend that the atomic weight needs to be greater than sodium, which would apply for all of our metals, and others take Hg or Ca as a boundary weight. Another criterion is the density. The boundary value for this parameter is ranging between 3.5 and 7.0 g/cm$^3$ depending on the author. Other criteria involve their behavior as Lewis acid. It is therefore difficult to apply an "appropriate" term, but we defined our use of the term in the revised manuscript (lines 45-46).

*Lines 69-75: here you talk about the physiological effects. This is interesting but you are looking at the incorporation in the shell which is different (cf comment earlier). The information is correct but it has to be clear that you will not have a look at this aspect yourself in this study.*

- See Response to Reviewer 1.

*Line 86: "bioavailability". I guess that this is correct to say that if the element is found in the shell, it is bioavailable since it might (depending on the biomineralisation process involved) goes through the cell. However, I would say that this is different from toxicity effect (cf comment earlier).*

- Bioavailability and toxicity are definitely different and the sentence was reformulated (lines 97-99).

*Line 43: replace "they" by "the composition of their test"*

- This was changed (lines 51-52).

*Line 46: check in Kotthoff et al. (2017) that Mn/Ca is actually used for O2 or redox reconstructions and not for contamination.*

- This is true and is already described (lines 54-55).

*Line 46: add Guo et al. (2019)*

- We added this citation (line 55).

*Line 53: These species are also dominant in intertidal mudflats, not only subtidal areas.*

- This is a misunderstanding. "near-shore" does not mean subtidal as the "shore" is legally defined by the Mean High Water level. As such, mudflats are well near the shore. Perhaps it is more precise to say "intertidal and shelf environments". This was changed (line 63).

*Lines 61-64: add Barras et al., 2018 for Mn*

- We added this citation (line 72).

**Material and Methods**

*It would be nice to document with SEM pictures and light pictures the 3 species of this study. I think it is even more important knowing that Ammonia and Elphidium are species rising lot of identification discussions! Whatever the name given, it is essential to have to possibility to look at the picture and compare it to literature and also recent DNA papers.*

- This is not necessary. The species from our sampling locations are already well documented in the literature (Lutze, 1965; Nikulina et al., 2008; Schweizer et al., 2011; Francescangeli et al., 2021; Schmidt and Schönfeld, 2021). We added this information and citations to the revised version of the manuscript (lines 157-159).

*Lines 118-121: I am wondering if this information is relevant for the manuscript. Since the text is too long, I would suggest to delete this part. Also, the authors mention cores sampled for ecological study which are not presented in the manuscript. This is maybe not necessary?*

- We deleted these lines (lines 133-136).

*Lines 138-150: There are too many details here (eg the size of the petri dish). Some information is repeated several times. For example, the fact that the authors checked several times to be sure that the forams were alive (lines 142-143: "glossy, transparent and undamaged test... cytoplasm present", line147 "structural infill of cytoplasm", line 151 "the color of the cytoplasm was checked"). I don't think the precision of this check at each step is necessary... The important information is that the forams used at the end in the experiment were labelled with calcein and exhibited a green cytoplasm proving that they were active.*

- We shortened this section where possible but avoided losing any important information in the revised manuscript (chapter 2.2.1, lines from 157).

*Lines 151-156: I had some difficulties to understand (when I first read the manuscript) when this labelling step happened? Is it only once at the beginning of the entire experiment (before phase 0)? But in this case, the forams added for example at phase 3 could have calcified new chambers in the meantime... Or is it done before each phase in order to add freshly labelled forams in the new introduced vessels? Here the authors should precise this aspect.*

- The labelling took place before each phase to ensure that freshly labelled foraminifera are inserted in the well plates. This was clarified in the revised manuscript (lines 176-179).

*Although the culturing system is well described, it is difficult to not get lost since everything is described with lots of details. Therefore I would recommend to always use the same term when describing one part of it (eg "vessels" for the box containing the well plates, that you*

*should name this way on line 186). On line 195-196, the term chamber is used but we do not really know to what it refers to: well-plate cavities? Vessel? Please try to keep it simple and clear.*

- We clarified this sections and we used the same terms for respective parts in the culturing system (chapter 2.2.2 to 2.2.4).

*Lines 223-224: One vessel was left from phase 0 to phase 4 (84 days). What was the interest of this vessel? Were the forams from this vessel analysed? If this is not the case, you should say it to avoid any confusion!*

- The intention was to explore the metal incorporation during all four phases in one individual specimen, but the foraminifera have not been analysed yet. We deleted this sentence to avoid confusions (lines 275-278).

*Lines 286-290 "the total number of chambers was counted before and after the experiment for every specimen (Table 2)": I don't see the interest of counting all the chambers of each foraminifera before and after the phase since the authors used calcein. And this information is not given in Table 2. Moreover, I agree that this is possible to count the total number of chamber in Ammonia species since they are trochospiral. However, this is not the case for Elphidium species since spires of new chambers recover the initial chambers! Therefore, if forams where indeed labelled with calcein just before their introduction into the culture system, I would keep it simple and only mention calcein to identify newly formed chambers.*

- Chamber counting was to double check if foraminifera grew during the experiment, because calcein staining may eventually fail. This was stated in the revised manuscript (lines 344-347).

*Line 312-313: Could you explain why you chose to use NIST612 for calibration and monitoring of instrument drift since the elemental concentrations in this standard are way above the concentrations found in the forams? Moreover, you chose to use a glass standard as quality control whereas it would be more appropriate, to my opinion, to use a carbonate standard with similar matrix to your forams. Moreover, the conditions are similar between carbonate standards and forams (I guess) whereas NIST standards are measured with higher energy and frequency. Please explain.*

- The glass standard was chosen because all elements of interest but Hg are reported in the literature, which is not the case for carbonate standards. For further quality control, a variety of carbonate-based reference materials have been measured. All values can be found in Table A3 in the Appendix. Furthermore, Dueñas-Bohórquez et al. (2009) demonstrated that different energy density between the foraminiferal calcite and the glass standard does not affect the Laser ablation analyses. This information was added (lines 374-376 and 377-378).

*Line 334: The authors considered the data as usable if above LOD. However, the limit above which the data can be used for quantitative purposes is commonly defined as the LOQ (limit of quantification). This is defined as 10*SD of the blank. How many data would be excluded from the dataset if the authors use LOQ instead of LOD?*

- No data below the LOQ were interpreted, which was added to the revised version of the manuscript (lines 401-402).

*Lines 364-365: It is not described in the Material and Methods how the living forams were differentiate from the dead ones at the end of each phase. Did the forams lost the colour of the cytoplasm (or their cytoplasm itself) so quickly that you could see it?*

- Indeed, the foraminifera loose the color of their cytoplasm quickly. Furthermore, they do not gather particles or food any more, thus are lacking a detritus cyst before their aperture. This information was added in the Material and Methods section (lines 342-343).

*Line 102: what is Hallig Hooge? Is it still on the field?*

- "Hallig Hooge" is an island in the North Frisian Wadden Sea and yes, it is still there (line 116).

*Line 225: Use PSU everywhere or even no unit at all for salinity.*

- We agree and unified it (e.g., lines 113-114, 142, 198).

*Line 114: add ", respectively" after "and F.S. Alkor"*

- Done (line 129).

*Line 115: "each station" rather than "both stations"?*

- The sentence was deleted (lines 130-131).

*Line 119: change "was found" to "were found".*

- These lines were deleted. See comment above.

*Figure 2 legend: remove "left" and "right".*

- We removed this (lines 149-150).

*Figure 2a: If I understood properly, there were only 2 vessels per incubator so, to avoid confusion, you should remove 6 of the 8 vessels drawn in figure 2a.*

- This was adjusted (Figure 2).

*Figure 2e: this picture is not very clear. Is the shell of the foram entirely fluorescent (ie born in calcein bath)? Otherwise, how many chambers are labelled here? I have the feeling that this is the cytoplasm that exhibit high fluorescence at the bottom since the fluorescence is patchy and fill half of the last chamber...Could you try to show a better picture?*

- No, the shell is not entirely labelled. Only the last 2 ½ chambers are labelled. It can be excluded that only the cytoplasm is fluorecenting because the specimen was dead, cleaned and dried. Therefore, no cytoplasm should be there anymore. This information was added to the figure caption (lines 152-154).

*Line 152: Why did the authors used a concentration of 16mg/L which is different from the recommended concentration given by Bernhard et al. (2006)?*

- See Reply to Reviewer 1 (lines 176-179).

*Line 156: To my opinion, this is not enough time to remove the calcein from the vesicles in the cytoplasm. Anyway, if this seawater is used to calcify 1 new chamber in your experiments, you can hope that this new chamber would exhibit a small fluorescence.*

- Reviewer 2 is right. A sufficient time is needed to remove the calcein from seawater vesicles in the cytoplasm. If the foraminifera are taken directly from the calcein staining bath for incubation, all subsequent chambers will be stained (see Haynert et al., 2011). In our case, the youngest chambers were not stained in that a purification time of 1 or 2 days was sufficient. We added this (lines 574-576).

*Line 160: Dagan et al., 2016 is a report. Is it available online somewhere?*

- The report is not available to the public but Woehle et al. (2018) reported the experimental setup as well in the online supplement. Dagan et al. (2016) is therefore deleted (lines 191, 210, 1158-1161).

*Line 171: it is the air that was filtered?*

- Yes. This is clarified in the revised version of the manuscript (line 199).

*Line 171: The authors do not mentioned pH or alkalinity measurements. Did they measure carbonate chemistry during the experiments? At least pH has been measure since it is mentioned in discussion on line 580 "As the pH during the experiment was stable around 8.0 ±0.1 (measured twice a week)". This information should arrive in material and methods.*

- Carbonate chemistry was not measured during the experiment. We added the information that pH was measured in the "Material and Methods" chapter (lines 201-202).

*Line 221: Give the flow rate in ml/min. This would be more adapted.*

- This was adapted (line 272).

*Line 225: Please give SD of the data.*

- The SD was added (lines 279-280).

*Line 313-314: The authors specify the conditions used to measure glass standards. Could they precise the ones used for carbonate standards?*

- This was specified (lines 381-382).

*Line 323: remove "are expressed"*

- Done (line 389).

*Line 331: change "(Rosenthal et al., 1999)" to "Rosenthal et al. (1999)"*

- This was changed (line 397).

**Results**

*Table 2: In C2 for A. aomoriensis, does it mean that on the 10 forams recovered, 2 were dead but all of them (10) had calcified new chambers?*

- Yes.

*Line 368: Since the Ammonia calcified usually more than 4 new chambers, is it possible to see the evolution of seawater metal concentration in the successive chambers of 1 given individual? At least in phase 3? This could help to gain precision in the estimated DTE…*

- The evolution of the metal concentration in seawater of phase 3 was only indicated in some individuals of *Ammonia aomoriensis* and *Ammonia batava*. Particularly, the first high concentration of certain heavy metals could be found in the first chambers after the staining (i.e. the first chamber built in culture). But this was not the case for all individuals, which is most likely due to the individual timing of calcification. It also cannot be determined, at which point in time the foraminifera calcified within one phase. Therefore, a mean value over the whole culturing phase is most representative. We added this information to the revised manuscript (lines 897-902).

*Figure B1: Could you indicate the error of the measurement on the graph? ON line 340-344, the authors explained that they fit a regression curve on the data to calculate a weighted mean per phase. This seems a good idea when 4 data points are available within a given phase and that a trend can be seen (eg phase 3 for Cr, Ag, Sn). However, this seems difficult when only 2 data points are available and very different (eg Cu) or when the trend is not regular (eg phase 3 for Mn, Ni…). Actually, did you realise that Mn, Ni, Zn and Cd show similar variation though time in phase 3 (lower value at the second sampling time) compared to Cr, Ag, Sn or Pb which show decreasing trends?*

- As these are single measurements, the error that could be provided would be based on frequent measurements of the seawater reference material. The respective values are already given in Table A2 and are therefore not included into figure B1. Furthermore, Figure 3 also displays the SD of the measurements.

- When only 2 data points are available a linear regression was made, which is in our opinion the only way to account for the different concentrations because we do not know at which time within a phase the new chamber was built. If no clear trend was observed, the regression with the highest fit (highest p- and $R^2$-value) was chosen. This information was added (lines 410-411).

- It is indeed interesting to note that Mn, Ni, Zn and Cd show similar patterns in phase 3.

*Figure 3: I have the idea that the use of weighted means and standard error of the mean instead of standard deviations, the authors reduce artificially a lot the real elemental variations that they have, mainly in phase 3. Maybe the figure could be completed showing the range of values actually measured in shadow or use box plot to better represent the variability of this artificially created dataset...*

- It is true, that the variation carry less weight in this figure and this is why we added figure B1 in the appendix. Nevertheless, the variability in the seawater during one phase was expressed as the SD instead of the SE in the revised version of the manuscript (Figure 3).

*Line 427: remove Cu in "Mn, Ni, Cu, Hg and Sn".*

- "Cu" was removed (line 508).

*Figure 4:*

*How are calculated the statistics of the correlations? These correlations should not be based only on the mean values per phase but on the all data set. For example for Ag and Pb, the R² and p values are really good but the D is only based on the Phase 3 data which has a high variability! Therefore the D value is not precise and robust.*

- The statistics of figure 4 are indeed based on the mean value per phase and not on the entire data set. The plots were made using the software Grapher, which is calculating statistics along plotting. Furthermore, the program PAST was used to calculate statistics. We actually also calculated $R^2$ and p-values based on the whole data set, which was e.g., for *A. aomoriensis* with phase 3 comparable to the statistics based on the mean. This is why we decided to rather consider the mean values. We added this information in the revised manuscript (lines 539-541).

*Figure 4 and Table 4: The authors have no objective reasons to fit the correlation through 0 for some elements and not for others. It could be decided on statistical arguments but I have the idea that the authors did not check this.*

- We tried to fit all element correlations in all species through the origin, because a real correlation would also include the origin. Only in cases where this was clearly not possible (Mn of *A. batava* with phase 3 and Hg of *E. excavatum* without phase 3), because the course of the regression line changed significantly or the $R^2$ value decreased, no forcing through the origin was applied. We clarified this in the revised version of the manuscript (lines 421-426).

*For A. aomoriensis Mn/Ca, there is a problem with the correlation line. This is not possible that the line don't go through the phase 3 datapoint. Please check.*

- The line is not going through the data point from phase 3 because the line is forced through the origin, which is changing the course of the line minimally. It should be clear to the readership that not only the data point from phase 3 but also the data from the other phases are driving the course of the regression line. Furthermore, the $R^2$-value of the regression line did not decrease when forcing through the origin, this is why we decided to include the origin. We went further into detail in the revised manuscript version (lines 421-426).

*The graphs for this figure should have similar y axis range for a given element for the 3 species so that the difference of incorporation between species is highlighted. All graphs should start at 0on the y and x axis. I think that the main (and most robust) output of this study is the difference of incorporation between Ammonia and Elphidium species and this is at the moment only shortly discussed and observable in graphs. This is a shame.*

- The axes were adjusted. In the revised version of the manuscript, all axes start at 0, both *Ammonia* species have the same axis, and for *Elphidium excavatum*, a different axis was used if necessary. These adaptations make a comparison between the three species easier. Furthermore, the SD of the slope of the regression line (= partitioning coefficient $D_{TE}$) was added in the figure and the corresponding table (Figure 4 and Table 4). We also went more into detail in comparing *Ammonia* and *Elphidium* and

added Figure 5, which further helps to distinguish between the species (chapters 4.2 and 4.3).

*This is a really good idea that the authors also analysed their data without the data from phase 3. To my opinion, this phase is important to get a trend because the problem when you remove it is that you have no correlation anymore, probably because the range of seawater elemental concentration is not wide enough. On the other hand, when phase 3 is considered, then a more relevant D value can be calculated but the correlation are only based on this data points and therefore the correlation is not statistically robust.*

- Yes, it is true that the correlation gets lost because the range of the metals in the seawater is very narrow without phase 3. Furthermore, it is also true that point three makes the correlation statistically more robust but nevertheless, figure B2 also shows that the general trend is still visible without phase 3 for some elements. Forcing through the origin further adds a fix point, which provides 3 points, though artificially, and not 2 only. We clarified this in the revised version of the manuscript (lines 630-632).

*Line 459-463 and line 467-468: Repetition compared to what is written in mat and met and in the legend of figure 4 and table. Delete this paragraph.*

- We deleted this paragraphs (lines 547-551 and 555-556).

*All sentences starting with species name, the genera name should be written entirely.*

- We adapted this (line 562).

*Line 473: Now authors are removing phase 3 and 2?*

- We did not remove phase 2 from the calculation of the regression line but if one has a look at every data point from a phase individually (meaning without any calculation of regression), the $D_{Cd}$ and $D_{Cr}$ values from phase 0 and 1 are >1 while the $D_{Cd}$ and $D_{Cr}$ from phase 2 and 3 are <1. We deleted this sentence to avoid confusion (lines 561-562).

*Line 464 to 481: this is very descriptive and difficult to follow…*

- As this is part of the "Results" section, a description of the data is appropriate. We rearranged the paragraph to make it easier to follow for the reader and focused more on the results that are discussed later in the manuscript (see paragraph from line 552).

**Discussion**

*It is not possible to discuss the significance and meaning of partitioning coefficient that are showing a very high range since this variation is meaningless to my opinion in terms of biomineralisation processes… For example, DCd are varying from values below 1 to values such as 10-20 even 50 in all species (lines 678-679). In terms of incorporation mechanisms, that would mean that some specimens are fractionating against Cd whereas some others (from the same species and in the same condition) would concentrate this element! I would suggest to the author to rather focus on:*

*Elements were a positive correlation is found but instead of using the mean TE/Ca value (eg line 557-559), they should take into account the variability of the data and give a SD for the slope (ie for the DTE). They should also be aware and acknowledge in the manuscript that these correlations are driven by the phase 3 data and might be imprecise.*

- We added the SD for the slope of the partition coefficients and went more into detail concerning the uncertainties of the calculated $D_{TE}$ (Figure 4 and Table 4). It is also reasonable to note that the correlation is driven by the data point from phase 3. We added this to clarify the circumstances for the reader in the Experimental uncertainties chapter (lines 630-638).

*Elements were the range is relatively low so that a general tendency/interpretation might be given.*

- We focused more on elements with a smaller variability (chapter 4.2).

*Finally, do not discuss further forward the other elements that exhibit vary wide DTE also if no literature is available on this element, it is interesting to know that this is incorporated and measurable in foraminiferal calcite.*

- We shortened the discussion of the elements with higher variability (see chapter 4.2, e.g., lines 821-822). But nevertheless, a proxy is only as good as its variability, and therefore we think that it is important to mention variable $D_{TE}$´s too.

*I have the feeling that the authors use the DTE with or without phase 3 when it helps them to compare with the literature. This is bothering me: is phase 3 really usable to calculate a partitioning coefficient knowing that the seawater concentration of the metal was not stable during this phase and the regression line is totally driven by this single condition?*

- The regression is not only driven by the data point from phase 3, because other points and the origin also play a role, which is already demonstrated above and can be seen in Figure B2. But nevertheless, phase 3 is very much driving the slope of the regression line. Even though the seawater concentration was not as stable as during phases 1 or 2, we are convinced that it is appropriate and justified to use a mean value calculated from the individual fit curve for every element and to establish it as mean value of the foraminiferal calcite. It is possible, that the variability of the seawater concentration in our study is higher because we measured more often than other studies did. This means that other studies simply not monitor the variability. Furthermore, pollution events in nature are also transient events rather than stable emissions. We discussed this further in the "Experimental uncertainties" section (lines 590-594).

*Line 506: The authors mentioned the growth of algae as a reason for element concentration changes in the seawater but I understood that the algae were given dead. Therefore, one would not expect algal growth in the experimental set up?*

- This is a misunderstanding. The algae that were fed were dead, but germs of other algae were introduced without purpose together with the living foraminifera and grew during the experiment. These algae preferentially grow on plastic surfaces and create biofilms. Therefore, it is well possible that these films also took up metals. We added this information (lines 609-610).

*Line 523-528: this paragraph should be more or less upside down. Since you used calcein prior to the experiment, you do not have to worry that this probe could have impacted the elemental concentration in your forams. This paragraph could therefore be shorten.*

- We agree and moved the calcein paragraph to the beginning of the chapter (lines 574-581).

*Line 532: Since most of the references concern culture experiments, I would suggest to add Barras et al., 2018 for Mn.*

- This was added (line 648).

*Lines 551-552: according to Erez endocytosis biomineralisation, I thought that the composition of the seawater vesicle (ie Mg content) was also modified somehow?*

- Yes, this is partially correct. "Endocytosis" as such describes only the uptake of a seawater vacuole, which is subsequently modified during their pathway in the cell. We changed this sentence (lines 668-672).

*Lines 559-561: this is interesting but where can we see this information (ie. D vs seawater trace element concentration)?*

- Figure 4 shows this indirectly, we referred to this figure (line 680).

*Line 559: if D>1, this means that the foram is concentrating the element inside its shell. Therefore, I would not define this as a "non-selective uptake", no?*

- "Non-selective" at this point refers to an uptake that is not driven by the chemical property or the size of the metal ion itself. This was clarified in the revised version of the manuscript (lines 674-678).

*Line 561: other studies have observed the same trend of decreasing D with increasing seawater concentrations: Mewes et al. (2015) for Mg and Barras et al. (2018) for Mn.*

- The references were taken into account (line 681).

*Figure 5: The authors refer to this figure for each element but I think that this is also interesting to observe that there is apparently no trend between D and the ionic radius to charge ratio.*

- Yes, this is true, but this figure should mainly provide information whether the $D_{TE}$ is higher or lower than 1 to the reader. We decided to removed this figure.

*Figure 5: it is strange to me that the author used a single data point for each LogD value. Is it the mean of all measurements? In this case, it would be nice to see the SD since D might be highly variable.*

- See Comment above.

*The authors compare their D values to the literature. Sometimes they compare these values to tropical symbiont-bearing large benthic forams (high Mg content species) or miliolids (line 635, 671, 708) which are known to incorporate much more elements than Ammonia for example and other small benthic foraminifera (low Mg species) (cf van Dijk et al.,2017). This should be specified and discussed.*

- See Reply to Reviewer 1.

*Chapter 4.3: as mentioned before I think that Figure 4 should be reworked (or a new figure) in order to observe more easily the differences between species (e.g. similar axis for Ammonia species and different (if needed) for Elphidium). Maybe differences between species would be even better observed when considering only phases where the seawater elemental concentrations are stable?*

- Figure 4 was adjusted. See Comment above. We added Figure 5 comparing different species.

*Line 724-725: Food is added quite regularly during the experiment. Could the deposition of a layer of food at the surface of the "sediment" could create microenvironments within the hole of the weel-plate? Indeed, the food would be degraded and could influence pH and O2 conditions for exemple…*

- Indeed, the leftover food deposited as a thin layer on top of the sediment, which could have created a microhabitat. This effect would be the same for all cavities and therefore for all three species. In account of this, species–specific differences in the heavy metal incorporation cannot be caused by this effect. This was mentioned in the revised version of the manuscript (lines 870-872).

*Line 727: Read van Dijk et al. (2017) paper but I don't think that the hhigh elemental incorporation of symbiont bearing forams is due to the presence of symbionts but rather to the fact that they are high-Mg content species. Other symbiont barren large benthic forams exhibit high elemental incorporation.*

- We agree, the high Mg-content of the calcium carbonate of the species in the tropics at high temperatures and salinities could play a role, which is discussed in the revised version of the manuscript (Figure 5, lines 905-907).

*Line 735-737: be aware that there is a difference between number of chamber added (individual growth rate) and calcification rate (crystal growth rate). Depending on the element, one could expect that slower calcificiation would give more time to remove (or discriminate more against) the element as it is the case for example for Mg.*

- This is an interesting aspect, which was included into the discussion (lines 888-890).

*Line 738-742: this is an interesting point. I think the authors could potentially unravel this problem if they compare Elphidium data with the first chambers calcified after the calcein stained chamber. Indeed, that would be the forst chamber calcified in the experiment when the seawater elemental concentration was probably the highest. Moreover, as previously mentioned, you could have a look at successive chamber composition to see if you can*

*observe a decreasing elemental composition for the elements exhibiting decreasing trend in seawater.*

- *Elphidium* mostly build only one chamber, which means that the data presented here are already from the first chamber calcified after staining. This makes a tracking of the decreasing concentration impossible. For both *Ammonia* species, see comment above.

*Table 5: this table is very interesting and complete but to my opinion, it could be moved in supplementary materials.*

- We moved the Table to the appendix (Table A4).

*Table 5 : how were the metals analysed in these studies? Analytical techniques used? Extractions? Speciation of the metal?*

- We added the information, which pretreatment and analytical technique was used to determine the heavy metal concentration in the comparing studies (Table A4).

*Line 488: "…after a while". Please be more specific.*

- "A while" cannot be specified further, because water samples were taken once a week. The equilibration is expected to be much faster, because the system was well mixed (see comment above).

*Line 587: change "possible" to "possibly"*

- This was changed (line 708).

*Line 592: change "effected" to "affected"*

- This was changed (line 715).

*Line 707: this paper from Remmelzwaal refers to post-depositional overprinting. I don't kown this study but are you sure that this DCr corresponds to primary calcite values?*

- Yes. They performed culturing experiments with different foraminiferal species and calculated this $D_{Cr}$ based on these experiments.

*Line 729: change "Cesborn" to "Cesbron"*

- This removed this sentence and therefore also the citation.

*Lines 753-758: too long…*

- We deleted the whole paragraph and gave the information in an earlier point of the manuscript (lines 252-264).

**Conclusion**

*I think that the authors could highlight the interest to use fossil records (or regular sampling of living forams through time or space) to determine the relative variations of seawater metal concentrations in porewater through time. Although quantitative reconstructions are to my*

*opinion not feasible at the moment, relative variations are usable for elements where a correlation was observed between shell and seawater ratios (not for all elements). The authors should be more realistic in their conclusions.*

- We agree, reference to the fossil record was given in the revised version of the manuscript (line 999).

*The authors could also highlight the interest of forams as they are integrating in their shell the metal concentration over a certain period of time. Indeed, dissolved metal concentrations measured directly in seawater (for monitoring purposes) give the concentration the day of the sampling but this concentration may vary very rapidly… Both aspects should even be mentioned already in the introduction.*

- We agree that foraminifera offer the opportunity for long- and short-term monitoring of changes in the heavy metal concentration, because they are recording the environmental signal. This added this in the introduction and in the conclusions (lines 60-61 and 1000-1002).

*Line 795-796: ok but there is no impact on survival or growth in your experiments.*

- Any organism reacts in a protective way before harmful or lethal effects do occur. This is also why a reduced incorporation of a certain metal could point towards the onset of a protective mechanism prior to damage of the organism and may also prior to a reduced growth or death. This was paragraph was rearranged and reworded (lines 984-1005).